# Frequency-Balanced Retinal Representation Learning with Mutual Information Regularization

**Seunghoon Lee**[1], **Seongjae Kang**[1], **Inhyuk Park**[1],
**Gitaek Kwon**[1], **Jihyeon Baek**[1], **Doohyun Park**[1,*]
[1]VUNO Inc.
aanna0701@gmail.com, {seongjae.kang, inhyuk.park}@vuno.co
{gitaek.kwon, jihyeon.baek, doohyun.park}@vuno.co

## Abstract

We propose a frequency-oriented perspective on retinal representation learning by analyzing masked autoencoders (MAE) through the lens of spatial frequency. Our analysis shows that MAE favors low-frequency content while under-encoding diagnostically critical high-frequency structures in retinal images. Because retinal pathology often manifests in high-frequency detail, this bias limits diagnostic performance and motivates frequency-balanced representations. Within a mutual-information (MI) formulation of MAE, we introduce the *Frequency-Balanced Retinal Masked Autoencoder (RetMAE)*, which augments the reconstruction objective with a MI regularizer that suppresses low-frequency redundancy and accentuates clinically salient high-frequency information. Without altering the architecture, RetMAE learns frequency-balanced features that surpass those of MAE-based retinal encoders in both quantitative and qualitative evaluations. These results suggest that a frequency-oriented view provides a principled foundation for future advances in ophthalmic modeling, offering new insight into how MAE's reconstruction objective amplifies ViT's low-pass tendencies in spatially heterogeneous retinal images and enabling a simple MI-based correction that improves retinal encoders.

## 1 Introduction

Vision foundation models learn generalizable representations from large-scale pre-training, transferable to diverse downstream tasks. This paradigm shows promise in medical imaging, particularly fundus photography, where specialized domain knowledge is crucial. In the fundus domain, recent foundation model approaches have explored two primary directions: 1) self-supervised learning (He et al., 2022; Oquab et al., 2023; Fang et al., 2023; 2024b) and 2) vision-language pre-training (Radford et al., 2021; Wang et al., 2022). Self-supervised learning approaches design pretext tasks that generate supervisory signals directly from the unlabeled data, including masked autoencoders (MAE) which reconstruct masked image patches without requiring manual annotations (Zhou et al., 2023). Vision-language pre-training approaches leverage contrastive learning to learn joint representations by aligning vi-

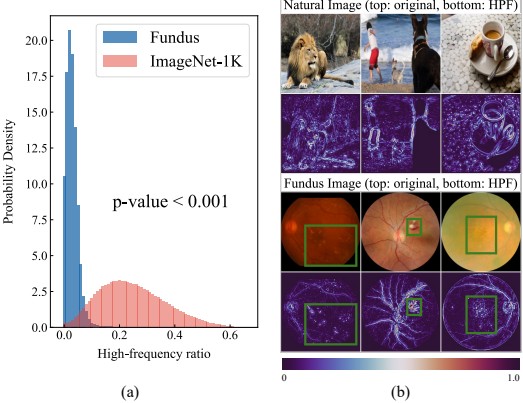

Figure 1: **High-frequency in natural vs. fundus images.** (a) Histogram of high-frequency ratios for fundus and ImageNet-1K (Appendix A.1). (b) High-pass filtered examples: fundus high-frequency content concentrates near lesions, the optic disc, and vessels (**green boxes**).

---

*Corresponding author

sual features with clinical text descriptions (Du et al., 2024; Wu et al., 2024; Silva-Rodriguez et al., 2025; Yu et al., 2024). However, high-quality paired image–text data are scarce and costly in fundus imaging, which limits the practicality of vision-language models. As a result, self-supervised methods such as MAE, which leverage abundant unlabeled retinal images, are more suitable for most publicly available datasets.

However, MAE encounters fundamental limitations when applied to retinal fundus imaging due to its unique characteristics. Unlike natural images, fundus photographs exhibit a distinctive frequency distribution as described in Fig. 1: diagnostically critical structures are sparse and concentrated in high-frequency bands, while the majority of image content comprises smooth, low-frequency backgrounds (Barriga et al., 2009; Agurto et al., 2011; Jindra, 1993; Zhang et al., 2022; Yu et al., 2025). The standard MAE reconstruction objective with random masking and pixel-wise losses implicitly assumes uniform information density across regions, inducing a bias toward smooth, low-frequency backgrounds while underrepresenting the sparse but clinically crucial high-frequency details required for reliable disease recognition. This goes beyond previously observed low-pass tendencies of ViTs by revealing a mismatch between MAE's uniform-information assumption and the strongly heterogeneous spatial distribution of diagnostic signal in retinal images.

In our preliminary study in Section 4, we systematically analyze two aspects using centered kernel alignment (CKA) (Kornblith et al., 2019) between MAE features and frequency-separated inputs: (1) how standard MAE representations exhibit a preference for low-frequency over high-frequency components, and (2) how this frequency bias affects downstream diagnostic performance. We apply the discrete Fourier transform (DFT) with a Butterworth (Butterworth et al., 1930) filter to separate fundus images into high-frequency (vessel boundaries, lesion edges) and low-frequency (smooth backgrounds) components. Our analysis reveals a critical mismatch in standard MAE (Table 1): while MAE representations strongly align with low-frequency components (CKA = 0.990), they poorly capture high-frequency structures (CKA = 0.164). Downstream linear probing performance further shows the opposite dependence—high-frequency components outperform low-frequency ones across multiple datasets, achieving higher macro-average area under the receiver operating characteristic curve (AUROC) (0.641 vs. 0.727). In retinal fundus photography, this inverse relationship between representational alignment and diagnostic utility indicates that standard MAE preferentially encodes the least informative (low-frequency) band. These findings motivate us to propose a frequency-balanced approach to retinal representation learning.

To this end, we propose the *Frequency-Balanced Retinal Masked Autoencoder (RetMAE)*, a pretraining framework that addresses frequency imbalance in fundus images under the mutual information (MI) principle. From an information-theoretic perspective, MI provides a principled basis for learning representations that are compact yet diagnostically sufficient. RetMAE instantiates this principle with a novel objective—the *High-frequency MI regularizer (HighFreqMI)* (Fig. 2)—which prioritizes the efficient encoding of sparse, clinically important high-frequency signals without requiring paired-text supervision, while attenuating low-frequency redundancy. Importantly, no architectural changes are required—performance gains arise from the MI objective alone. This objective encourages *frequency-balanced retinal representations* that suppress irrelevant content while retaining essential diagnostic cues.

Our main contributions are as follows:

- **Frequency bias of MAE:** We show that standard MAE pretraining under-encodes clinically salient high-frequency information while over-representing low-frequency background. Building on these findings, we introduce RetMAE, which incorporates the High-frequency MI regularizer (HighFreqMI) to learn frequency-balanced retinal representations.
- **MI-regularized compactness and sufficiency:** Through comprehensive representational analyses, we demonstrate that HighFreqMI yields embeddings that are both compact and sufficient—reducing redundancy while preserving clinically meaningful features—thereby providing a principled mutual-information-based correction that directly targets MAE's under-utilization of high-frequency retinal structure.
- **Data efficiency without paired text:** RetMAE consistently outperforms retinal foundation models, including MAE-based encoders, and also competes favorably with non-MAE paradigms (text-guided and vision–language models), while requiring substantially fewer pretraining samples. Using only ∼25.6k unlabeled fundus images, it achieves a macro-average AUROC of 0.940 across five benchmarks, highlighting strong data efficiency without paired text.

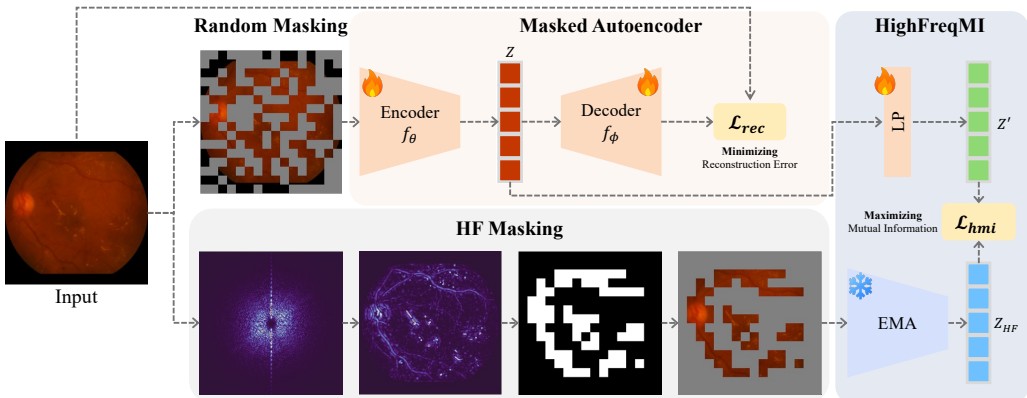

Figure 2: **Overview of RetMAE.** RetMAE extends MAE with a mutual-information regularizer. High-frequency MI (HighFreqMI) aligns the encoder to a high-frequency context using frequency-selected (HF-masked) patches. Here, $Z'$ are linearly projected latents and $Z_{HF}$ are high-frequency latents; HF, high frequency; EMA, exponential moving average; LP, linear projection.

## 2 RELATED WORKS

**Fundus foundation models.** Large-scale pretraining on fundus images has enabled strong retinal encoders. RETFound (Zhou et al., 2023) shows that masked image modeling (MIM) on unlabeled fundus photographs can transfer across retinal diseases. Multimodal approaches—RET-CLIP (Du et al., 2024), KeepFIT (Wu et al., 2024), and FLAIR (Silva-Rodriguez et al., 2025)—further infuse expert knowledge by aligning images with diagnostic reports, while UrFound (Yu et al., 2024) embeds anatomical priors via anatomy-guided masking within MIM. Masking-based fundus hierarchies (Lin et al., 2025) likewise employ masking-and-reconstruction to learn disease-indicative features, but focus on stage-robust, spatially invariant cues rather than explicitly correcting the low-frequency bias of MAE. Our frequency-balanced MI regularizer instead steers the MAE bottleneck toward lesion-centric high-frequency tokens, and could in principle be combined with such hierarchical objectives. However, in clinical settings, high-quality paired image–text corpora are scarce and expensive to curate, which constrains language-supervised scaling. Our work targets this regime: we keep the backbone architecture fixed and avoid paired text, using MIM on modest unlabeled collections together with an optional, off-the-shelf, retina-informed context latent.

**Frequency structure.** Classical analyses show that low-frequency bands capture coarse, global appearance, whereas high-frequency bands encode fine structure (Oppenheim et al., 1979; Oppenheim & Lim, 1981; Piotrowski & Campbell, 1982; Hansen & Hess, 2007). In retinal fundus images, clinically salient lesions (microaneurysms, exudates, and hemorrhages), drusen-related textures, and spectral changes in retinal nerve fiber layer (RNFL) are predominantly contained in high-frequency components (Zhang et al., 2022; Yu et al., 2025; Barriga et al., 2009; Agurto et al., 2011; Jindra, 1993).

Recent MIM variants leverage Fourier or band-aware objectives on frequency-domain inputs or reconstruction targets (Xie et al., 2022; Wang et al., 2024b). In contrast, we regularize semantic latent representations derived from raw high-frequency regions, making our approach complementary.

**MI-based representation learning.** From an information-theoretic perspective, the information bottleneck favors encoders that preserve task-relevant content while discarding nuisances (Tishby et al., 2000; Tishby & Zaslavsky, 2015), and MAE admits an MI formulation linking inputs, masked regions, and latents (Huang et al., 2025). Within this view, RetMAE couples complementary MI regularizers to suppress low-frequency redundancy and amplify diagnostically informative high-frequency cues, yielding frequency-balanced retinal representations without architectural changes or paired text.

## 3 PRELIMINARIES

We briefly review MAE, an information-theoretic interpretation of its objective, and the high-frequency token scoring procedure that underpins our regularizer.

**Masked autoencoders.** MAE (He et al., 2022) randomly masks a subset $\mathcal{M}$ of image patches and trains a ViT-based encoder–decoder to reconstruct the missing content from the remaining visible patches. We consider an image that has been partitioned into $N = HW/P^2$ non-overlapping patches of size $P \times P$, and denote by $\mathbf{x}_i \in \mathbb{R}^{P^2 C}$ the vectorized pixel values of the $i$-th patch for $i = 1, \dots, N$, where $N$ denotes the number of patches per image. Let $\mathcal{M} \subset \{1, \dots, N\}$ denote the masked indices and $\mathcal{V} = \{1, \dots, N\} \setminus \mathcal{M}$ the visible ones. The reconstruction loss is computed only on the masked patches using mean squared error:

$$\mathcal{L}_{\text{rec}} = \frac{1}{|\mathcal{M}|} \sum_{i \in \mathcal{M}} \|\mathbf{x}_i - \hat{\mathbf{x}}_i\|_2^2 \ , \tag{1}$$

where $\hat{\mathbf{x}}_i \in \mathbb{R}^{P^2 C}$ denotes the reconstructed pixel vector of the $i$-th patch. This objective encourages the encoder to infer masked content from visible context and yields transferable features for downstream tasks (Kong & Zhang, 2023).

**An information-theoretic view.** MI offers a principled lens on representation learning (Tishby et al., 2000; Tishby & Zaslavsky, 2015). A recent analysis (Huang et al., 2025) shows that the MAE objective can be viewed as minimizing the Lagrangian

$$\mathcal{L} = I(X_V; Z) + \beta\, I(X_V; X_M \mid Z) \ , \tag{2}$$

where $X = [\mathbf{e}_1, \dots, \mathbf{e}_N]$ denotes the sequence of patch embeddings $\mathbf{e}_i \in \mathbb{R}^D$ obtained from an image (e.g., via a linear projection), with $D$ the embedding dimension. We denote the visible and masked subsets by $X_V = [\mathbf{e}_i]_{i \in \mathcal{V}}$ and $X_M = [\mathbf{e}_i]_{i \in \mathcal{M}}$, respectively, and let $Z$ be the encoder's latent representation of $X_V$. The scalar $\beta > 0$ is a weighting coefficient, and $I(\cdot; \cdot)$ and $I(\cdot; \cdot \mid \cdot)$ denote (conditional) Shannon mutual information (Shannon, 1948).

In deep networks, successive layers compress inputs into internal representations; when this compression is too strong, task-relevant information can be lost, a phenomenon known as *information distortion* (Tishby et al., 2000; Tishby & Zaslavsky, 2015). From this viewpoint, Eq. 2 can be interpreted as minimizing a Lagrangian that explicitly trades off the complexity of the latent representation against such distortion: the first term $I(X_V; Z)$ quantifies the complexity of the latent description $Z$, whereas the second term $I(X_V; X_M \mid Z)$ plays the role of an information-distortion term whose minimization drives $Z$ to retain sufficient information to predict $X_M$.

This view motivates MI-based regularization of the MAE bottleneck beyond the standard reconstruction loss, and Huang et al. (2025) shows that constraining the complexity term $I(X_V; Z)$ improves MAE performance on natural images. Building on the same information-bottleneck formulation, we introduce a domain-specific high-frequency regularizer: instead of enforcing mask invariance as in MI-MAE, we align $Z$ with a high-frequency retinal context so that the bottleneck prioritizes clinically salient high-frequency structure over low-frequency background in fundus images.

**High-frequency extraction.** For each fundus image, we first apply a Soft-FOV mask to the green channel, which provides the strongest vessel and lesion contrast (Biswas et al., 2022; Ooi et al., 2021; Kumar et al., 2020). We suppress low-frequency content via Gaussian blur, transform the result to the Fourier domain (Brigham, 1988), and apply a Butterworth high-pass filter (Butterworth et al., 1930) tuned on a small held-out set with vessel/lesion annotations; the inverse transform yields a high-pass response map. We then reapply a binarized Soft-FOV mask to attenuate residual background and boundary responses, and compute a scalar high-frequency score for each ViT (Dosovitskiy et al., 2020) token by averaging the masked response over its corresponding non-overlapping $P \times P$ patch. Tokens in the top $25\%$ of scores are treated as high-frequency tokens for our regularizer; implementation details and hyperparameters are provided in Appendix A.2.

## 4 Uncovering Frequency Biases in MAE Representations

We assess whether a standard MAE with a ViT backbone captures diagnostically salient high-frequency content by pretraining on fundus images (see Sec. A.5) and evaluating the encoder using CKA alongside linear-probing AUROC (see Sec. 6.1). Because ViT is a token-based architecture, changing the visible token set naturally induces different internal representations, and Kong & Zhang (2023) uses CKA to compare such representations under different training schemes. Following this perspective, Table 1 reports a CKA-based comparison across token subsets. More specifically, we

treat the representation from the *full*, unmasked input as the baseline representation learned by MAE and use CKA to quantify how closely each subset-induced representation aligns with it. In addition to random masking subsets, we also consider frequency-based subsets obtained by ranking tokens with our high-frequency scores and assigning the top 25% to *high-freq. only* and the remaining 75% to *low-freq. only* (see Appendix A.2 for details). High CKA means that a subset leaves this baseline largely unchanged, whereas low CKA indicates that information specific to that subset is not well reflected in the baseline representation.

As summarized in Table 1, three observations emerge. (1) *25% masked* (which retains 75% of tokens) maintains AUROC comparable to *full* and exhibits near-reference CKA, revealing substantial redundancy in the MAE representations. (2) *low-freq. only* shows very high CKA yet a clear AUROC drop, indicating that low-frequency background structure dominates the baseline representation while contributing limited diagnostic signal. (3) *high-freq. only*, which keeps only 25% of tokens, yields low CKA yet achieves the best AUROC, consistently outperforming *75% masked* at the same token budget (25% visible)—showing that a small set of high-

Table 1: **CKA and linear probing across token subsets.** AUROC is the macro-average across five benchmarks; per-dataset results are provided in Appendix A.9.

| Subset | CKA | AUROC |
|---|---|---|
| full | Baseline | 0.685 |
| 25% masked | 0.996 | 0.686 |
| low-freq. only | 0.990 | 0.641 |
| 75% masked | 0.890 | 0.647 |
| high-freq. only | 0.164 | 0.727 |

frequency tokens carries most of the diagnostic signal but is under-emphasized in the baseline representation. Taken together, standard MAE redundantly encodes low-frequency backgrounds and under-encodes high-frequency diagnostic structure, motivating a representation-level correction.

## 5 FREQUENCY-BALANCED RETINAL MASKED AUTOENCODERS

We introduce a *frequency-balanced retinal masked autoencoder (RetMAE)* that mitigates the frequency imbalance identified in Sec. 4 through an MI formulation. As illustrated in Fig. 2, RetMAE augments the standard MAE reconstruction loss with a novel MI regularizer—the *high-frequency MI maximization objective (HighFreqMI)*—to steer the encoder toward compact and task-sufficient representations. We ground the approach in the MI principle (Sec. 5.1) and then detail the objective and training procedure (Sec. 5.2).

### 5.1 MUTUAL INFORMATION AS A PRINCIPLE

We optimize MAE under an MI perspective to embed diagnostically relevant retinal cues. In this framework (Eq. 2), the conditional term $I(X_V; X_M \mid Z)$ is instantiated by the standard MAE objective. To regulate the marginal term $I(X_V; Z)$, we align the trainable representation $Z$ with a high-frequency–focused context latent used as a reference. Optimizing the conditional and marginal terms jointly yields a frequency-balanced encoder suitable for retinal diagnosis.

**Reconstruction as conditional mutual information minimization.** In the decomposition of Eq. 2, the term $I(X_V; X_M \mid Z)$ corresponds to the MAE reconstruction objective. This relationship becomes clear when the decoder is modeled as an isotropic Gaussian with fixed variance, following probabilistic autoencoder formulations (Bishop & Nasrabadi, 2006; Kingma et al., 2013; 2014; Ciampiconi et al., 2023). Under this assumption, the mean squared error (MSE) is proportional to the negative log-likelihood, a standard and analytically convenient interpretation that links reconstruction to conditional mutual information, as shown in the theorem below.

**Theorem 1.** *Let $Z = f_\theta(X_V)$ be the encoder output and let the decoder $f_\phi$ map $Z$ to the input space. Assume an isotropic Gaussian reconstruction model with fixed variance,*

$$p_\phi(X \mid Z) = \mathcal{N}\big(\hat{X}, \ \sigma^2 I\big), \qquad \hat{X} = f_\phi(Z) \in \mathbb{R}^{N \times (P^2 C)},$$

*where $\sigma^2 > 0$ is constant. Then minimizing the MAE reconstruction loss,*

$$\min_{\theta, \phi} \ \mathcal{L}_{\text{rec}},$$

is equivalent, up to a positive affine rescaling determined by $\sigma^2$, to minimizing the conditional mutual information between visible and masked patches,

$$\min_{\theta,\phi} \ I(X_V; X_M \mid Z).$$

See Appendix A.3 for the proof. The standard reconstruction loss therefore serves as a principled surrogate for minimizing $I(X_V; X_M \mid Z)$.

**Context alignment as marginal mutual information minimization.** Within Eq. 2, the marginal term $I(X_V; Z)$ can be bounded by aligning the trainable representation to a compact and task-informative context. If the context encoder discards irrelevant variation while preserving diagnostic cues, then aligning the trainable encoder to this context drives $Z$ toward a similarly compact encoding of the visible input.

**Theorem 2.** *Let $Z_c = g(X)$ be a context representation that is $\varepsilon$-compact, meaning $I(X; Z_c) \leq \varepsilon$, and let $Z = f_\theta(X_V)$ be the trainable representation produced from visible patches. Suppose training achieves mutual-information alignment between $Z$ and $Z_c$ up to a small error $\delta_{\mathrm{align}}$ and capacity matching up to a small mismatch $\delta_{\mathrm{cap}}$. Define $\delta := \max\{\delta_{\mathrm{align}}, \delta_{\mathrm{cap}}\} \geq 0$. Then*

$$I(X_V; Z) \ \leq \ I(X_V; Z_c) + \delta \ \leq \ \varepsilon + \delta.$$

Appendix A.4 provides the proof. In practice, standard MAE training already produces a reasonably compact trainable representation, and the capacity gap between the trainable and context encoders is typically modest. Maximizing $I(Z; Z_c)$ then aligns $Z$ to the $\varepsilon$-compact context and tightens control of the marginal term $I(X_V; Z)$ in Eq. equation 2. When alignment error and capacity mismatch are negligible, the bound approaches $I(X_V; Z) \leq \varepsilon$.

Taken together, Theorems 1 and 2 yield a clear protocol. The MAE loss reduces the conditional term $I(X_V; X_M \mid Z)$, while alignment to an $\varepsilon$-compact context upper-bounds the marginal term $I(X_V; Z)$. Together, these mechanisms yield compact yet diagnostically sufficient representations and jointly optimize the MI Lagrangian in Eq. equation 2.

## 5.2 TRAINING OBJECTIVE

Guided by Theorem 2, we control the marginal $I(X_V; Z)$ by *maximizing* $I(Z_c; Z)$ between the trainable latent $Z = f_\theta(X_V)$ and a compact context latent $Z_c$. Since mutual information is generally intractable to compute exactly, we instead maximize a Donsker–Varadhan-based lower bound on $I(Z_c; Z)$ using the Mutual Information Neural Estimator (MINE) (Donsker & Varadhan, 1983; Belghazi et al., 2018). With a critic $f_\psi : \mathbb{R}^D \times \mathbb{R}^D \to \mathbb{R}$ scoring joint pairs $(Z_c^i, Z^i) \sim p(Z_c, Z)$ and shuffled (product-marginal) pairs $(Z_c^i, Z^j)_{j\neq i} \sim p(Z_c) \otimes p(Z)$, the objective is

$$\mathcal{L}_{\mathrm{MINE}}(Z_c, Z) = -\ \mathbb{E}_{p(Z_c, Z)}[f_\psi(Z_c, Z)] \ + \ \log \mathbb{E}_{p(Z_c) \otimes p(Z)}\big[\exp\{f_\psi(Z_c, Z')\}\big], \qquad (3)$$

where $Z' \sim p(Z)$ is independent. Our implementation of MINE is based on an open-source reference implementation.[1]

**High-frequency MI regularization.** We construct a high-frequency context latent $Z_c^{\mathrm{HF}}$ by feeding frequency-selected visible tokens (Appendix A.2) into an exponential moving-average (EMA) teacher of the encoder. The HighFreqMI objective maximizes the mutual information between the trainable representation $Z$ and this context, estimated with MINE:

$$\mathcal{L}_{\mathrm{hmi}} = \mathcal{L}_{\mathrm{MINE}}\big(Z, \ Z_c^{\mathrm{HF}}\big). \qquad (4)$$

Our base RetMAE augments MAE reconstruction with HighFreqMI,

$$\mathcal{L}_{\mathrm{total}} = \lambda_{\mathrm{rec}} \, \mathcal{L}_{\mathrm{rec}} + \lambda_{\mathrm{hmi}} \, \mathcal{L}_{\mathrm{hmi}}. \qquad (5)$$

By Theorem 2, the context must be compact; accordingly, we activate HighFreqMI only after a short warm-up period so that the EMA teacher stabilizes. The loss $\mathcal{L}_{\mathrm{hmi}}$ acts as a high-frequency MI regularizer on the shared latent $Z$, mitigating MAE's low-frequency bias and encouraging frequency-balanced representations. A detailed analysis of the computational overhead introduced by our high-frequency regularization is provided in Table 13 in Appendix A.9.

---

[1] `https://github.com/Linear95/CLUB`

**Auxiliary-loss–augmented RetMAE.** Recent work improves MAE encoders by adding auxiliary objectives beyond reconstruction (e.g., text supervision or alignment to features from a pretrained vision model) (Fang et al., 2023; 2024b; Yu et al., 2024). Following this paradigm, we consider an auxiliary-loss–augmented variant that adds a generic term $\mathcal{L}_{\text{aux}}$ on top of MAE+HighFreqMI:

$$\mathcal{L}_{\text{total}} = \lambda_{\text{rec}}\,\mathcal{L}_{\text{rec}} + \lambda_{\text{hmi}}\,\mathcal{L}_{\text{hmi}} + \lambda_{\text{aux}}\,\mathcal{L}_{\text{aux}}. \tag{6}$$

In our experiments, $\mathcal{L}_{\text{aux}}$ is instantiated as MINE-based feature alignment between $Z$ and frozen features $Z_c^{\text{aux}}$ extracted from a pretrained fundus encoder (e.g., RET-CLIP):

$$\mathcal{L}_{\text{aux}} = \mathcal{L}_{\text{MINE}}(Z,\, Z_c^{\text{aux}})\,. \tag{7}$$

Here, $\lambda_{\text{rec}}, \lambda_{\text{hmi}}, \lambda_{\text{aux}} \geq 0$ are non-negative scalar weights; in all experiments, we fix $\lambda_{\text{rec}} = 1$ and use $\lambda_{\text{hmi}} = 0.1$ and $\lambda_{\text{aux}} = 0.01$. A sensitivity analysis of these loss weights and the contribution of each component is provided in Appendix A.9. The results indicate that incorporating $\mathcal{L}_{\text{hmi}}$ provides the largest improvements, both alone and in combination with the auxiliary term.

## 6 Experiments

We evaluate RetMAE on five retinal fundus benchmarks and probe the mechanisms behind its gains, testing the hypothesis that it learns frequency-balanced, task-sufficient representations. Sec. 6.1 specifies baselines, datasets, and evaluation protocols. Sec. 6.2 reports downstream performance, loss ablations, and pretraining data efficiency. Sec. 6.3 examines frequency balance via (1) layer-wise CKA under frequency-masked inputs, (2) PCA visualizations of class-to-patch attention, and (3) the linear decodability of high-frequency targets from frozen patch embeddings.

### 6.1 Experimental Setup

**Models.** We evaluate a broad set of fundus pretraining approaches, including a vision–language baseline (RET-CLIP (Du et al., 2024)) and MAE-based methods (RETFound (Zhou et al., 2023), UrFound (Yu et al., 2024)). Complete model configurations and training protocols are provided in Appendix A.6.

**Datasets.** We evaluate RetMAE on four public fundus benchmarks: IDRiD, RFMiD, CHAKSU, and APTOS, spanning three diagnostic categories—diabetic retinopathy (DR), age-related macular degeneration (AMD), and glaucoma (GL). RFMiD is split into DR and AMD subsets, which are evaluated independently. APTOS is used solely as an *out-of-distribution test set* to avoid data leakage. A detailed description of tasks, labels, image counts, and splits is provided in Appendix A.7.

**Evaluation protocols.** To evaluate the quality of the learned representations, we adopt a linear probing strategy in which the encoder is frozen and only a linear head is trained. Performance is measured using the AUROC.

### 6.2 Main Results

**Comparison with signal-level baselines.** We compare latent high-frequency (HF) regularization via $\mathcal{L}_{\text{hmi}}$ against two signal-level variants built on top of a standard MAE encoder and evaluated under the same linear probing protocol. HF masking augments MAE by ranking patches based on their high-frequency energy and allocating visible tokens accordingly, echoing salience- and attention-guided masking approaches (Choi et al., 2024; Sick et al., 2025). HF input concatenation modifies MAE by injecting an additional high-pass channel into the encoder, following prior frequency-guided designs (Xie et al., 2022; Wang et al., 2024b). Both approaches

Table 2: **Average (AVG) AUROC for frequency-aware baselines.**

| Method | AVG |
|---|---|
| MAE | 0.685 |
| + $\mathcal{L}_{\text{hmi}}$ | 0.750 |
| MAE w/ HF masking | 0.679 |
| + $\mathcal{L}_{\text{hmi}}$ | 0.737 |
| MAE w/ HF input | 0.746 |
| + $\mathcal{L}_{\text{hmi}}$ | **0.769** |

expose HF cues directly at the input but do so through signal manipulation rather than latent-level guidance. As shown in Table 2, adding $\mathcal{L}_{\text{hmi}}$ consistently improves all MAE-based variants, indicating that latent-level HF regularization captures information that input-level methods alone do not recover. Although MAE with HF input and $\mathcal{L}_{\text{hmi}}$ attains the highest score, HF input requires architectural changes and additional preprocessing, whereas $\mathcal{L}_{\text{hmi}}$ is architecture-agnostic and lightweight.

Table 3: **Linear probing performance (AUROC).** Columns marked $^\dagger$ are out-of-distribution test sets. AVG is the macro-average across datasets. Values in light gray denote evaluation datasets seen during pretraining. *Auxiliary loss:* ✓ indicates the use of auxiliary signals beyond images (e.g., text guidance or a retina-informed off-the-shelf encoder); ✗ indicates image-only self-supervised pretraining.

| Method | Auxiliary loss | IDRiD | RFMiD (DR) | RFMiD (AMD) | CHAKSU | APTOS$^\dagger$ | AVG |
|---|---|---|---|---|---|---|---|
| MAE | ✗ | 0.726 | 0.721 | 0.793 | 0.371 | 0.812 | 0.685 |
| RETFound | ✗ | 0.736 | 0.760 | 0.784 | 0.464 | 0.706 | 0.690 |
| RetMAE | ✗ | 0.816 | 0.848 | 0.852 | 0.516 | 0.862 | 0.779 |
| UrFound | ✓ | 0.836 | 0.955 | 0.953 | 0.604 | 0.927 | 0.855 |
| MAE | ✓ | 0.887 | 0.949 | 0.959 | 0.912 | 0.910 | 0.923 |
| RET-CLIP | ✓ | 0.898 | 0.955 | 0.962 | 0.930 | 0.940 | 0.937 |
| RetMAE | ✓ | 0.910 | 0.952 | 0.980 | 0.911 | 0.952 | 0.941 |

Table 4: **Full fine-tuning performance (AUROC).**

| Method | IDRiD | RFMiD (DR) | RFMiD (AMD) | CHAKSU | APTOS$^\dagger$ | AVG |
|---|---|---|---|---|---|---|
| RETFound | 0.856 | 0.926 | 0.942 | 0.755 | 0.902 | 0.876 |
| RET-CLIP | 0.879 | 0.947 | 0.916 | 0.836 | 0.973 | 0.910 |
| RetMAE | 0.856 | 0.956 | 0.963 | 0.903 | 0.961 | 0.928 |

For this reason, and to cleanly isolate the effect of latent-level HF guidance, we use the standard MAE as our primary baseline. Table 2 reports AUROC (linear probing) averaged over the datasets described in Sec. 6.1.

**Linear probing performance.** Table 3 reports linear probing results across five benchmarks. Ret-MAE attains the best macro-average AUROC (0.941). On APTOS, it achieves the top AUROC score (0.952), indicating strong out-of-distribution (OOD) generalization. Across datasets, RetMAE with auxiliary losses surpasses all image-only MAE variants (MAE, RETFound, UrFound), indicating that MI-based emphasis on high-frequency retinal structure improves MAE pretraining. Because knowledge transfer degrades under distribution shift (Zhang et al., 2025), the auxiliary-only baseline tends to underperform RET-CLIP. Nevertheless, adding HighFreqMI improves macro-average AUROC ($\Delta$AUROC $+0.018$) and matches or surpasses RET-CLIP. These results indicate that *explicit high-frequency alignment, rather than language supervision, is the principal driver of the gains.* Appendix A.9 reports additional results for the precision–recall curve (AUPRC) in Table 11, as well as performance on multi-disease diagnosis datasets (Table 7).

**Full fine-tuning performance.** Table 4 reports AUROC under full fine-tuning on five retinal benchmarks. RetMAE attains the best average performance (0.928), exceeding RET-CLIP (0.910) and RETFound (0.876), with particularly strong gains on RFMiD (DR/AMD) and CHAKSU while remaining competitive on IDRiD and APTOS. These results show that our high-frequency regularization improves not only linear-probe performance but also full fine-tuning, the regime most relevant for clinical deployment.

**Pretraining data efficiency.** We assess pretraining data efficiency by subsampling each training fold into nested random subsets at $\{75, 50, 25, 10, 5, 1\}\%$ of the full split ($S_{75\%} \supset S_{50\%} \supset \cdots \supset S_{1\%}$). Figure 3 reports the macro-average AUROC across five datasets as a function of pretraining set size; per-benchmark curves and additional details are provided in Appendix A.9. RetMAE achieves strong downstream performance with substantially fewer images: with only $\sim1\%$ of the pretraining set (2.6k images), it surpasses RETFound trained on 904k images (AUROC 0.741 vs. 0.690); with $5\%$ (12.8k images), it also exceeds UrFound trained on 187k images (AUROC 0.925 vs. 0.855).

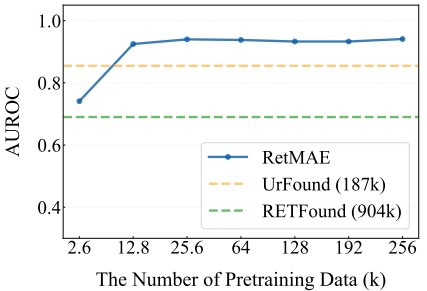

Figure 3: **Pretraining data efficiency.**

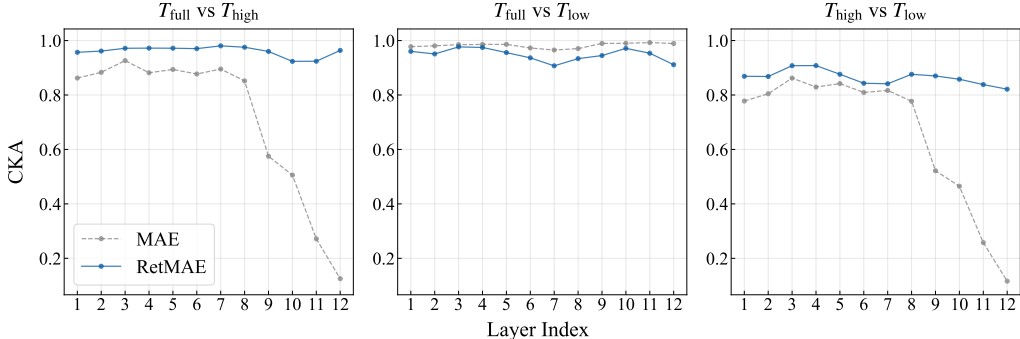

Figure 4: **Frequency-aware CKA analysis across layers.** The three panels compare (left) the full input with high-frequency tokens, (center) the full input with low-frequency tokens, and (right) the high- and low-frequency subsets directly. RetMAE achieves more balanced alignment across frequencies, highlighting its frequency-balanced representation.

Although the curve plateaus beyond approximately 12.8k images, a full discussion of the underlying reasons, including domain redundancy, effective data diversity, and scaling considerations, is provided in Appendix A.8.

## 6.3 Frequency-Aware Representation Analysis

This section evaluates whether RetMAE learns frequency-balanced representations using a frequency-centric analysis of its embeddings. We employ three probes: (1) CKA to assess how high-frequency (HF) and low-frequency (LF) content is encoded and how these subsets align with the full representation; (2) principal component analysis (PCA) of class-to-patch attention to qualitatively visualize the organization of frequency-specific retinal structures; and (3) HF decodability to quantify the linear predictability of HF content from patch tokens. Together, these probes show that, in retinal imaging, MAE-style pretraining further biases ViT representations toward low-frequency backgrounds due to its uniform-information assumption, while our MI-based regularizer restores high-frequency sensitivity without sacrificing low-frequency structure, providing mechanistic insight into how MAE can be corrected in this domain.

**CKA similarity.** Building on Sec. 4, we compute CKA between representations obtained under different frequency-visibility conditions: the full-input tokens ($T_{\text{full}}$), the high-frequency-only tokens ($T_{\text{high}}$), and the low-frequency-only tokens ($T_{\text{low}}$). Concretely, we report layer-wise CKA for three pairs: $T_{\text{full}}$ vs. $T_{\text{high}}$, $T_{\text{full}}$ vs. $T_{\text{low}}$, and $T_{\text{high}}$ vs. $T_{\text{low}}$. Results for MAE and RetMAE are shown in Fig. 4. (1) $T_{\text{full}}$ vs. $T_{\text{high}}$: MAE exhibits high similarity in early layers that declines with depth, indicating progressive attenuation of HF content in the learned embedding; in contrast, RetMAE sustains higher similarity through depth, consistent with preserving task-relevant HF cues under compression. (2) $T_{\text{full}}$ vs. $T_{\text{low}}$: Both models maintain consistently high similarity ($\approx 0.9$–$1.0$) across layers. Taken together with (1), this shows that RetMAE preserves HF alignment without sacrificing LF structure, whereas MAE becomes increasingly LF-biased with depth. (3) $T_{\text{high}}$ vs. $T_{\text{low}}$: The models are similar in early layers, but with depth MAE similarity approaches zero (strong separation of frequency components), while RetMAE remains moderate-to-high, indicating a more balanced co-embedding of HF/LF components rather than collapsing. Overall, MAE tends toward LF-dominated representations, whereas RetMAE maintains frequency diversity across layers and keeps HF information coupled to the global embedding.

**PCA visualization of class-to-patch attention.** We visualize how the global representation attends to local structure by applying PCA to the concatenated class-to-patch attention maps across heads and mapping the top three principal components to RGB (following Oquab et al. (2023)). This yields a compact chromatic embedding in which tokens with similar class semantics appear in similar colors. Models that preserve richer HF detail exhibit sharper chromatic contrast aligned with retinal anatomy. As illustrated in Fig. 5, RetMAE shows clearer separation of clinically salient regions than baseline models (e.g., lesion boundaries are cleanly delineated from smooth background, and the optic disc is consistently isolated). These visualizations indicate that RetMAE organizes tokens into frequency-aware, anatomically coherent clusters, preserving high-frequency detail

| Input | High-Freq. | **RetMAE (Ours)** | RET-CLIP | RETFound | UrFound |

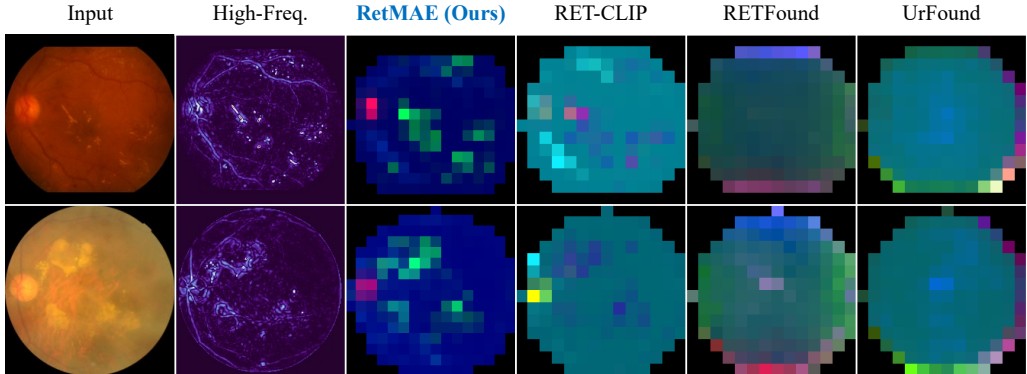

Figure 5: **PCA visualization of class-to-patch attention.** RetMAE shows pronounced chromatic separation—optic disc (red), hemorrhages/exudates (green), and background (blue)—with clear segregation aligned with clinical priors, a property important for retinal diagnosis. Additional examples, visualization details, and attention maps for large lesion areas are provided in Appendix A.9.

while maintaining low-frequency structure. Across the three analyses, we find that RetMAE learns frequency-balanced representations: it aligns well with high-frequency information while preserving low-frequency structure. The learned representations maintain high- and low-frequency components distinct rather than collapsing and exhibit clearer anatomical organization. This frequency balance, in turn, contributes to improved diagnostic performance observed on downstream tasks (Sec. 6.2).

**High-frequency decodability of patch tokens.** To assess whether HF information remains decodable from learned features, we predict the patch-level HF targets defined in Eq. 22 from patch tokens using ridge regression, and compute the coefficient of determination $R^2$ layer-wise (averaged over images). Figure 6 summarizes results for early, middle, and late layers, while per-image $R^2$ distributions are provided in Appendix A.9. RetMAE yields near-ceiling $R^2$ across all depths (early 0.975, middle 0.994, late 0.991; all $p < 10^{-15}$), demonstrating that HF content is robustly linearly decodable from patch tokens. In contrast,

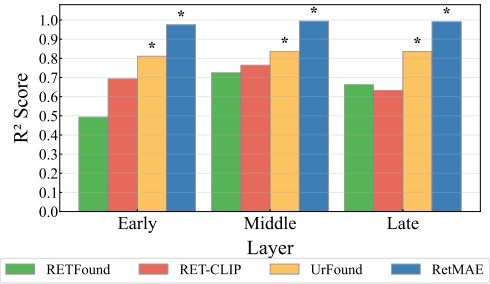

Figure 6: **Layer-wise high-frequency decodability ($R^2$).** Asterisks denote $p < 0.001$.

other models show markedly lower $R^2$, often peaking at middle layers but declining at later ones (e.g., RET-CLIP 0.764 at middle; RETFound 0.724 at middle), reflecting reduced HF sensitivity at the representation level. When averaged across layers, RetMAE achieves the highest mean HF decodability ($\overline{R^2} = 0.987$), consistent with the frequency-aware structure revealed by analysis in Fig. 4.

## 7 DISCUSSION

This work provides, to our knowledge, the first latent-level mutual-information formulation that explicitly diagnoses and mitigates the low-frequency bias of MAE in medical imaging. We introduce RetMAE, which augments the reconstruction objective with a high-frequency MI regularizer to reduce low-frequency redundancy and emphasize clinically salient structure, without modifying the architecture or requiring paired supervision. We further find that signal-level frequency-aware approaches, including high-frequency input augmentation and frequency-aware masking, provide complementary benefits when combined with our latent-level HighFreqMI regularizer, supporting the generality of the proposed framework. A key limitation of this study is that our evaluation is restricted to color fundus images. While the same mechanism may extend to domains in which task-relevant signals are partially concentrated in high-frequency structure, such as industrial anomaly detection or other medical imaging modalities, this generalization remains to be validated. Future work includes extending RetMAE to additional imaging modalities, exploring alternative backbone architectures, adopting adaptive or data-driven context targets, and conducting broader distribution-shift evaluations.

## REPRODUCIBILITY STATEMENT

All implementation details are in Appendix A.5; baselines, datasets, splits, and metrics are in Appendix 6.1, with dataset notes in Appendix A.7 and frequency preprocessing in Appendix A.2. We will release the codebase with model implementation. All runs used fixed random seeds. Experiments were run on $8\times$ NVIDIA RTX 3090 (24 GB) using PyTorch Lightning 2.4.0; training used `torch.compile` mode (no mixed precision). Training and evaluation were conducted on the same machine; multi-GPU runs used DDP (NCCL, fixed global batch size). The proposed method (Ret-MAE) and the MAE baseline instantiate backbones via `timm` (version $\geq 1.0.12$); all other baselines were obtained from their official GitHub repositories. To facilitate reproducibility, our complete codebase is publicly available at https://github.com/vuno/RetMAE.

## ETHICS STATEMENT

This work uses only de-identified, publicly available retinal fundus datasets (see Appendix A.7); no additional patient data were collected. Models and code are released for research use only and are not intended for clinical decision-making without further validation and regulatory approval. We acknowledge potential dataset biases and report results across multiple benchmarks to support transparent evaluation.

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

# A APPENDIX

## A.1 FUNDUS IMAGES CONTAIN LESS HIGH-FREQUENCY CONTENT

As illustrated in Figure 1, natural images typically exhibit a broad frequency spectrum in which edges, textures, and object boundaries contribute substantial high-frequency (HF) energy. By contrast, fundus photographs are markedly low-frequency–dominated: most pixels belong to smooth background regions, while clinically meaningful structures—hemorrhages, drusen, hard exudates, the optic disc, and vessels—are sparse and concentrated in HF bands. We substantiate this observation with the percentile-threshold analysis described below.

**Quantitative Method: 75th-Percentile Threshold**    Let $\mathrm{HF}(x)$ denote the per-pixel HF magnitude map of image $x$ (see Sec. A.2). We define a dataset-level threshold from ImageNet-1K (Deng et al., 2009) as

$$T = \mathrm{quantile}_{0.75}\big(\mathrm{HF}(\text{ImageNet-1K})\big).$$

For any image $x$ with $N$ pixels, we then compute the fraction of pixels exceeding this reference threshold:

$$R(x;T) = \frac{1}{N}\big|\{\, i \,:\, \mathrm{HF}(x)_i > T \,\}\big|.$$

Thus, $R(x;T)$ measures how many pixels in $x$ are "HF-active" relative to the ImageNet-derived reference distribution.

**Quantitative Results: Histogram Analysis**    Figure 1 depicts the empirical distribution of $R(x;T)$ for fundus images and ImageNet-1K. The fundus distribution concentrates near zero, whereas ImageNet exhibits a broader, right-shifted distribution, indicating a substantially larger fraction of HF-active pixels. A Mann–Whitney U test (Mann & Whitney, 1947) indicates a highly significant difference ($p = 4.75 \times 10^{-12}$), confirming that fundus photographs contain markedly less HF content. These results quantitatively corroborate the qualitative patterns in Figure 1: clinically relevant structures in fundus images are sparse and localized within HF regions, while the overall image is dominated by low-frequency background.

## A.2 HIGH-FREQUENCY EXTRACTION

To isolate diagnostically relevant high-frequency signals, we first extract and normalize the green channel, then generate a soft field-of-view mask to attenuate edge artifacts. We apply this mask before Gaussian blurring so that only the central retinal region contributes to the frequency analysis. After suppressing low-frequency background with the blur, we transform the result to the Fourier domain and filter it using a Butterworth high-pass filter whose cutoff and order were tuned on a held-out set of fundus images. The dataset used for this adjustment, which included annotations of the ground truth vessel and the lesion, was strictly excluded from both pretraining and evaluation. Inverting the filtered spectrum back to the spatial domain produces a high-pass filtered (HPF) image, which we re-mask and normalize to eliminate any residual boundary effects.

**Soft-FOV Mask Generation**   To smoothly attenuate high-frequency artifacts at the field-of-view (FoV) boundaries and enhance lesion-related high-frequency signals in retinal fundus images, we compute a soft field-of-view (Soft-FOV) mask. Given an input image $\mathbf{I} \in \mathbb{R}^{C \times H \times W}$, we first form a grayscale image by channel averaging:

$$Y_{h,w} = \frac{1}{C} \sum_{c=1}^{C} I_{c,h,w}. \tag{8}$$

We define a threshold

$$T_s = \tau_{\text{fov}} \max_{h,w} Y_{h,w}, \tag{9}$$

and generate a binary mask

$$B_{h,w} = \begin{cases} 1, & Y_{h,w} > T_s, \\ 0, & \text{otherwise.} \end{cases} \tag{10}$$

To soften the edges, we convolve $B$ with a separable 2D Gaussian kernel

$$\mathcal{K}_{\sigma_s}(x,y) = k_{\sigma_s}(x)\, k_{\sigma_s}(y), \quad k_{\sigma_s}(t) = \frac{1}{\sqrt{2\pi}\,\sigma_s} \exp\big(-t^2/(2\sigma_s^2)\big),$$

applied horizontally and then vertically:

$$\widetilde{B} = (B *_h k_{\sigma_s}) *_v k_{\sigma_s}. \tag{11}$$

Finally, we clamp $\widetilde{B}$ to $[0,1]$ to obtain the Soft-FOV mask $S \in [0,1]^{H \times W}$:

$$S_{h,w} = \min\big(\max(\widetilde{B}_{h,w}, 0), 1\big). \tag{12}$$

**High-pass Filtering**   After obtaining $S$, we extract high-frequency components from the green channel of the input, denoted $I^{(g)} \in \mathbb{R}^{H \times W}$, which maximizes vessel/lesion contrast (Biswas et al., 2022; Ooi et al., 2021; Kumar et al., 2020). We perform min–max normalization (with $\varepsilon > 0$ for stability):

$$\widetilde{I}^{(g)}_{h,w} = \frac{I^{(g)}_{h,w} - \min_{h',w'} I^{(g)}_{h',w'}}{\max_{h',w'} I^{(g)}_{h',w'} - \min_{h',w'} I^{(g)}_{h',w'} + \varepsilon} \in [0,1]. \tag{13}$$

We then apply the Soft-FOV mask and Gaussian blur (std. $\sigma_h$, kernel radius $r_h$):

$$I_{\text{blur}} = \mathcal{G}_{\sigma_h, r_h}\big(\widetilde{I}^{(g)} \odot S\big) \in \mathbb{R}^{H \times W}. \tag{14}$$

The 2D discrete Fourier transform (DFT) is

$$F(u,v) = \sum_{h=0}^{H-1} \sum_{w=0}^{W-1} I_{\text{blur}}(h,w) \, \exp\big[-i2\pi\big(\tfrac{u\,h}{H} + \tfrac{v\,w}{W}\big)\big]. \tag{15}$$

Let the radial distance from the spectrum center be

$$D(u,v) = \sqrt{\Big(u - \tfrac{H}{2}\Big)^2 + \Big(v - \tfrac{W}{2}\Big)^2}. \tag{16}$$

A Butterworth high-pass filter of order $n$ and cutoff $D_0$ is

$$\mathcal{H}_{\mathrm{BW}}(u, v) \;=\; \frac{1}{1 + \left(\frac{D_0}{D(u,v)}\right)^{2n}}.$$ (17)

Applying the filter in the frequency domain and inverting gives the raw high-pass response:

$$F_{\mathrm{HP}}(u, v) \;=\; \mathcal{H}_{\mathrm{BW}}(u, v)\, F(u, v),$$ (18)

$$H^{\mathrm{hp}}(h, w) \;=\; \big|\, \mathcal{F}^{-1}\{F_{\mathrm{HP}}\}(h, w) \,\big| \;\in \mathbb{R}^{H \times W}.$$ (19)

To remove background and boundary responses, we binarize the Soft-FOV:

$$S_{\mathrm{th}}(h, w) \;=\; \mathbf{1}\big[S(h, w) > \beta\big],$$ (20)

and obtain the final high-frequency map without additional normalization:

$$H^{\mathrm{hf}} \;=\; H^{\mathrm{hp}} \odot S_{\mathrm{th}}, \qquad H^{\mathrm{hf}}_{h,w} \;=\; \begin{cases} H^{\mathrm{hp}}_{h,w}, & S(h, w) > \beta, \\ 0, & \text{otherwise.} \end{cases}$$ (21)

**High-frequency Token Masking** Our Vision Transformer (ViT) backbone partitions the input into $P \times P$ patches, each mapped to a token (Dosovitskiy et al., 2020). To identify tokens enriched with high-frequency content, we construct a token mask from $H^{\mathrm{hf}} \in \mathbb{R}^{H \times W}$.

First, average $H^{\mathrm{hf}}$ within each non-overlapping patch:

$$A_{u,v} \;=\; \frac{1}{P^2} \sum_{i=0}^{P-1} \sum_{j=0}^{P-1} H^{\mathrm{hf}}_{uP+i,\, vP+j}, \quad u = 0, \ldots, \tfrac{H}{P} - 1,\ v = 0, \ldots, \tfrac{W}{P} - 1.$$ (22)

Then, rank $\{A_{u,v}\}$ and mark the top $r_{\mathrm{hf}}\%$ of patches to form $M \in \{0, 1\}^{\frac{H}{P} \times \frac{W}{P}}$:

$$M_{u,v} \;=\; \begin{cases} 1, & A_{u,v} \text{ is in the top } r_{\mathrm{hf}}\% \text{ of } \{A_{u,v}\}, \\ 0, & \text{otherwise.} \end{cases}$$ (23)

The resulting high-frequency token mask $M$ delineates token positions carrying abundant high-frequency information.

Table 5: Optimized hyperparameters for the Soft-FOV mask and high-frequency (HF) extraction. Values were selected by maximizing the Dice score between the HF maps and available lesion/vessel ground-truth (GT) masks on held-out development data.

| Notation | Description | Value |
|---|---|---|
| $D_0$ | Butterworth cutoff frequency | 14.0470 |
| $n$ | Butterworth filter order | 2 |
| $\tau_{\mathrm{fov}}$ | Soft-FOV threshold | 0.0869 |
| $\sigma_s$ | Soft-FOV Gaussian sigma | 10.1332 |
| $\beta$ | Boundary cutoff | 0.6701 |
| $\sigma_h$ | Gaussian blur sigma | 0.5185 |
| $r_h$ | Gaussian blur radius | 1 |
| $r_{\mathrm{hf}}$ | High-frequency token masking ratio (fraction) | 0.25 |

**Hyperparameter Search** We optimized the Soft-FOV parameters $\{\tau_{\mathrm{fov}}, \sigma_s\}$, the HF filtering parameters $\{\sigma_h, r_h, D_0, n, \beta\}$, and the HF token masking ratio $r_{\mathrm{hf}}$ using the tree-structured Parzen estimator (TPE) (Watanabe, 2023). The search comprised 10,000 iterations on IDRiD (Porwal et al., 2020) and FIVES (Jin et al., 2022) images that were excluded from both pretraining and downstream evaluation. IDRiD provides ground truth (GT) masks for hemorrhages, hard exudates, cotton wool patches, and microaneurysms, while FIVES provides GT vessel masks. For each candidate configuration, we generated HF maps and scored them against the corresponding GT masks using the Dice coefficient; the configuration achieving the highest mean Dice was selected. Table 5 reports the resulting hyperparameters.

**Qualitative Observations on Filter Generalization**

Although tuned on fundus data, the filter yields plausible HF representations on ImageNet images. As illustrated in Figure 1, edges and fine textures in natural images are preserved in the extracted HF maps, indicating that the filter captures domain-agnostic HF cues. This behavior supports the use of a single parameterization for cross-domain comparisons and suggests favorable generalization beyond the fundus domain.

### A.3 PROOF OF THEOREM 1

**Setup** Let $X = [\mathbf{x}_i]_{i=1}^N$ denote the sequence of patch tokens with visible/masked split $X_V$ and $X_M$, and let $Z$ be the encoder representation. Assume a reconstruction model with isotropic Gaussian likelihood

$$p_\phi(X \mid Z) = \mathcal{N}(\hat{X}, \sigma^2 I), \quad \hat{X} = f_\phi(Z),$$

and conditional factorization over masked patches given $Z$.

**Step 1: Upper bounding the conditional mutual information by a conditional entropy** By definition,

$$I(X_V; X_M \mid Z) = H(X_M \mid Z) - H(X_M \mid X_V, Z).$$

Since conditional entropy is nonnegative, $H(X_M \mid X_V, Z) \geq 0$, it follows that

$$I(X_V; X_M \mid Z) \leq H(X_M \mid Z).$$

**Step 2: Evaluation of $H(X_M \mid Z)$ under the Gaussian decoder** Using the assumed likelihood and the conditional factorization over $i \in \mathcal{M}$,

$$H(X_M \mid Z) = -\mathbb{E}_{p_\phi(X_M \mid Z)}[\log p_\phi(X_M \mid Z)] = \sum_{i \in \mathcal{M}} \mathbb{E}_{p_\phi(x_i \mid Z)}[-\log p_\phi(x_i \mid Z)]$$

$$= \frac{MD}{2} \log(2\pi\sigma^2) + \frac{1}{2\sigma^2} \sum_{i \in \mathcal{M}} \|x_i - \hat{x}_i\|^2,$$

where $M = |\mathcal{M}|$ and $D = P^2 C$ denote, respectively, the number of masked patches and their dimensionality.

**Step 3: Identification of an affine relation to the MAE reconstruction loss** Define the mean-squared reconstruction loss over masked patches

$$\mathcal{L}_{\text{rec}} = \frac{1}{M} \sum_{i \in \mathcal{M}} \|x_i - \hat{x}_i\|^2.$$

Then

$$H(X_M \mid Z) = \alpha \mathcal{L}_{\text{rec}} + \text{const}, \qquad \alpha = \frac{M}{2\sigma^2} > 0.$$

**Conclusion** Combining the previous steps yields the affine upper bound

$$I(X_V; X_M \mid Z) \leq \alpha \mathcal{L}_{\text{rec}} + \text{const}.$$

Since $\alpha > 0$ is fixed for given $\sigma^2$, any minimizer of $\mathcal{L}_{\text{rec}}$ with respect to $(\theta, \phi)$ is a minimizer of the right-hand side, and hence

$$\min_{\theta,\phi} \mathcal{L}_{\text{rec}} \implies \min_{\theta,\phi} I(X_V; X_M \mid Z).$$

**Remarks** (1) The result relies on the isotropic Gaussian likelihood with fixed variance; more generally, any fixed-variance quadratic negative log-likelihood induces the same monotone relation. (2) If $\sigma^2$ is learned, an explicit control (e.g., regularization or variance constraints) is required to keep the affine coefficient $\alpha$ well-defined and to prevent trivial solutions.

## A.4 PROOF OF THEOREM 2

**Setup** Let $Z_c = g(X)$ be the context representation with $I(X; Z_c) \leq \varepsilon$, and let $Z_s = f_\theta(X_V)$ be the student representation. Assume that training enforces (1) mutual-information alignment between $Z_s$ and $Z_c$ and (2) capacity matching so that the entropy of $Z_s$ does not exceed that of $Z_c$ by more than a small margin.

**Step 1: Propagation of the teacher bound to the visible part** By the chain rule,

$$I(X; Z_c) = I(X_V; Z_c) + I(X_M; Z_c \mid X_V),$$

whence

$$I(X_V; Z_c) \leq I(X; Z_c) \leq \varepsilon.$$

**Step 2: Control of the student–context gap by alignment** The identity

$$I(X_V; Z_s) - I(X_V; Z_c) = I(X_V; Z_c \mid Z_s) - I(X_V; Z_s \mid Z_c)$$

together with nonnegativity of mutual information yields

$$\left| I(X_V; Z_s) - I(X_V; Z_c) \right| \leq \max\{ I(X_V; Z_c \mid Z_s),\ I(X_V; Z_s \mid Z_c) \}.$$

Using $I(A; B \mid C) \leq H(B \mid C)$,

$$\left| I(X_V; Z_s) - I(X_V; Z_c) \right| \leq \max\{ H(Z_c \mid Z_s),\ H(Z_s \mid Z_c) \} =:\ \delta_{\text{align}}.$$

**Step 3: Capacity matching** If, in addition, the entropies satisfy $|H(Z_s) - H(Z_c)| \leq \delta_{\text{cap}}$, the asymmetry between the two conditional bounds above is uniformly controlled. Set $\delta := \max\{\delta_{\text{align}}, \delta_{\text{cap}}\}$.

**Conclusion** Combining the previous steps,

$$I(X_V; Z_s) \leq I(X_V; Z_c) + \delta \leq \varepsilon + \delta.$$

Under perfect alignment and exact capacity matching, i.e., $\delta \to 0$, it follows that $I(X_V; Z_s) \leq \varepsilon$.

**Remarks** Alignment refers to the requirement that $Z_s$ and $Z_c$ be mutually predictable, equivalently that both $H(Z_s \mid Z_c)$ and $H(Z_c \mid Z_s)$ be small. Capacity matching refers to the requirement that the entropy of $Z_s$ be close to that of $Z_c$, which prevents $Z_s$ from encoding additional information beyond what is present in $Z_c$.

## A.5 IMPLEMENTATION DETAILS

**Data preparation** All models were pretrained on 256,044 fundus images collected from 39 publicly available datasets (see Appendix A.7 for details). All images are intensity-normalized to $[0, 1]$, and rectangular inputs are zero-padded to preserve the aspect ratio. Images were resized to $224 \times 224$, and standardized using the ImageNet mean and standard deviation. Pre-training was performed for 100 epochs, and the last checkpoint was used for downstream tasks. Data augmentation was applied with Kornia (Riba et al., 2020), including random rotation ($\pm 10°$), random resized cropping with a scale range of $[0.2, 1.0]$, random horizontal flipping, and color jittering with brightness, contrast, and saturation factors of $0.3$.

**Architecture** The encoder was a ViT backbone with four register tokens, initialized from Dinov2 weights (Oquab et al., 2023), and used a patch size of $14$ to match the Dinov2 configuration. The decoder comprised eight Transformer layers. For latent features, ViT-based encoders used the `[CLS]` token for both $Z$ and $Z_c$. For HighFreqMI, the trainable encoder's latent representation $Z$ was linearly projected into the latent space of the context features. Models were trained with input resolutions of $224 \times 224$.

**Optimization and training schedule** We used AdamW (Loshchilov & Hutter, 2017) with $\beta_1 = 0.9$, $\beta_2 = 0.95$, and a batch size of 768. The learning rate was set to $3 \times 10^{-4}$ for the encoder/decoder and the HighFreqMI projection heads, following a cosine decay schedule after a 10-epoch warm-up. The encoder/decoder weight decay was cosine-scheduled. During masked autoencoder (MAE) pretraining, the masking ratio was fixed at $80\%$. For the HighFreqMI objective, the model was trained with the MAE objective alone for the first 40 epochs, after which high-frequency alignment with the exponential moving-average (EMA) context encoder was enabled. The EMA momentum was cosine-scheduled from $0.994$ to $1.0$.

A.6 MODELS

Table 6 summarizes three pretraining paradigms considered in this work: (1) a CLIP-style image–text model (RET-CLIP), (2) masked image modeling (MIM) baselines (RETFound, UrFound), and (3) our RetMAE. RET-CLIP optimizes contrastive alignment between retinal images and associated text, whereas the MIM baselines pretrain ViT backbones by reconstructing masked patches. Ret-MAE retains the MAE backbone and masking scheme but augments the reconstruction objective with complementary mutual-information (MI) regularizers that suppress low-frequency redundancy and emphasize clinically salient high-frequency content, yielding *frequency-balanced* retinal representations. All models use comparable ViT architectures and input resolutions; training follows each method's standard protocol without manual labels or architectural modifications.

Table 6: **Summary of pretraining strategies.** Comparison of model architecture, parameter count, input resolution, and use of text and model supervision across methods.

| Method | Arch. | Params. | Res. | Text Sup. | Model Sup. |
|---|---|---|---|---|---|
| *Contrastive Language–Image Pre-Training* | | | | | |
| RET-CLIP (Du et al., 2024) | ViT-B/16 | 86M | 224 | ✓ | ✗ |
| *Masked Image Modeling* | | | | | |
| UrFound (Yu et al., 2024) | ViT-B/16 | 86M | 224 | ✓ | ✗ |
| RETFound (Zhou et al., 2023) | ViT-L/16 | 305M | 224 | ✗ | ✗ |
| *Ours* | | | | | |
| RetMAE | ViT-B/14 | 86M | 224 | ✗ | ✓ |

## A.7 DATASETS

**Pretraining Datasets** Table 7 enumerates the 39 publicly available fundus datasets used for Ret-MAE pretraining together with their training-image counts. Spanning a broad range of clinical conditions (from diabetic retinopathy to glaucoma), these sources collectively contribute 256,097 images. To avoid leakage across sources, we performed image-level deduplication and retained only unique samples; consequently, our totals may differ slightly from those reported in the original releases. This large, diverse corpus enables learning robust retinal representations without manual labels.

Table 7: **Number of pretraining images per dataset.** 39 public fundus datasets were used.

| Dataset | # images | Dataset | # images |
|---|---|---|---|
| 1000fundus (Cen et al., 2021) | 996 | AGAR300 (Derwin et al., 2020) | 26 |
| ARIA (Farnell et al., 2008) | 143 | AVRDB (Akram et al., 2020) | 99 |
| Benitez (Benítez et al., 2021) | 1,406 | BRSET (Nakayama et al., 2023) | 16,265 |
| Cataract (Kaggle, 2020) | 601 | DeepDRiD (Liu et al., 2022) | 2,000 |
| DiaRetDB1 (Kauppi et al., 2007) | 117 | DiaRetDB2 (Guo et al., 2024) | 28 |
| DR1-2 (Pires et al., 2014) | 1,567 | drimdb (Chakraborty, 2024) | 194 |
| DRD (Kaggle, 2015) | 88,702 | DRIONS-DB (Carmona et al., 2008) | 110 |
| FGADR (Zhou et al., 2021) | 1,828 | FIRE (Hernandez-Matas et al., 2017) | 124 |
| FUND-OCT (Hassan et al., 2019) | 163 | G1020 (Bajwa et al., 2020) | 1,020 |
| HEI-MED (Giancardo et al., 2012) | 169 | HRF (Budai et al., 2013) | 79 |
| IOSTAR (Zhang et al., 2016) | 30 | JICHI (Takahashi et al., 2017) | 9,939 |
| JustRAIGS (Madadi et al., 2025) | 101,423 | LAG (Li et al., 2019a) | 4,854 |
| LES (Orlando et al., 2020b) | 22 | MSHF (Jin et al., 2023) | 500 |
| ODIR (NIHDS-PKU, 2019) | 6,996 | OIA-DDR (Li et al., 2019b) | 9,504 |
| ORIGA (Zhang et al., 2010) | 650 | PALM (Fang et al., 2024a) | 1,174 |
| PAPILA (Kovalyk et al., 2022) | 488 | RC-RGB-MA (Dashtbozorg et al., 2018) | 242 |
| REFUGE (Orlando et al., 2020a) | 1,200 | RetinalLesion (Wei et al., 2020) | 1,593 |
| RIDB (Abdul Salam et al., 2020) | 100 | RIGA (Almazroa et al., 2018) | 270 |
| ROC (Niemeijer et al., 2009) | 100 | SUSTech-SYSU (Lin et al., 2020) | 1,218 |
| TREND (Popovic et al., 2021) | 104 | | |
| Total | | | 256,044 |

**Evaluation Datasets** Table 8 summarizes the downstream classification benchmarks, all disjoint from the pretraining corpus. The in-domain datasets (e.g., IDRiD, RFMiD, CHAKSU) are used to evaluate diabetic retinopathy (DR), age-related macular degeneration (AMD), and glaucoma (GL) detection, while APTOS is reserved exclusively for out-of-distribution (OOD) testing. For each dataset, labels are mapped to task-specific binaries (e.g., referable vs. non-referable DR) following the dataset's official taxonomy.

Table 8: **Evaluation datasets.** These sets were not used for pretraining.

| Dataset | Lesions | # Train | # Val | # Test |
|---|---|---|---|---|
| **Classification** | | | | |
| IDRiD (Porwal et al., 2020) | DR | 408 | – | 102 |
| RFMiD (Pachade et al., 2021) | DR, AMD | 2,560 | – | 640 |
| CHAKSU (Kumar et al., 2023) | GL | 1,009 | – | 336 |
| APTOS (Maggie & Dane, 2019) | DR | – | – | 3,394 |

**Data Splits** For each in-domain benchmark, we adopt the official train/test split. Within the training portion, 20% of the data is held out for validation and the remaining 80% is used for training; model selection and early stopping rely solely on this validation set. We report test performance on the official test split and assess generalization on the OOD set (APTOS). All methods use identical folds and preprocessing to ensure a fair comparison.

## A.8 Data scaling, redundancy, and fundus statistics

Figure 3 plots downstream macro-AUROC on five supervised retinal benchmarks as a function of pretraining set size for a fixed encoder (ViT-Base/16) and a fixed training budget. In this setting, RetMAE already surpasses much larger retinal foundation models with substantially fewer pretraining images: with only ∼1% of the corpus (≈2.6k images), it exceeds RETFound trained on 904k images; with 5% (≈12.8k images), it also outperforms UrFound trained on 187k images. The apparent plateau beyond ∼12.8k images might therefore seem surprising when viewed through the lens of natural-image foundation models.

**Fundus as a low-entropy, highly redundant Domain**  As illustrated in Figure 1, the vast majority of pixels in a fundus photograph belong to a relatively homogeneous background dominated by low-frequency content (retinal surface, illumination, and overall color tone), while clinically informative structures occupy only a small fraction of the field of view. Our high-frequency extraction suppresses this background and concentrates the signal into sparse, local high-frequency patterns such as microaneurysms, hemorrhages, and exudates. From an information-theoretic perspective, this implies that the *intrinsic entropy* of the domain is relatively low and the dataset is highly redundant: many images share very similar low-frequency backgrounds, and the diagnostic information is concentrated in a comparatively small set of high-frequency deviations. RetMAE is explicitly designed to focus on these high-frequency signals through HighFreqMI. Once the pretraining corpus is large enough to cover the diverse lesion and vessel patterns present in the population, additional images tend to be incrementally redundant under a fixed backbone and training schedule. In such a regime, it is natural for downstream performance curves to approach a ceiling with substantially fewer images than in unconstrained natural-image corpora.

**Connection to subset selection, pruning, and scaling laws**  Our observations align with prior work showing that, in redundant datasets, carefully selected subsets can match or even outperform full-data training. Coreset and subset-selection methods such as CRAIG demonstrate that models trained on representative subsets achieve performance comparable to full-data training while using significantly fewer examples and updates (Mirzasoleiman et al., 2020). In large language models, data-pruning studies report that retaining only 30–50% of the pretraining corpus (ranked by simple quality metrics such as perplexity) can preserve or improve downstream performance compared to using all data (Marion et al., 2023). In computer vision, pruning strategies that prioritize images with higher intrinsic perceptual complexity (e.g., bits-per-pixel–based entropy scores) have been shown to match full-dataset performance on classification and segmentation tasks (Singh, 2024). In medical imaging, deep active learning studies—including work on retinal fundus photographs—find that actively selected subsets can reach or exceed the performance of models trained on all available images while greatly reducing labeling and training cost (Wang et al., 2024a; Paul et al., 2022). Taken together, these results support the view that, in a low-entropy, highly structured domain like fundus imaging, the *effective* number of distinct, task-relevant patterns is much smaller than the raw image count. Once these patterns are well covered, simply adding more similar images yields diminishing returns under a fixed-capacity encoder and fixed compute. This perspective is also consistent with neural scaling laws, which model performance as a power-law function of model size, data, and compute (Hestness et al., 2019; Kaplan et al., 2020; Hernandez et al., 2021; Hoffmann et al., 2022; Dehghani et al., 2023): continued gains typically require *joint* scaling of model capacity, data diversity, and the number of optimization steps. Our experiments intentionally fix the backbone and training budget to isolate the effect of our MI-based objective on sample efficiency; exploring joint capacity–data–compute scaling for RetMAE on more heterogeneous multi-center retinal and non-retinal datasets is an important direction for future work.

A.9   ADDITIONAL RESULTS

**Ablation on loss components**   All AUROC/AUPRC values in Table 9 are macro-averages across five benchmarks. The reconstruction-only baseline is weakest. Introducing $\mathcal{L}_{\mathrm{hmi}}$ yields large gains (AUROC +0.094, AUPRC +0.128), indicating that recovering high-frequency cues is the primary lever. Crucially, $\mathcal{L}_{\mathrm{hmi}}$ is complementary: when added to the auxiliary setting, it provides further improvements (AUROC +0.018,

Table 9: **Ablation of loss components.**

| $\mathcal{L}_{\mathrm{rec}}$ | $\mathcal{L}_{\mathrm{aux}}$ | $\mathcal{L}_{\mathrm{hmi}}$ | AUROC | AUPRC |
|---|---|---|---|---|
| ✓ | – | – | 0.685 | 0.486 |
| ✓ | – | ✓ | 0.779 | 0.614 |
| ✓ | ✓ | – | 0.923 | 0.799 |
| ✓ | ✓ | ✓ | 0.941 | 0.849 |

AUPRC +0.050). Collectively, these results show that HighFreqMI increases high-frequency sufficiency and effectively improves retinal MAE variants.

**Loss-weight ablation**   To assess the sensitivity of RetMAE to the relative weighting of the reconstruction, HighFreqMI, and auxiliary losses, we conducted a loss-weight ablation in which we varied one coefficient at a time while fixing $\lambda_{\mathrm{rec}} = 1$ and disabling the remaining non-varied term (i.e., setting its coefficient to 0). For each configuration, we pre-trained the model on fundus images and evaluated the frozen encoder via linear probing, reporting the macro-averaged AU-ROC across five benchmarks (IDRiD, RFMiD-DR, RFMiD-AMD, CHAKSU, and APTOS). Table 10 summarizes the results. In both cases, performance is relatively stable across a broad range of weights, with a mild optimum around $\lambda_{\mathrm{hmi}} = 0.1$ for the HighFreqMI term and a similarly flat region for the auxiliary loss. Motivated by these observations, we adopt $\lambda_{\mathrm{rec}} = 1$, $\lambda_{\mathrm{hmi}} = 0.1$, and $\lambda_{\mathrm{aux}} = 0.01$ as our default configuration in the main experiments, which provides a robust trade-off between reconstruction, high-frequency regularization, and auxiliary alignment.

Table 10: **Sensitivity of RetMAE to the HighFreqMI and auxiliary loss weights.**

| $\lambda_{\mathrm{hmi}}$ | $\lambda_{\mathrm{aux}}$ | AUROC |
|---|---|---|
| 0 | 0 | 0.685 |
| 1.0 | 0 | 0.738 |
| 0.1 | 0 | **0.779** |
| 0.01 | 0 | 0.732 |
| 0 | 1.0 | 0.927 |
| 0 | 0.1 | 0.926 |
| 0 | 0.01 | **0.932** |
| 0.1 | 0.01 | 0.941 |

**Additional AUPRC results**   Table 11 reports linear probing performance in terms of AUPRC across the same five benchmarks. RetMAE attains the best macro-average AUPRC (0.849) among all methods with auxiliary losses, and achieves the top AUPRC on APTOS (0.960), further confirming its strong OOD generalization. Compared to image-only MAE variants (MAE, RETFound, UrFound), RetMAE consistently improves AUPRC, indicating that MI-based emphasis on high-frequency retinal structure yields more discriminative features. Moreover, with auxiliary losses, RetMAE slightly surpasses RET-CLIP in macro-average AUPRC, reinforcing our conclusion that *explicit high-frequency alignment, rather than language supervision alone, is the principal driver of the gains observed in both AUROC and AUPRC.*

Table 11: **Linear probing performance (AUPRC).** Columns marked [†] are out-of-distribution test sets. AVG is the macro-average across datasets. Values in light gray denote evaluation datasets seen during pretraining. *Auxiliary loss:* ✓ indicates the use of auxiliary signals beyond images (e.g., text guidance or a retina-informed off-the-shelf encoder); ✗ indicates image-only self-supervised pretraining.

| Method | Auxiliary loss | IDRiD | RFMiD (DR) | RFMiD (AMD) | CHAKSU | APTOS[†] | AVG |
|---|---|---|---|---|---|---|---|
| MAE | ✗ | 0.874 | 0.396 | 0.191 | 0.116 | 0.855 | 0.486 |
| RETFound | ✗ | 0.878 | 0.515 | 0.140 | 0.142 | 0.732 | 0.481 |
| RetMAE | ✗ | 0.916 | 0.679 | 0.381 | 0.194 | 0.899 | 0.614 |
| UrFound | ✓ | 0.919 | 0.870 | 0.601 | 0.243 | 0.936 | 0.714 |
| MAE | ✓ | 0.949 | 0.850 | 0.588 | 0.686 | 0.921 | 0.799 |
| RET-CLIP | ✓ | 0.957 | 0.900 | 0.606 | 0.797 | 0.952 | 0.842 |
| RetMAE | ✓ | 0.959 | 0.857 | 0.759 | 0.711 | 0.960 | 0.849 |

**Multi-disease evaluation** To assess performance in more clinically realistic, multi-disease settings, we additionally evaluate RetMAE on two *multi-disease* benchmarks, FIVES (Jin et al., 2022) and RFMiD2 (Panchal et al., 2023). FIVES comprises four diagnostic categories (AMD, DR, glaucoma, and normal), which already span lesions with distinct spatial and frequency characteristics (e.g., microaneurysms, hemorrhages, and

Figure 7: **Linear probing AUROC on multi-disease fundus benchmarks.**

| Method | FIVES | RFMiD2 | AVG |
|---|---|---|---|
| RETFound | 0.837 | 0.806 | 0.822 |
| RET-CLIP | 0.943 | 0.808 | 0.876 |
| RetMAE | 0.922 | 0.884 | 0.903 |

exudates in DR versus optic-nerve cupping and nerve-fiber-layer defects in glaucoma). RFMiD2 is even more challenging: it provides over 40 expert-defined retinal disease labels, including vascular occlusions, macular edema, neovascularization, optic-nerve anomalies, inflammatory conditions, myopic degeneration, tessellation, pigment-epithelium changes, and others, and individual images often carry multiple labels simultaneously. This multi-label, multi-disease structure more faithfully reflects real clinical scenarios, where overlapping disease signatures are common rather than isolated.

Table 7 summarizes linear probing AUROC on these benchmarks. RetMAE achieves the best macro-average performance (AVG 0.903), improving over the self-supervised RETFound baseline from 0.837 to 0.922 on FIVES and from 0.806 to 0.884 on RFMiD2 (AVG 0.822 → 0.903). On FIVES, RET-CLIP attains the highest AUROC (0.943), with RetMAE achieving a competitive second-best score (0.922); on RFMiD2, however, RetMAE clearly outperforms both RETFound and RET-CLIP (0.884 vs. 0.808). Importantly, RetMAE also surpasses RET-CLIP—which leverages language supervision—in terms of the overall average AUROC (0.903 vs. 0.876). These results indicate that our domain-specific high-frequency regularizer is beneficial not only for comparatively simple binary classification tasks, but also for multi-disease, multi-label settings that better capture the clinical complexity and practical value of real-world fundus imaging.

**Pretraining-efficiency per benchmark**  Figure 8 shows per-dataset AUROC as a function of pretraining size. We construct nested subsets at $\{1, 5, 10, 25, 50, 75, 100\}\%$ of the full pretraining set ($\sim \{2.6, 12.8, 25.6, 64, 128, 192, 256\}$k images), pretrain RetMAE$_{\text{retclip}}$ on each subset, and evaluate with a linear probe on IDRiD, RFMiD (DR/AMD), CHAKSU, and APTOS. RetMAE$_{\text{retclip}}$ is highly data-efficient: with only $1\%$ of data it attains a macro-average AUROC of $0.741$, exceeding RETFound ($0.690$); with $5\%$ it reaches $0.925$, surpassing UrFound ($0.855$). Most gains accrue by $10$–$25\%$ ($0.940$ and $0.938$), with diminishing returns thereafter; the best macro-average is $0.941$ at $100\%$. Scaling behavior is task-dependent: RFMiD–AMD improves steadily with more data ($0.836{\rightarrow}0.980$), RFMiD–DR peaks near $50\%$ ($0.966$), while CHAKSU and the OOD set APTOS largely saturate by $10\%$ ($0.932$ and $0.954$) and vary only slightly beyond that. Per-dataset highlights include IDRiD $0.717 \rightarrow 0.910$ from $1\%$ to $100\%$; RFMiD–DR $0.806 \rightarrow 0.966$ (peak at $50\%$); RFMiD–AMD $0.836 \rightarrow 0.980$; CHAKSU $0.693 \rightarrow 0.932$ by $10\%$; and APTOS $0.655 \rightarrow 0.954$ by $10\%$. These trends align with our frequency-oriented view: once frequency-balanced features are established, additional fundus images primarily add low-frequency background redundancy, yielding modest gains, whereas tasks driven by richer high-frequency structure (e.g., AMD) benefit more from scale.

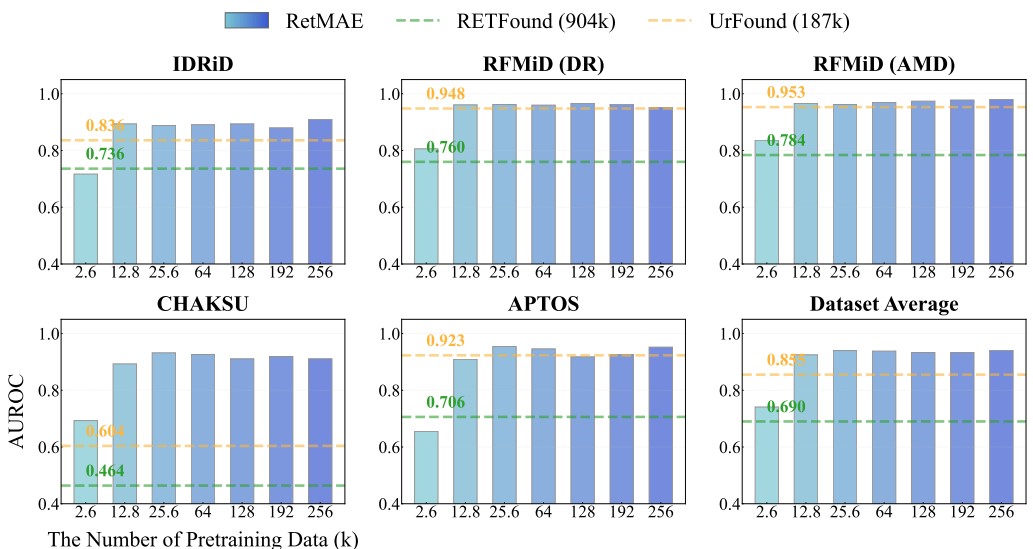

Figure 8: **Pretraining data efficiency of RetMAE.** For all datasets considered, AUROC improves as pretraining size increases. RetMAE achieves the target performance with far fewer images than RETFound and UrFound, and consistently outperforms MAE-based retinal baselines.

**CKA and linear-probe performance across MAE token subsets**  Table 12 reports, for each subset, CKA computed with respect to the *full* input embedding and per-dataset linear-probe AUROC on five benchmarks (IDRiD, RFMiD-DR, RFMiD-AMD, CHAKSU, APTOS).

We observe three regularities. (1) *25% masked* (which retains 75% of tokens) closely matches *full* in AUROC while achieving near-unity CKA (0.996), suggesting substantial redundancy in the MAE representations. (2) *low-freq. only* attains high alignment (CKA = 0.990) yet weak diagnostic signal—yielding the column minima on IDRiD (0.662), RFMiD-DR (0.667), and APTOS (0.718)—and remaining below *full* on RFMiD-AMD (0.778 vs. 0.793). (3) *high-freq. only*, which keeps only 25% of tokens, shows the lowest alignment to *full* (CKA = 0.164) yet the strongest AUROC on IDRiD (0.737), RFMiD-DR (0.792), RFMiD-AMD (0.867), and CHAKSU (0.439), while remaining competitive on the out-of-distribution APTOS set (0.802 vs. 0.812 for *full*).

Overall, the results indicate that MAE representations emphasize low-frequency background structure (high CKA) with limited diagnostic utility, whereas a small subset of high-frequency tokens—despite low alignment to the full-input embedding—captures clinically salient information and yields superior linear-probe performance.

Table 12: **CKA and linear-probing performance across token subsets.** High-frequency tokens yield the strongest diagnostic performance despite low representational alignment with the full input. The best value in each column is shown in **blue** and the lowest in **red**.

| Subset | CKA | IDRiD | RFMiD (DR) | RFMiD (AMD) | CHAKSU | APTOS |
|---|---|---|---|---|---|---|
| full | *Baseline* | 0.726 | 0.721 | 0.793 | **0.371** | **0.812** |
| 25% masked | 0.996 | 0.727 | 0.725 | 0.794 | 0.380 | 0.806 |
| low-freq. only | 0.990 | **0.662** | **0.667** | 0.778 | 0.379 | **0.718** |
| 75% masked | 0.890 | 0.668 | 0.677 | **0.768** | 0.396 | 0.725 |
| high-freq. only | 0.164 | **0.737** | **0.792** | **0.867** | **0.439** | 0.802 |

**High-Frequency decodability of patch tokens and class-to-attention per image** For each pretrained encoder, we regressed the patch-level HF targets (Eq. 22) from frozen patch tokens using ridge regression and computed per-image $R^2$, summarizing both the distribution and a pooled (overall) $R^2$ across images. The per-image distributions are visualized in Fig. 9a (left panel of Fig. 9). RetMAE exhibits uniformly high decodability of HF signal: overall $R^2 = 0.9909$, with mean per-image $0.9896 \pm 0.0032$ and a very tight range $[0.9854, 0.9925]$; the corresponding significance test strongly rejects $H_0 \colon R^2 = 0$ ($p = 1.1 \times 10^{-16}$). UrFound is a distant second (overall $R^2 = 0.8353$; mean $0.8185 \pm 0.0671$; range $[0.7219, 0.8772]$; $p = 6.3 \times 10^{-7}$), showing both a lower central tendency and broader dispersion than RetMAE. RETFound and RET-CLIP yield substantially lower and statistically non-significant overall $R^2$ (0.6620 and 0.6329), consistent with weaker linear decodability of HF content from their patch embeddings. Quantitatively, RetMAE's advantage over the next best model amounts to $\Delta R^2 \approx 0.156$ at the overall level, while its markedly narrower per-image spread indicates that HF information is recoverable *consistently* across images rather than being driven by a subset of easy cases. Taken together, these outcomes corroborate the main-text claim that RetMAE preserves HF information across depth and images, providing linearly decodable access to diagnostically salient structure.

We repeated the analysis using class-to-patch attention features to assess HF decodability from attention-derived representations. The corresponding per-image $R^2$ distributions are shown in Fig. 9b (right panel of Fig. 9). Absolute $R^2$ values are lower—a natural consequence of what is being regressed: class-to-patch attention provides allocation weights rather than feature vectors, is spatially smoothed by softmax and head averaging, and aggregates cues not specific to HF content—yet the ranking remains consistent. RetMAE attains the highest overall $R^2$ (0.3868; mean per-image $0.3243 \pm 0.2296$; range $[0.0728, 0.6244]$; $p = 1.1 \times 10^{-16}$), followed by RET-CLIP (0.2872), UrFound (0.2760), and RETFound (0.1407); all four are significant at $p < 0.05$. Notably, baseline models exhibit broader and occasionally negative per-image $R^2$ values (e.g., minima below zero for RET-CLIP and RETFound), indicating poor linear recoverability of HF targets from their class-attentional structure, whereas RetMAE's distribution is shifted upward with a positive lower bound. These trends align with the frequency-oriented analyses in Sec. 6.3: the PCA visualization of class-to-patch attention (Fig. 5) shows sharper, anatomy-consistent chromatic separation for RetMAE, providing a qualitative counterpart to the elevated HF decodability observed here. Together with the patch-token results above, this supports the view that RetMAE learns *frequency-aware, diagnostically informative* embeddings whose HF components remain linearly accessible.

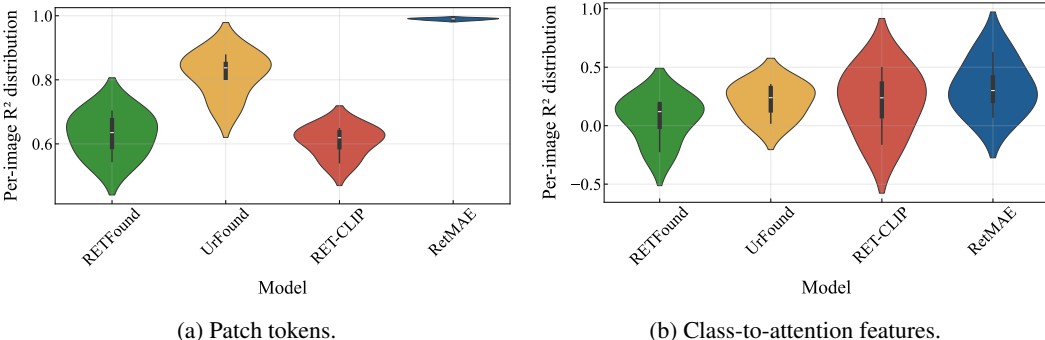

(a) Patch tokens.  (b) Class-to-attention features.

Figure 9: **Per-image $R^2$ violin plots for frequency decodability.** Left: patch-token analysis; right: class-to-attention analysis.

**Computational complexity** We quantify the additional computational cost introduced by our high-frequency regularization. All measurements are conducted on a Tesla V100 GPU using a ViT-Base/16 backbone with input resolution $224 \times 224$, batch size 16, and averaged over 100 iterations. Inference time is measured using CUDA events (`torch.cuda.Event`) for accurate GPU-side timing without CPU–GPU synchronization overhead, and floating point operations per second (FLOPs) are estimated with `fvcore`'s `FlopCountAnalysis` for each component. Our method introduces two additional components on top of the baseline MAE forward pass: (1) high-frequency component extraction and encoding, where high-frequency patches are processed by an EMA encoder, and (2) the HighFreqMI loss, which estimates mutual information between the main encoder latent and the high-frequency context encoder latent via a lightweight critic. Table 13 summarizes the resulting overhead.

As shown in Table 13, the overall overhead is minimal: the proposed regularization adds less than $4\%$ to the baseline inference time (1.88 ms over 52.48 ms) and less than $0.25\%$ to the FLOPs. This efficiency is primarily due to the fact that the high-frequency encoder processes only a subset of patches (high-frequency regions), and the MINE-based mutual information estimator relies on a lightweight critic network with a small number of parameters. Consequently, our high-frequency regularization yields substantial gains in representation quality for fundus imaging tasks while remaining highly practical for real-world deployment.

Table 13: **Computational overhead of the proposed high-frequency regularization on ViT-Base/16 with input resolution** $224^2$**.** Inference time is measured per forward pass on a Tesla V100 GPU. Percentages are reported relative to the baseline MAE.

| Component | Inference time (ms) | FLOPs (GFLOPs) |
|---|---|---|
| Baseline | 52.48 (100.00%) | 69.87 (100.00%) |
| High-frequency component | 1.56 (2.97%) | 0.105 (0.15%) |
| HighFreqMI loss | 0.32 (0.61%) | 0.050 (0.07%) |
| **Total overhead** | **1.88 (3.58%)** | **0.156 (0.22%)** |

**Large lesions and retinal detachment**     We additionally visualize class-to-patch attention for lesions that are not purely high-frequency, including retinal traction detachment and large preretinal hemorrhages (Fig. 10). In the cases of retinal detachment in panels (a)–(b) and a large preretinal hemorrhage in panel (c), the PCA-projected class-token attention remains well aligned with the lesion, with strong responses along the detachment margins and hemorrhage boundaries. This behavior is consistent with our frequency-based view: even when the pathological region covers a broad area, the transition zones at the lesion boundary still correspond to high-frequency structure, and RetMAE continues to emphasize these regions. In the extreme case in panel (d), where pathology occupies almost the entire field-of-view and thus behaves effectively as a low-frequency signal, the class token no longer attends uniformly across the full lesion interior; however, it still concentrates on boundaries and locations where the fundus signal changes abruptly. Clinically, lesion discrimination in fundus photography is largely driven by such sharp intensity and texture changes relative to the surrounding background or neighboring structures, so accurately attending to these boundaries is more important than uniformly covering the entire lesion area.

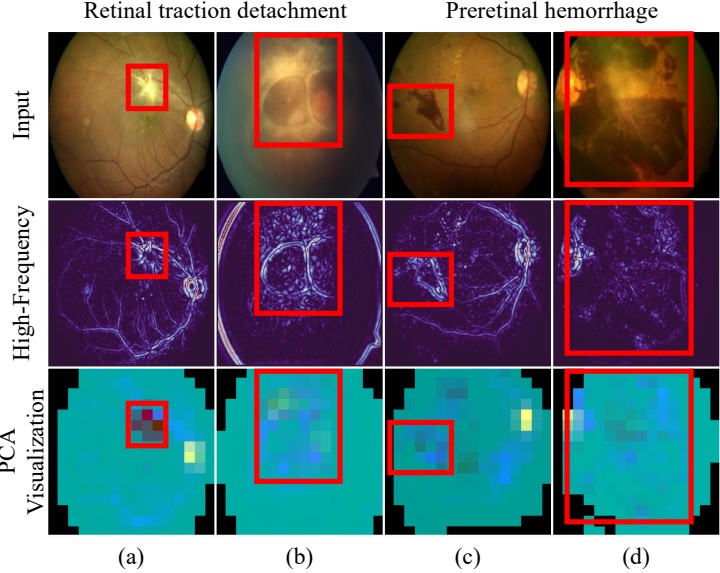

Figure 10: **Attention on large lesions and retinal detachment.** PCA-projected class-token attention maps for (a)–(b) retinal traction detachment, (c) a large preretinal hemorrhage, and (d) an eye with near-global pathology. In (a)–(c), the class-token attention remains well aligned with the lesion, with strong responses along the detachment margins and hemorrhage boundaries. In the extreme case in (d), where pathology occupies almost the entire field of view and thus behaves effectively as a low-frequency signal, the class token does not attend uniformly across the lesion interior, but still concentrates on boundaries and locations where the fundus signal changes abruptly.

**Additional PCA visualizations of class-to-patch attention** We present additional examples of the PCA visualization of class-to-patch attention.

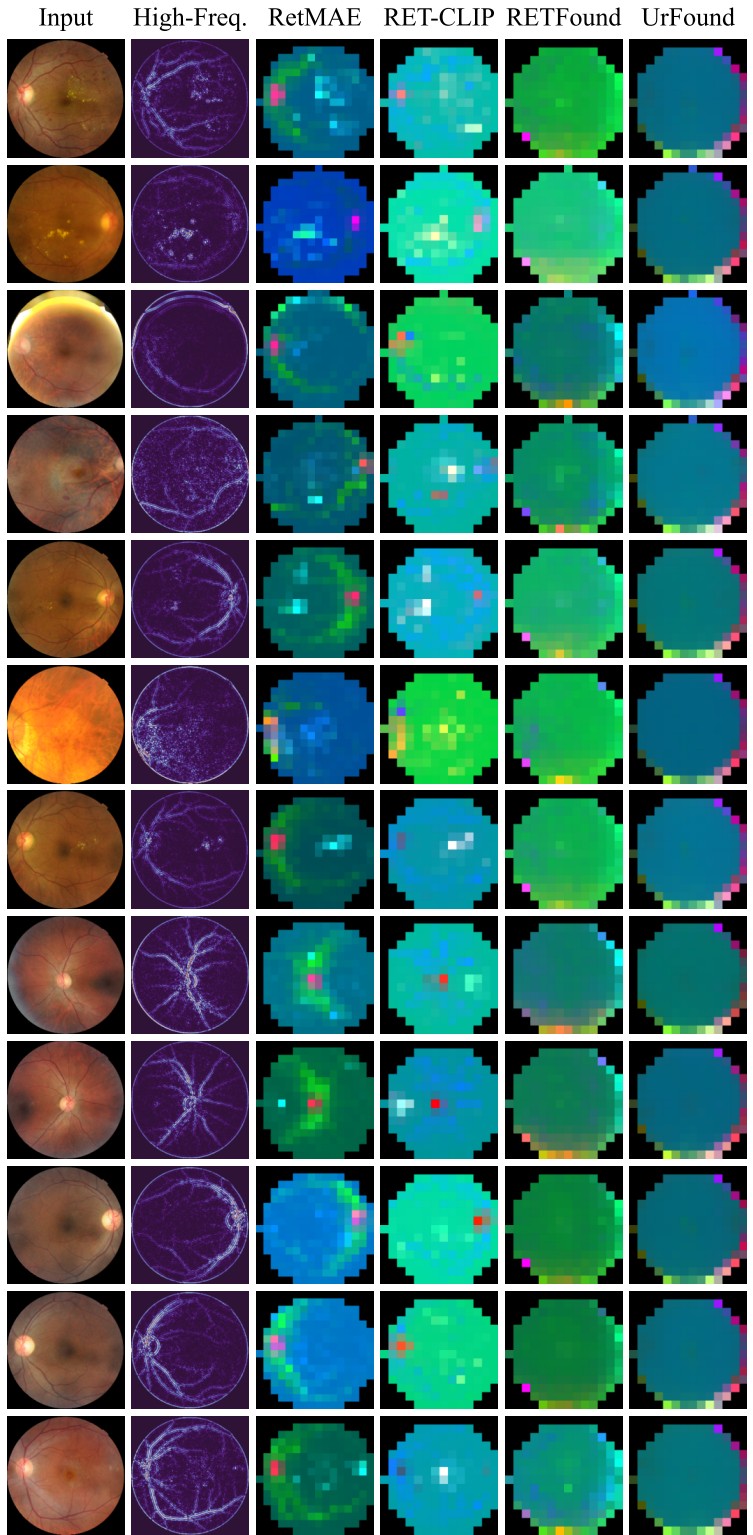

Figure 11: **Example 1 of PCA visualization of class-to-patch attention.**

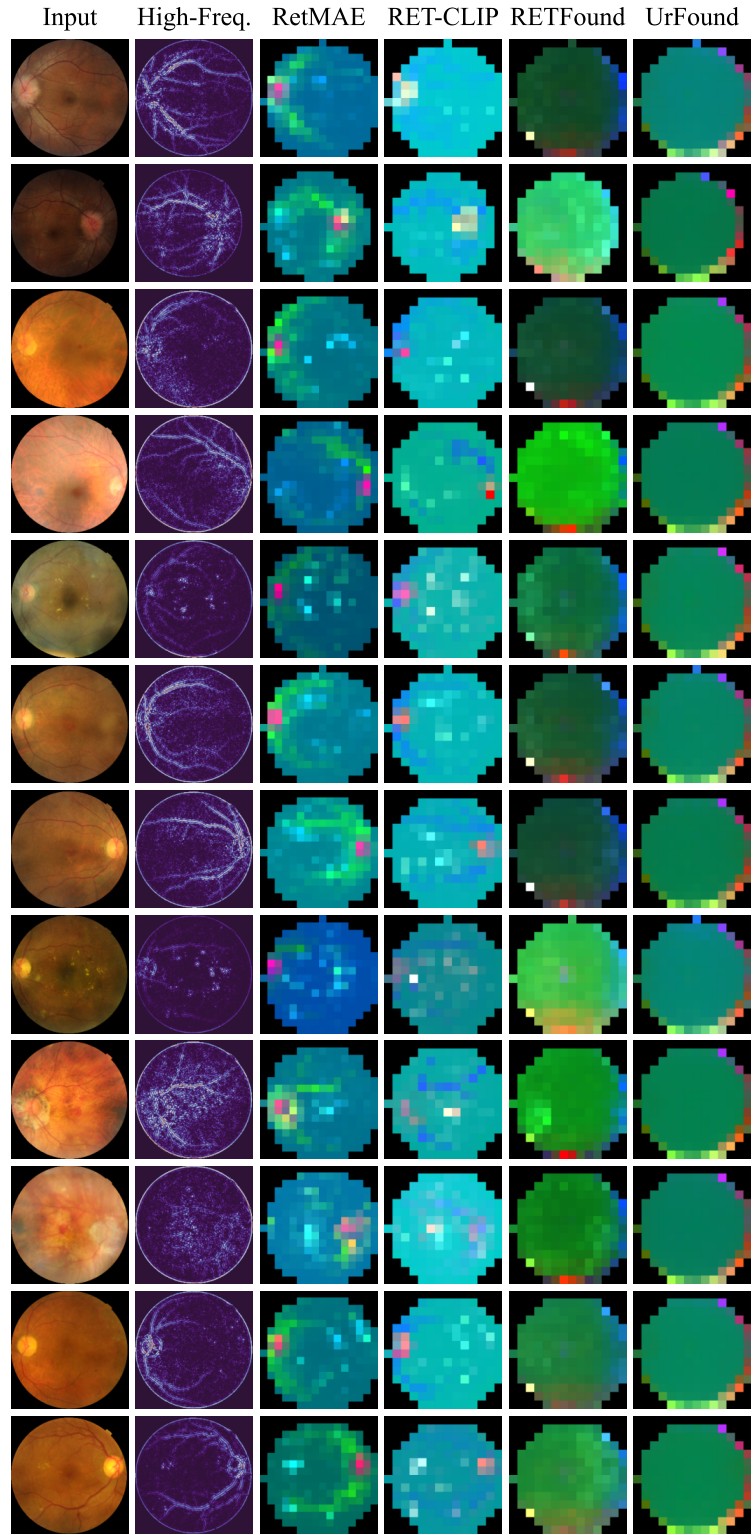

Figure 12: **Example 2 of PCA visualization of class-to-patch attention.**

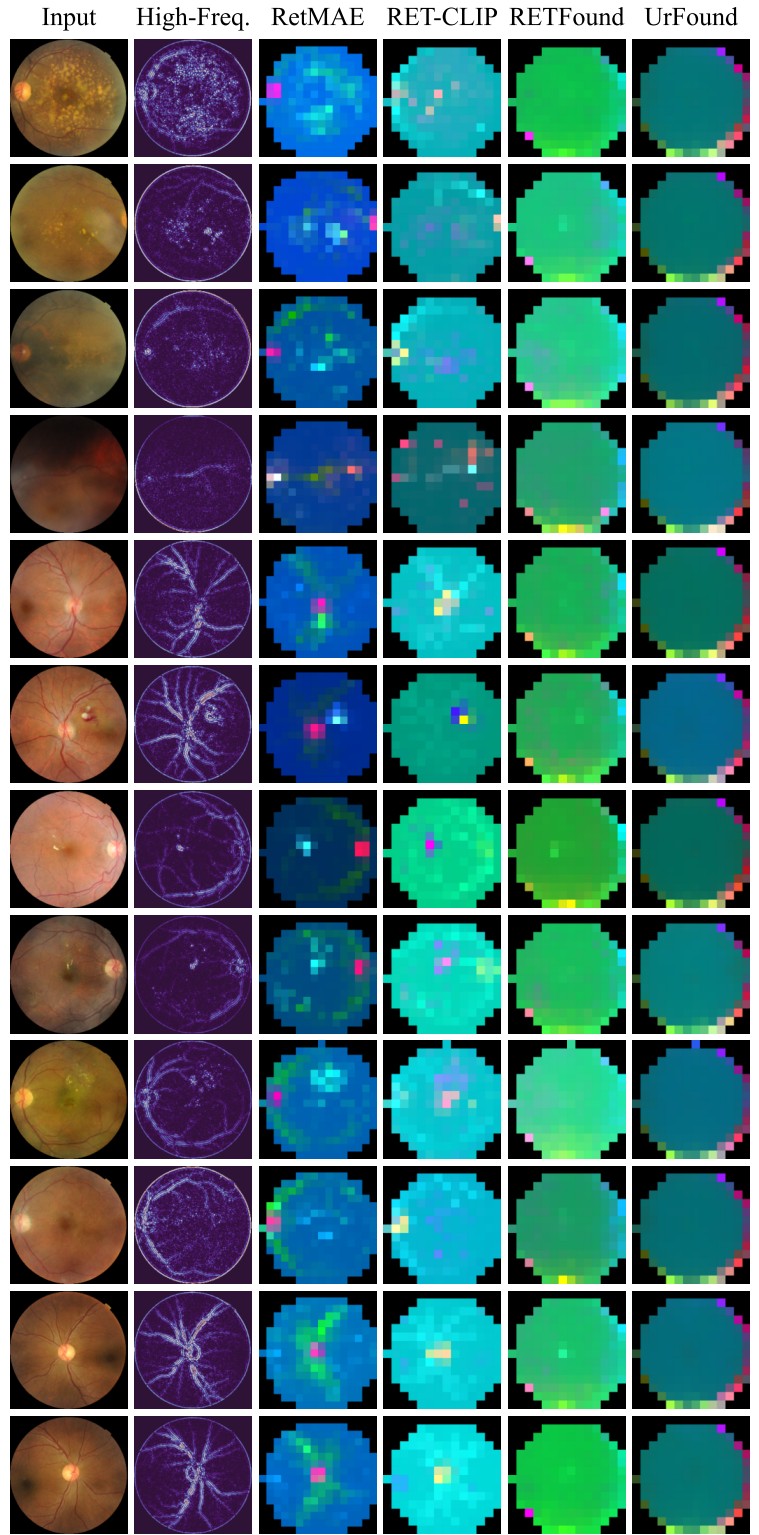

Figure 13: **Example 3 of PCA visualization of class-to-patch attention.**

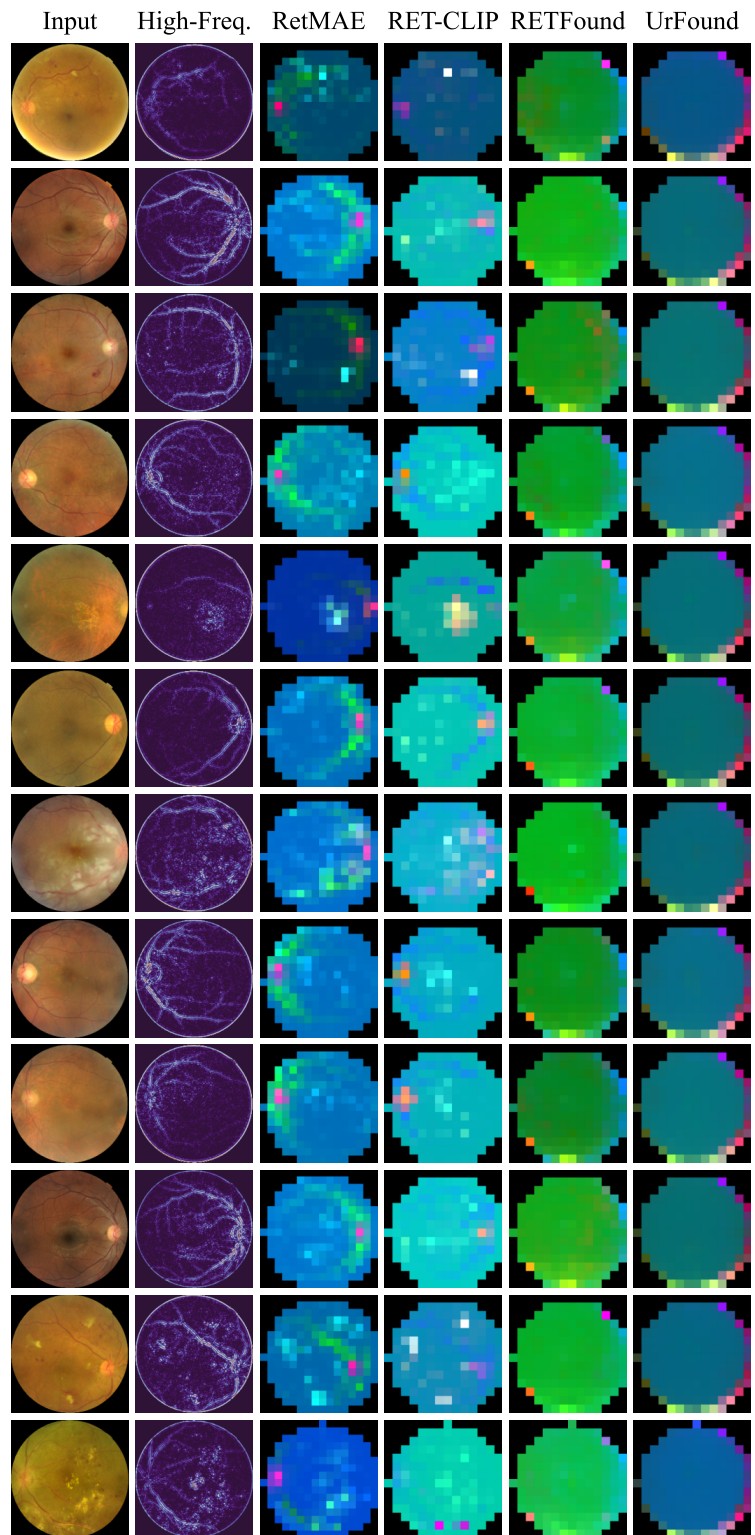

Figure 14: **Example 4 of PCA visualization of class-to-patch attention.**

