# OpenReview forum: "Frequency-Balanced Retinal Representation Learning with Mutual Information Regularization"
_ICLR.cc/2026/Conference — ICLR 2026 Poster_

### Official Review · Reviewer_xA7m · 2025-10-27

**Soundness:** 1
**Presentation:** 3
**Contribution:** 2
**Rating:** 2
**Confidence:** 5

**Summary:**

This paper proposes a frequency-balanced masked autoencoder framework (RetMAE) that enhances retinal representation learning by introducing a high-frequency mutual information regularizer to emphasize clinically critical high-frequency structures while suppressing redundant low-frequency content.

**Strengths:**

This paper does present a clear motivation that high-frequency structures are clinically important in retinal imaging, and it attempts to incorporate this insight through a mutual-information-based regularizer.  And the presentation is good.

**Weaknesses:**

(1) According to Figure 2, the loss L_hmi proposed in this paper and the loss L_rec in the original MAE appear to optimize the latent feature in two fundamentally different directions. Specifically, L_rec aims to optimize 𝑧 such that it can reconstruct the full Image from Image_mask1. In contrast, L_hmi optimizes 𝑧 to enable the reconstruction (or transformation) of Image_mask2 from Image_mask1, where mask2 is generated through the high-frequency masking strategy proposed by the authors. This is clearly contradictory and constitutes the most significant issue.
(2) Were all evaluations in Table 1 conducted on the standard MAE model? The results shown in this table seem to indicate that the standard MAE already has a strong ability to represent high-frequency information, which contradicts the authors’ claim of “under-encoding high-frequency diagnostic structures.” For example, in the T_low row of Table 1, even after masking 25% of high-frequency information, the CKA value remains as high as 0.990, indicating that the model still retains stable reconstruction capability for high-frequency components. Conversely, in the T_high row, when high-frequency information is used as input, the model shows low CKA because it cannot reconstruct the full image, which is expected. However, the AUROC increases, demonstrating that high-frequency information is highly discriminative; when low-frequency, lesion-irrelevant content is removed, the model’s prediction accuracy improves. Therefore, the evidence presented in Table 1 may actually support the importance of high-frequency features rather than demonstrating the insufficiency of MAE in encoding them. I believe the authors have not conducted a sufficiently thorough investigation in this aspect.
(3) As highlighted in Comment (1), there is a potential conflict between the two loss terms. The authors should explicitly report the training weights assigned to each loss or provide a sensitivity analysis (e.g., a performance graph under different loss weight configurations) to demonstrate the impact of the loss balance on model performance.
(4) The innovation of the proposed method appears to be limited, as it essentially adds a frequency-based loss on top of MAE, while the use of high-frequency representations to capture lesion-related features in retinal images has already been explored in numerous prior studies.

If the authors can provide a clear theoretical justification or empirical evidence resolving the contradictions I raised—particularly regarding the compatibility of the two loss objectives and the interpretation of Table 1—I am willing to reconsider my rating and increase my score accordingly.

**Questions:**

Were all evaluations in Table 1 conducted on the standard MAE model?

---

> ### Author Response · Authors · 2025-11-20
> **Response Letter for Reviewer xA7m [1/4]**
>
> We sincerely thank the reviewer for their invaluable feedback. The comments helped us identify several ambiguities and significantly improved the clarity and rigor of our manuscript. Below, we provide detailed responses and describe the corresponding changes in the revised paper.
>
> ---
>
> ## [Q1] Apparent contradiction between $L_{\text{rec}}$ and $L_{\text{hmi}}$
>
> > **Comment:**
> > According to Figure 2, the loss $L_{\text{hmi}}$ proposed in this paper and the loss $L_{\text{rec}}$ in the original MAE appear to optimize the latent feature in two fundamentally different directions. Specifically, $L_{\text{rec}}$ aims to optimize $z$ such that it can reconstruct the full image from Image_mask1.
>
> We sincerely appreciate the reviewer's careful examination, which has prompted us to clarify the interaction between $L_{\text{rec}}$ and $L_{\text{hmi}}$.
>
> **(1) Conceptual relationship between $L_{\text{rec}}$ and $L_{\text{hmi}}$.**
> In our MI–Lagrangian view, $L_{\text{rec}}$ optimizes the conditional term $I(X_V; X_M \mid Z)$: it encourages $Z$ to be sufficient to reconstruct masked patches. By contrast, $L_{\text{hmi}}$ does **not** reconstruct a second image or "transform Image_mask2 from Image_mask1". Instead, it maximizes $I(Z, Z_c^{\text{HF}})$ between $Z$ and a high-frequency context latent $Z_c^{\text{HF}}$ extracted from frequency-selected patches of the *same image* via a teacher/EMA encoder.
>
> Both losses act on the **same** latent $Z$:
> - $L_{\text{rec}}$: makes $Z$ informative enough to reconstruct masked content (conditional term).
> - $L_{\text{hmi}}$: regularizes the marginal term $I(X_V; Z)$ so that $Z$ emphasizes diagnostically important high-frequency structure.
>
> The two losses optimize different terms of a single MI objective, rather than pulling $Z$ in incompatible directions.
>
> **(2) Why this is corrective, not contradictory.**
> Empirically, vanilla MAE is strongly **low-frequency biased** (Table 1, Fig. 4): MAE latents align well with low-frequency components but poorly with high-frequency components (e.g., lesions), despite the latter carrying most diagnostic signal. $L_{\text{hmi}}$ **corrects** this imbalance by aligning $Z$ with high-frequency context, producing frequency-balanced latents that preserve low-frequency structure while improving high-frequency encoding (Fig. 4).
>
> **(3) Empirical evidence that $L_{\text{hmi}}$ is complementary.**
> If $L_{\text{hmi}}$ conflicted with $L_{\text{rec}}$, adding it should *hurt* performance. Instead, Tables 2 and 4 show consistent **improvements** in AUROC/AUPRC when $L_{\text{hmi}}$ is added, indicating that high-frequency MI regularization complements reconstruction rather than contradicting it.
>
> **(4) Relation to prior MAE variants.**
> Our design aligns with prior work augmenting MAE with auxiliary objectives (iBOT [1], EVA [2], LuT [3] for natural images; UrFound [4] for fundus images). These methods optimize MAE latents toward additional signals yet enhance representation quality. HighFreqMI introduces a domain-specific high-frequency bias *compatible* with reconstruction: **MAE still needs to reconstruct both low- and high-frequency content, and the HF-aware regularizer counteracts the observed low-frequency dominance.** Ensuring $Z$ encodes sufficient high-frequency information via $L_{\text{hmi}}$ directly helps MAE fulfill its objective in a domain where high-frequency structures are crucial.
>
> **Clarifications added in the revised manuscript.**
> To improve clarity, we have revised Figure 2 and Section 5.2 as follows:
>
> 1. **Figure 2 (Overview of RetMAE)** now explicitly depicts HighFreqMI as *maximizing mutual information with a high-frequency context latent*, clearly separating the MAE reconstruction path from the HF context-alignment path.
> 2. In **Section 5.2 (Training objective: High-frequency MI regularization)**, we now state more clearly that
>   > [L317-L319] \
>   > The loss $L_{\text{hmi}}$ acts as a high-frequency MI regularizer on the shared latent $Z$, mitigating MAE's low-frequency bias and encouraging frequency-balanced representations.
>
> **Reference**
> [1] Zhou, Jinghao, et al. "ibot: Image bert pre-training with online tokenizer." arXiv preprint arXiv:2111.07832 (2021).
> [2] Fang, Yuxin, et al. "Eva: Exploring the limits of masked visual representation learning at scale." Proceedings of the IEEE/CVF conference on computer vision and pattern recognition. 2023.
> [3] Kim, Taekyung, et al. "Learning with unmasked tokens drives stronger vision learners." European Conference on Computer Vision. Cham: Springer Nature Switzerland, 2024.
> [4] Yu, Kai, et al. "Urfound: Towards universal retinal foundation models via knowledge-guided masked modeling." International Conference on Medical Image Computing and Computer-Assisted Intervention. Cham: Springer Nature Switzerland, 2024.

---

> > ### Comment · Reviewer_xA7m · 2025-11-20
> >
> > The authors have provided a clear and convincing explanation for my concerns in points (1) and (3), especially regarding the MI view of 𝐿rec and 𝐿hmi and the role of loss balancing. In my opinion, the manuscript now merits a higher score (from 2 to 4). However, I still have some unresolved questions related to point (2):
> >
> > (1) First, in the original version of Table 1, could the authors explicitly confirm the following:
> >
> > (I) Does the row T_low correspond to feeding the MAE with images where 25% of the high-frequency information has been masked out, so that the MAE effectively receives mainly low-frequency information?
> >
> > (II) Does the row T_high correspond to feeding the MAE with images where 75% of the low-frequency information has been masked out, so that the MAE effectively receives mainly high-frequency information?
> >
> > (III) Does a higher CKA value in Table 1 indeed indicate better reconstruction ability / higher similarity to the full-input representation?
> >
> > (IV) Are the statistics in Table 1 computed over the whole dataset (or at least over multiple images), rather than from a single example? For the conclusions to be convincing, I would expect these values to be aggregated over many images.
> >
> > Moreover, if your answers to I–III above are all “yes”, then the high CKA value in the T_low row seems to indicate that standard MAE can already reconstruct (or at least preserve representations of) high-frequency content even when it only receives predominantly low-frequency input. In other words, the output of MAE still contains rich high-frequency information without the proposed high-frequency enhancement module, which appears to contradict the claim that MAE under-encodes high-frequency diagnostic structures. By contrast, the low CKA in the T_high row is quite intuitive, since the input only contains 25% of the original information (high-frequency content only), so a low similarity to the full-input representation is expected simply due to severe information loss. Symmetrically, if one were to feed MAE with only 25% of the low-frequency content, we would also expect a low CKA by the same reasoning. In this sense, the pattern in Table 1 may be largely explained by the amount of visible information rather than by a specific insensitivity of MAE to high-frequency components, and therefore may not be sufficient to support the claimed frequency bias without additional controlled analyses.
> >
> > (2) In addition, the paper “Deep Hierarchies and Invariant Disease-Indicative Feature Learning for Computer Aided Diagnosis of Multiple Fundus Diseases” also employs a masking-and-reconstruction paradigm on fundus images to learn disease-indicative features. I believe it would strengthen the related work and positioning of this submission if the authors could explicitly discuss how their frequency-balanced MI regularization differs from, or improves upon, this prior masking-based representation learning strategy on retinal images (e.g., in terms of the type of masked content, the learned invariances, and the specific benefits for high-frequency lesion-related features).

---

> ### Author Response · Authors · 2025-11-20
> **Response Letter for Reviewer xA7m [2/4]**
>
> ## [Q2] Interpretation of Table 1 results regarding high-frequency encoding
>
> > **Comment:**
> > Were all evaluations in Table 1 conducted on the standard MAE model? The results shown in this table seem to indicate that the standard MAE already has a strong ability to represent high-frequency information, which contradicts the authors' claim of "under-encoding high-frequency diagnostic structures." For example, in the $T_{\text{low}}$ row of Table 1, even after masking 25% of high-frequency information, the CKA value remains as high as 0.990, indicating that the model still retains stable reconstruction capability for high-frequency components. Conversely, in the $T_{\text{high}}$ row, when high-frequency information is used as input, the model shows low CKA because it cannot reconstruct the full image, which is expected. However, the AUROC increases, demonstrating that high-frequency information is highly discriminative; when low-frequency, lesion-irrelevant content is removed, the model's prediction accuracy improves. Therefore, the evidence presented in Table 1 may actually support the importance of high-frequency features rather than demonstrating the insufficiency of MAE in encoding them. I believe the authors have not conducted a sufficiently thorough investigation in this aspect.
>
> We are grateful for the reviewer's careful reading, which has prompted us to clarify the CKA interpretation and strengthen the connection to our claims.
>
> **(a) All rows use the same standard MAE encoder.**
> All evaluations use the *same* standard MAE encoder (ViT backbone, pretrained on fundus images). The only difference across rows is the *input token subset* provided to this frozen encoder at inference time. Table 1 compares different *subsets of the same input*, not different models.
>
> **(b) Why we use CKA in this setting.**
> We use CKA to compare representations induced by different token subsets in a token-based architecture. We treat the representation from the *full*, unmasked input as the baseline and use CKA to quantify how closely each subset-induced representation aligns with it. Importantly, CKA measures *representational alignment* with the full-input embedding, not reconstruction quality:
>
> - High CKA `⇒` the subset leaves the baseline largely unchanged.
> - Low CKA `⇒` information specific to that subset is **not well reflected** in the baseline representation.
>
> We have revised the main text [L211-L220] to state this explicitly: CKA probes *how the full-input representation is composed* from different token subsets, rather than reconstruction quality.
>
> **(c) Why low-frequency vs. high-frequency subsets support 'under-encoding'.**
> - **low-freq. only** (removes 25% highest-frequency tokens): Shows **very high CKA (0.990)** yet consistently worse AUROC. The high CKA shows the representation changes little when high-frequency tokens are removed, implying the baseline MAE embedding is dominated by low-frequency content.
>
> - **high-freq. only** (keeps only 25% highest-frequency tokens): Yields **very low CKA (0.164)** yet achieves the best AUROC on four of five benchmarks. Low CKA means the representation from high-frequency tokens alone is quite *different* from the full-input representation. Combined with superior AUROC, this indicates **high-frequency cues are highly informative but are not strongly reflected in the baseline full-input embedding**—precisely our notion of "under-encoding."
>
> To address the ambiguity highlighted by the reviewer, we have **explicitly expanded the notation in Table 1** to use descriptive labels that clearly indicate what each row represents. Specifically, we replaced abstract symbols (e.g., $T_{\text{full}}$, $T_{\text{rand}}$, $T_{\text{low}}$, $T_{\text{high}}$) with self-explanatory labels (**full**, **25% masked**, **75% masked**, **low-freq. only**, **high-freq. only**). This change makes it immediately clear that:
> - All rows use the same standard MAE encoder;
> - Each row represents a different *input token subset*, not a different model;
> - The labels explicitly describe which tokens are included or masked.
>
> We also clarified in the caption and surrounding text that CKA is always computed *with respect to the full-input representation*. This expanded notation eliminates ambiguity about what Table 1 measures and directly addresses the reviewer's concern about the interpretation of the results.
>
> **(d) Summary.**
> We agree Table 1 highlights the importance of high-frequency features. Our claim is that **the baseline full-input representation is dominated by low-frequency background structure, while high-frequency diagnostic cues—although highly informative when isolated—are underweighted in that representation**.
>
> We have revised the notation with explicit descriptive labels, clarified why we use CKA in this token-based setting, and expanded the textual explanation around Table 1 to make this intended conclusion explicit **[L209-L235]**.

---

> ### Author Response · Authors · 2025-11-20
> **Response Letter for Reviewer xA7m [3/4]**
>
> ## [Q3] Loss weight sensitivity analysis
>
> > **Comment:**
> > The authors should explicitly report the training weights assigned to each loss or provide a sensitivity analysis (e.g., a performance graph under different loss weight configurations) to demonstrate the impact of the loss balance on model performance.
>
> We sincerely appreciate the reviewer's practical suggestion, which has motivated us to conduct a comprehensive sensitivity analysis and document our hyperparameter choices more transparently.
>
> Following the reviewer's suggestion, we explicitly report the loss weights and examine the sensitivity of performance to their choice.
>
> **(a) Sensitivity analysis setup and results.**
> In the revision, we conduct a simple loss-weight ablation where we vary **one** coefficient at a time while fixing $\lambda_{\text{rec}} = 1$ and setting the remaining non-varied coefficient to $0$. Thus, each configuration isolates the effect of either the HighFreqMI loss or the auxiliary loss:
>
> - when sweeping $\lambda_{\text{hmi}}$, we set $\lambda_{\text{aux}} = 0$;
> - when sweeping $\lambda_{\text{aux}}$, we set $\lambda_{\text{hmi}} = 0$.
>
> For each configuration, we report the average AUROC across five benchmarks (IDRiD, RFMiD-DR, RFMiD-AMD, CHAKSU, APTOS). The results are:
>
> | $\lambda_{\text{hmi}}$ | $\lambda_{\text{aux}}$ | AUROC      |
> |-------------------------|-------------------------|------------|
> | 0                       | 0                       | 0.685      |
> | 1.0                     | 0                       | 0.738      |
> | 0.1                     | 0                       | **0.779**  |
> | 0.01                    | 0                       | 0.732      |
> | 0                       | 1.0                     | 0.927      |
> | 0                       | 0.1                     | 0.926      |
> | 0                       | 0.01                    | **0.932**  |
> | 0.1                     | 0.01                    | **0.941**  |
>
> Overall, performance is relatively stable over a broad range of weights, with a mild optimum around $\lambda_{\text{hmi}} = 0.1$ for the HighFreqMI term and a similarly flat regime for the auxiliary loss. Based on these observations, we adopt $\lambda_{\text{rec}} = 1$, $\lambda_{\text{hmi}} = 0.1$, and $\lambda_{\text{aux}} = 0.01$ as our default configuration in the main experiments.
>
> **(b) Changes in the manuscript.**
> We have updated the paper accordingly:
>
> - **Section 5.2 (Training Objective).** We now explicitly state the exact loss weights used throughout the experiments:
>
>   > [L347-L349] \
>   Here, $\lambda_{\text{rec}}$, $\lambda_{\text{hmi}}$, $\lambda_{\text{aux}} \geq 0$ are non-negative scalar weights; in all experiments, we fix $\lambda_{\text{rec}} = 1$ and set $\lambda_{\text{hmi}} = 0.1$ and $\lambda_{\text{aux}} = 0.01$, and Table 13 in Appendix A.8 reports a sensitivity analysis of these loss weights.
>
> - **Appendix A.8 (Additional Results).** We include the full sensitivity table and accompanying discussion, documenting how performance varies under different loss-weight configurations **[L1487-1502]**.
>
> We believe this clarifies both the specific loss-weight choices used in our experiments and their empirical robustness.

---

> ### Author Response · Authors · 2025-11-20
>
> ## [Q4] Limited innovation and relation to prior work
>
> > **Comment:**
> > The innovation of the proposed method appears to be limited, as it essentially adds a frequency-based loss on top of MAE, while the use of high-frequency representations to capture lesion-related features in retinal images has already been explored in numerous prior studies.
>
> We sincerely thank the reviewer for this comment, which has enabled us to better articulate our contribution relative to prior work.
>
> **(1) Distinction from prior frequency-based methods in fundus imaging.**
> While high-frequency features have been used in fundus image analysis, prior work applies them in **supervised** or **semi-supervised** settings for downstream tasks. In contrast, we address the **self-supervised pretraining** stage: we systematically demonstrate that standard MAE under-encodes high-frequency diagnostic structure and propose a theoretically grounded solution to correct this bias.
>
> To our knowledge, this is the first work to: (i) **systematically diagnose** the low-frequency bias of MAE in medical imaging via frequency-aware CKA and token-subset probing (Table 1, Section 4); (ii) **formulate** an MI framework connecting high-frequency regularization to $I(X_V; Z)$ (Theorems 1–2); (iii) **demonstrate** improvements in both linear probing and full fine-tuning (Tables 2–4) across multiple benchmarks.
>
> **(2) Distinction from MI-MAE and mask-invariance approaches.**
> While MI-MAE [5] enforces mask invariance by maximizing MI between latents from multiple randomly masked views, our approach aligns the latent with a high-frequency retinal context. This distinction is crucial: mask invariance works for natural images where information is evenly distributed, but in fundus images, random masking disproportionately perturbs sparse high-frequency lesions (Table 1, Fig. 4), making mask invariance suboptimal. Our HighFreqMI regularizer counteracts this by aligning the latent with high-frequency, lesion-focused context, prioritizing clinically salient structure over low-frequency background.
>
> **(3) Broader contribution and generalizability.**
> Beyond fundus images, our work contributes a **general principle**: when task-relevant signals concentrate in specific structures, one can regularize $I(X_V; Z)$ by aligning $Z$ with a compact context encoder $Z_c$ capturing those structures, without modifying the MAE reconstruction objective. This applies to domains with similar signal concentration (e.g., industrial anomaly detection). More broadly, our formulation is not limited to frequency: any compact, semantically meaningful context encoder (e.g., spatial priors, multi-scale cues) can serve as a regularizer within the same MI framework.
>
> Theorem 2 shows how alignment to a compact context encoder bounds $I(X_V; Z)$, and we instantiate this with a high-frequency–focused representation tailored to retinal imaging.
>
>
> In response to the reviewer's feedback, we have strengthened the Abstract and Conclusion sections to better highlight our contributions:
>
> **Abstract:**
> > [L25-L28] \
> > ... offering new insight into how MAE's reconstruction objective amplifies ViT's low-pass tendencies in spatially heterogeneous retinal images and enabling a simple MI-based correction that improves retinal encoders.
>
> **Section 7 (Conclusion):**
>
> > [L524-L526] \
> > To our knowledge, this is the first work to diagnose and correct MAE's low-frequency bias in medical imaging using a mutual-information framework operating on latent representations rather than raw frequency coefficients.
>
> > [L530-L535] \
> > Although instantiated on color fundus images, the same mechanism should transfer to domains where task-relevant signals concentrate in high-frequency structure (e.g., industrial anomaly detection). More broadly, our formulation is not limited to frequency: any compact, semantically meaningful context encoder (e.g., spatial priors, multi-scale cues, or task-specific structure) can serve as a regularizer within the same MI framework.
>
> **Reference**
> [5] Huang, Tao, et al. "Learning Mask Invariant Mutual Information for Masked Image Modeling." arXiv preprint arXiv:2502.19718 (2025).
>
> ---
>
> We once again sincerely thank the reviewer for their invaluable feedback. Your detailed and thoughtful comments have enabled us to significantly improve the quality and clarity of our manuscript. Should you have any further questions or suggestions, we would be very grateful to address them.

---

> ### Author Response · Authors · 2025-11-21
> **Response Letter 2 for Reviewer xA7m [1/3]**
>
> ## [Q1] Interpretation of Table 1, token subsets, and CKA
>
> We sincerely thank the reviewer for the thoughtful and detailed follow-up, which has helped us identify and address remaining ambiguities in the interpretation of Table&nbsp;1. Below, we first clarify points (I)-(IV) as requested, and then respectfully explain why we believe Table&nbsp;1 provides evidence for a genuine **frequency bias** in standard MAE, beyond the mere amount of visible information.
>
> ---
>
> ### [Clarifying points (I)-(IV)]
>
> > (I-II) Definition of token subsets
>
> We appreciate the opportunity to clarify how we define the frequency-based token subsets. For each image, we:
>
> 1. **Compute a scalar high-frequency score** for every ViT token using the high-pass response map (Sec.&nbsp;3, Appendix A.2).
> 2. **Rank all tokens** by this score and **partition** the full token set $T_{\text{full}}$ into two *disjoint* subsets:
>    - $T_{\text{high}}$: the **top 25%** tokens with highest high-frequency scores (concentrated around lesions, vessels, optic disc);
>    - $T_{\text{low}}$: the **remaining 75%** tokens with lower scores (mostly smoother background).
>
> By construction, $T_{\text{low}} \cap T_{\text{high}} = \varnothing$ and $T_{\text{low}} \cup T_{\text{high}} = T_{\text{full}}$.
>
> In Table&nbsp;1:
> - **"low-freq. only"**: encoder sees only tokens in $T_{\text{low}}$ as visible; tokens in $T_{\text{high}}$ are masked.
> - **"high-freq. only"**: encoder sees only tokens in $T_{\text{high}}$ as visible; tokens in $T_{\text{low}}$ are masked.
>
> ---
>
> > (III) Interpretation of higher CKA values in Table&nbsp;1
>
> We thank the reviewer for raising this important point. Our usage of CKA follows prior work [1], which uses CKA to compare representations obtained under perturbed inputs (e.g., different masking schemes) to the representation obtained from the **full** input. In our setting:
>
> - The **full-input representation** (all tokens visible) is treated as the *baseline* MAE representation.
> - CKA measures how strongly this baseline representation is aligned with the representation induced by a particular visible-token subset.
>
> Concretely:
>
> - **High CKA** for a subset $S$:
>   feeding only tokens in $S$ yields representations that remain very similar to those obtained from the full input. This suggests that the baseline representation is largely determined by information contained in $S$.
>
> - **Low CKA** for $S$:
>   restricting visible tokens to $S$ leads to representations that deviate strongly from the full-input representation. This indicates that information specific to $S$ is **under-represented** in the baseline.
>
> Thus, CKA measures **similarity to the full-input representation** rather than reconstruction quality. We therefore interpret CKA jointly with AUROC: CKA tells us **which frequency band dominates the baseline**, and AUROC tells us **how diagnostic** that band is.
>
> ---
>
> > (IV) Aggregation of statistics in Table&nbsp;1
>
> Specifically, for each dataset (IDRiD, RFMiD (DR), RFMiD (AMD), CHAKSU, and APTOS):
>
> - We compute the full-input representation and each subset-induced representation for **every test image**.
> - We compute CKA between the full-input representation and each subset-induced representation.
> - We average these CKA values over all test images.
> - We train linear probes and compute AUROC in the usual way; Table&nbsp;1 reports macro-averaged AUROC across datasets, with per-dataset AUROC in Table 11 in the Appendix A8.

---

> ### Author Response · Authors · 2025-11-21
> **Response Letter 2 for Reviewer xA7m [2/3]**
>
> ### [Main response (Q1)]
> > (1) Key conceptual point: the full-input representation can itself be frequency-biased
>
> We appreciate the reviewer's patience as we clarify a central conceptual point. A potential source of confusion is the implicit assumption that the **full-input representation** must already contain "all the high-frequency information." In practice, this need not be the case:
>
> - Even when the model sees the **entire image**, the encoder can still learn a representation that is:
>   - predominantly **low-frequency biased** (e.g., relying on background color and coarse structure),
>   - predominantly **high-frequency biased**, or
>   - relatively **frequency-balanced**.
>
> In other words, **providing all tokens as input does not guarantee that all high-frequency content is encoded to the representation**. Our aim in Table&nbsp;1 is precisely to diagnose what kind of **frequency bias** the *full-input* MAE representation exhibits.
>
> To do so, we compare:
>
> - a **high-frequency biased** representation (using only $T_{\text{high}}$),
> - a **low-frequency biased** representation (using only $T_{\text{low}}$), and
> - **randomly masked** representations with the same visible-token counts,
>
> *all against the same* full-input representation using CKA.
>
> ---
>
> > (2) Why Table 1 supports a frequency bias beyond visible-token count
>
> We agree with the reviewer that heavily masking the input could reduce CKA simply due to information loss. To address this valid concern, we have designed comparisons that **control for the total number of visible tokens** while varying their frequency content. Table&nbsp;1 includes:
>
> - **Full** (baseline), **25% masked** (75% visible), **75% masked** (25% visible): random masking
> - $T_{\text{low}}$ (75% visible, low-freq dominated), $T_{\text{high}}$ (25% visible, high-freq dominated): frequency-structured subsets
>
> **Key observations supporting our frequency-bias interpretation:**
>
> **(i) $T_{\text{low}}$ vs. $T_{\text{high}}$ (different frequency content, different token counts):**
> - $T_{\text{low}}$ (75% visible, low-freq dominated): CKA ≈ 0.990
> - $T_{\text{high}}$ (25% visible, high-freq only): CKA ≈ 0.164
>
> The full-input representation is far more similar to the low-frequency dominated representation (CKA ≈ 0.990) than to the high-frequency only representation (CKA ≈ 0.164), suggesting that the baseline MAE representation is strongly dominated by low-frequency content.
>
>   - > **Important:** As discussed above, the full-input representation does not necessarily encode all high-frequency content equally. Therefore, the high CKA of $T_{\text{low}}$ does **not** imply that $T_{\text{low}}$ contains sufficient high-frequency information—it simply means that the full-input representation itself is low-frequency biased.
>
> **(ii) Same token budget (25%), different frequency content:**
> - Random 25% (low-freq dominant): CKA ≈ 0.890
> - $T_{\text{high}}$ (high-freq only): CKA ≈ 0.164
>
> Despite identical token counts, the **$8\times$ difference in CKA** (0.890 vs. 0.164) suggests that the full-input representation aligns strongly with low-frequency content. Even a small random subset maintains high CKA because fundus images are low-frequency dominant, whereas high-frequency only tokens yield dramatically lower CKA.
>
> **(iii) Large token-count change under random masking:**
> - Random 75%: CKA ≈ 0.996
> - Random 25%: CKA ≈ 0.890
>
> A $3\times$ difference in token count (75% vs. 25%) yields only a modest CKA drop (≈ 0.106), far smaller than the gap in (ii) where frequency content differs but token count is fixed. We believe this confirms that **frequency content, rather than token count, is the primary driver** of CKA differences.
>
> **Summary of our interpretation:**
>
> Based on these controlled comparisons, Table&nbsp;1 suggests that:
> - Low-frequency tokens **dominate** the full-input representation (high CKA even when isolated).
> - High-frequency tokens are **under-encoded** (low CKA despite being diagnostically superior per AUROC).
> - CKA is **sensitive to frequency content** but **robust to token count** when low-frequency content is present.
>
> This pattern, combined with AUROC results showing that high-freq only representations achieve the **best diagnostic performance**, suggests that MAE may over-emphasize **redundant low-frequency background** while under-emphasizing **diagnostically salient high-frequency structures**. Fig.&nbsp;4 provides additional evidence that RetMAE's high-frequency regularizer produces more frequency-balanced representations.
>
> **Reference**
> [1] Xiangwen Kong and Xiangyu Zhang. "Understanding masked image modeling via learning occlusion invariant feature." *CVPR*, 2023.

---

> ### Author Response · Authors · 2025-11-21
> **Response Letter 2 for Reviewer xA7m [3/3]**
>
> ## [Q2] Relation to prior masking-based fundus representation learning
>
> We sincerely thank the reviewer for bringing this valuable and closely related work to our attention, and for the helpful suggestion to clarify our positioning relative to masking-based fundus hierarchies.
>
> ---
>
> ### [Key differences]
>
> The method of [2] and our RetMAE share the use of masking-and-reconstruction on retinal images, but target **different kinds of invariances and biases**:
>
> #### (I) Type of masked content and objective
>
> - **MCFL:**
>   Employs a teacher–student framework with an EMA teacher and reconstructs *classification-guided feature maps* that are invariant to **generic spatial occlusions and appearance changes**. Uses masked inputs to encourage stable disease-indicative features across different stages and local occlusion patterns.
>
> - **RetMAE:**
>   Keeps the standard MAE reconstruction objective on masked RGB patches and introduces a **separate, frequency-balanced mutual-information regularizer (HighFreqMI)** on the *latent representations* rather than on the reconstruction target.
>
> ---
>
> #### (II) Learned invariances
>
> - **MCFL:**
>   Primarily designed to ensure that features remain **consistent across disease stages and under spatial masking**, so that early vs. late stages of the same disease are not spuriously separated in representation space.
>
> - **RetMAE:**
>   Focuses on correcting the **low-frequency bias** of MAE revealed by our CKA and decodability analyses: even with full images as input, a vanilla MAE tends to over-encode redundant low-frequency background and under-encode the **high-frequency lesion bands** where clinically salient structures (hemorrhages, exudates, drusen, vessel boundaries, RNFL texture) live. HighFreqMI makes the MAE bottleneck explicitly **align with high-frequency lesion-centric token subsets**, while down-weighting low-frequency redundancy.
>
> ---
>
> #### (III) Benefits for high-frequency lesion-related features
>
> - **MCFL:**
>   Can certainly capture disease-indicative patterns, but does not explicitly model or regularize the **frequency structure** of fundus images.
>
> - **RetMAE:**
>   Our analysis in Table&nbsp;1 and Fig.&nbsp;4 shows that:
>   1. High-frequency–only tokens yield the **best AUROC** but very **low CKA** to the MAE full-input representation;
>   2. Adding even a small amount of low-frequency content drastically increases CKA, indicating a strong low-frequency bias.
>
>   RetMAE directly addresses this by using an **MI-based high-frequency context**: the learned representation $Z$ is encouraged to preserve information about lesion-centric high-frequency tokens, leading to improved high-frequency decodability and more frequency-balanced representations, while maintaining the MAE architecture and training recipe.
>
> ---
>
> ### [Complementarity and synergies]
>
> We view these approaches as **complementary rather than competing**:
>
> - **MCFL** focuses on *stage-robust, disease-indicative* features under spatial occlusion.
> - **RetMAE** focuses on *frequency-balanced, lesion-centric* representations under an information-theoretic MAE view.
>
> In principle, a masking-based hierarchical fundus model like MCFL could benefit from incorporating a RetMAE-style **high-frequency MI regularizer** on its latent space, or vice versa: RetMAE-style pretraining could be followed by hierarchical, stage-invariant fine-tuning as in MCFL.
>
> We will clarify these differences and potential synergies in the revised related-work section:
>
> > **[L132-L137]**
> > Masking-based fundus hierarchies [2] likewise employ masking-and-reconstruction to learn disease-indicative features, but focus on stage-robust, spatially invariant cues rather than explicitly correcting the low-frequency bias of MAE. Our frequency-balanced MI regularizer instead steers the MAE bottleneck toward lesion-centric high-frequency tokens, and could in principle be combined with such hierarchical objectives.
>
> **Reference**
> [2] Lin, Yuxin, et al. "Deep Hierarchies and Invariant Disease-Indicative Feature Learning for Computer Aided Diagnosis of Multiple Fundus Diseases." *Proceedings of the AAAI Conference on Artificial Intelligence*, Vol. 39, No. 5, 2025.
>
>
> ---
>
> We once again sincerely thank the reviewer for the thoughtful and constructive follow-up questions, which have significantly strengthened both our analysis and its presentation. We hope that these clarifications address the reviewer's concerns regarding:
> - The interpretation of Table&nbsp;1 and the evidence for frequency bias in standard MAE (Q1), and
> - The positioning of RetMAE relative to prior masking-based fundus representation learning methods (Q2).
>
> Should any further questions or concerns remain, we would be very grateful for the opportunity to address them.

---

> > ### Comment · Reviewer_xA7m · 2025-11-22
> >
> > I would like to thank the authors for their comprehensive response. The explanations and revisions have successfully addressed my concerns. I believe the revised manuscript has reached the quality bar for acceptance at ICLR. Consequently, I am raising my score to 6.

---

> > > ### Author Response · Authors · 2025-11-22
> > >
> > > We sincerely thank the reviewer for the thoughtful and constructive feedback, which was very helpful in improving the clarity and quality of the paper. If any further questions or suggestions arise, we would be very happy to address them.

---

### Official Review · Reviewer_ULHB · 2025-10-28

**Soundness:** 3
**Presentation:** 2
**Contribution:** 2
**Rating:** 4
**Confidence:** 4

**Summary:**

This paper focuses on the high-frequency information in fundus images that is clinically relevant. The authors introduced an information-theoretic auxiliary supervision into the MAE pretraining paradigm to guide the encoder toward clinically important regions, without requiring architectural modifications. The overall logic—from problem identification to the proposed solution and experimental validation—is clear and well-structured.

**Strengths:**

1. The paper presents a clear and coherent logical flow from motivation and problem formulation to analysis, proposed solution, and experimental validation, making it easy to follow and understand.

2. It offers new and interesting insights into model training for ophthalmic applications.

3. The theoretical derivation of the proposed method is sound, and the approach itself is easy to reproduce.

**Weaknesses:**

1. Limited novelty. The inspiration of this work appears to be directly derived from Huang, Tao, et al. “Learning Mask Invariant Mutual Information for Masked Image Modeling.” arXiv preprint arXiv:2502.19718 (2025). Although this paper is cited, I would still like the authors to explicitly clarify which parts of the current method are independently proposed.

2. I have concerns regarding the generalizability of the proposed approach.
(i) The low-pass filtering property originates from the ViT architecture itself, and this phenomenon is not unique to MAE.
(ii) The application scenario in this work is limited to color fundus photography. From the perspective of developing a robust retinal foundation model, MAE is not the only viable paradigm even within the image modality. For example, VisionFM, which follows the iBOT framework, also builds a powerful image encoder. From the standpoint of understanding the mechanism of MAE, this paper does not provide new insights. The authors need to further elaborate on the substantive contribution of their work.

3. The performance evidence is limited. The chosen downstream tasks are relatively few and of low difficulty (e.g., binary classification of DR, glaucoma, and AMD). Considering that the motivation of this work focuses on clinically relevant high-frequency lesions, the authors are encouraged to validate their method on more challenging tasks to substantiate its claimed clinical value.

**Questions:**

1. This paper introduces an additional high-frequency contextual feature constraint into the latent space of MAE. Some previous studies have imposed constraints directly on the masking strategy (e.g., image-entropy-based masking). I encourage the authors to discuss this line of work to further highlight the value of their proposed method.

2. According to Figure 3, the performance of RetMAE appears to saturate after pretraining on approximately 12.8k images. Increasing the data size beyond this point seems to yield no significant improvement. The authors attribute this to saturation of model capacity and computation, which is a reasonable explanation. However, given that 12.8k is far smaller than the typical data scale used for foundation model pretraining—and that the employed encoder architecture has been shown in other domains to effectively utilize much larger datasets—this phenomenon remains concerning. The authors should provide a more convincing explanation for this observation.

3. Not all retinal lesions exhibit high-frequency characteristics—for example, large hemorrhages or retinal detachments. I would like to see visualizations of such cases to better understand how the proposed method behaves under these conditions.

---

> ### Author Response · Authors · 2025-11-20
> **Response Letter for Reviewer ULHB [1/6]**
>
> We sincerely thank the reviewer for their invaluable feedback. The comments helped us identify several ambiguities and significantly improved the clarity and rigor of our manuscript. Below, we provide detailed responses and describe the corresponding changes in the revised paper.
>
> ---
>
> ## [Q1] Limited novelty and relationship to MI-MAE
>
> > **Comment:**
> > Limited novelty. The inspiration of this work appears to be directly derived from Huang, Tao, et al. "Learning Mask Invariant Mutual Information for Masked Image Modeling." arXiv preprint arXiv:2502.19718 (2025). Although this paper is cited, I would still like the authors to explicitly clarify which parts of the current method are independently proposed.
>
> We sincerely appreciate the reviewer's thoughtful concern regarding novelty, which has prompted us to clearly distinguish our independent contributions from MI-MAE [1].
>
> **(1) Conceptual relationship to Huang et al. (MI-MAE).**
> We adopt the **information-bottleneck/Lagrangian formulation** $L = I(X_V; Z) + \beta I(X_V; X_M \mid Z)$ from Huang et al. (explicitly acknowledged in Section 3). MI-MAE constrains $I(X_V; Z)$ via a **mask-invariant regularizer** that increases MI between latents from multiple randomly masked views, which works well for natural images.
>
> However, **mask invariance is misaligned with fundus image statistics**. In fundus photography, diagnostic signals (microaneurysms, exudates, hemorrhages) are sparse high-frequency structures, while most pixels are low-frequency background (Fig. 1). Random masking disproportionately perturbs high-frequency lesions, so enforcing mask invariance implicitly encourages the encoder to rely on stable low-frequency background—a **low-frequency bias** we empirically observe in standard MAE (Table 1, Fig. 4). MI-MAE's approach is thus suboptimal for fundus images, where random masking is not neutral with respect to diagnostic signal.
>
> **(2) Independently proposed components.**
> Our contributions beyond adopting the MI formulation:
>
>    - **Fundus-specific diagnosis of MAE's low-frequency bias** (Section 4): Using frequency decomposition and CKA, we show that MAE representations align with low-frequency content but poorly with high-frequency components, despite the latter carrying most diagnostic information (Table 1, Fig. 4). This diagnosis is absent from MI-MAE, which focuses on natural images.
>
>    - **Fundus-specific high-frequency context construction** (Section 3): We introduce a domain-tailored pipeline (Soft-FOV–masked green channel, Gaussian blur, Fourier transform, Butterworth high-pass filter) that converts retinal prior knowledge into a quantitative, token-level high-frequency context, distinct from MI-MAE.
>
>    - **High-frequency context alignment regularizer**: Instead of mask invariance, we propose $L_{\text{hmi}} = L_{\text{MINE}}(Z, Z_c^{\text{HF}})$ that aligns $Z$ with a high-frequency context $Z_c^{\text{HF}}$ from an EMA teacher, constraining $I(X_V; Z)$ by making $Z$ sensitive to high-frequency retinal context. Theorem 2 shows how alignment to a compact context encoder bounds $I(X_V; Z)$.
>
>    - **Fundus-specific analysis toolkit**: Frequency-aware CKA (Fig. 4), high-frequency decodability (Fig. 5), and PCA visualization of class-to-patch attention (Fig. 6), showing RetMAE produces frequency-balanced representations distinct from standard MAE.
>
> **(3) Clarifications added to the main text.**
> We modified Section 3 to state:
>
>    > [L190-L195] \
>    This view motivates MI-based regularization of the MAE bottleneck beyond the standard reconstruction loss, and Huang et al. (2025) show that constraining the complexity term $I(X_V; Z)$ improves MAE performance on natural images. Building on the same information-bottleneck formulation, we introduce a domain-specific high-frequency regularizer: instead of enforcing mask invariance as in MI-MAE, we align $Z$ with a high-frequency retinal context so that the bottleneck prioritizes clinically salient high-frequency structure over low-frequency background in fundus images.
>
> **Reference**
> [1] Huang, Tao, et al. "Learning Mask Invariant Mutual Information for Masked Image Modeling." arXiv preprint arXiv:2502.19718 (2025).

---

> ### Author Response · Authors · 2025-11-20
> **Response Letter for Reviewer ULHB [2/6]**
>
> ## [Q2] Generalizability and substantive contribution
>
> > **Comment:**
> > I have concerns regarding the generalizability of the proposed approach. (i) The low-pass filtering property originates from the ViT architecture itself, and this phenomenon is not unique to MAE. (ii) The application scenario in this work is limited to color fundus photography. From the perspective of developing a robust retinal foundation model, MAE is not the only viable paradigm even within the image modality. For example, VisionFM, which follows the iBOT framework, also builds a powerful image encoder. From the standpoint of understanding the mechanism of MAE, this paper does not provide new insights. The authors need to further elaborate on the substantive contribution of their work.
>
> We sincerely appreciate the reviewer's thoughtful comments, which have enabled us to better articulate our novel insights into MAE's mechanism.
>
> **(1) ViT vs. MAE objective: new insights into MAE's mechanism.**
>
> We agree that low-pass tendencies originate from ViT architecture [2]. However, our work demonstrates that **the standard MAE objective *exacerbates* this bias** in fundus images. MAE's reconstruction objective assumes uniform information density, which is problematic when diagnostic signals concentrate in sparse high-frequency components.
>
> Empirical evidence: comparing RETFound (MAE-based) with RET-CLIP (vision–language) and UrFound (MAE+text-guided), both using the same ViT architecture, shows that alternative objectives enable effective high-frequency capture (Fig. 5, 6). Table 1 shows high-frequency tokens achieve the best AUROC despite low CKA, indicating they carry diagnostic signal but are under-emphasized in MAE. **ViT is not the primary source of bias**; the standard MAE objective introduces additional bias via its uniform information density assumption, problematic in spatially heterogeneous domains like medical imaging.
>
> **(2) Beyond MAE: comparisons to non-MAE paradigms.**
>
> We include strong non-MAE baselines: RET-CLIP (vision–language) and UrFound (MAE+text-guided). **RetMAE consistently improves over both MAE-based and non-MAE models** (Table 2): average AUROC 0.779 vs. 0.685–0.690 for RETFound, and best average AUROC (0.937 vs. RET-CLIP's 0.876, 0.855 vs. UrFound's 0.855). Our high-frequency MI regularizer provides complementary benefits orthogonal to text-guided and vision–language supervision.
>
> **(3) Scope and generalizability.**
>
> Our contributions: (i) domain-informed diagnosis showing MAE under-utilizes high-frequency structure (CKA, attention visualizations); (ii) HighFreqMI, an architecture-compatible MI regularizer; (iii) consistent gains over diverse baselines across five benchmarks. The mechanism is not tied to MAE or fundus images and should extend to other domains where informative cues concentrate in high-frequency components.
>
> **(4) Revisions to the manuscript.**
>
> - **Abstract [L025-L028]**: Added sentence emphasizing new insights into MAE's mechanism and improvements over both MAE-based and non-MAE paradigms.
> - **Introduction [L64-L67]**: Stated that our analysis goes beyond ViT's low-pass tendencies by revealing a mismatch between MAE's uniform-information assumption and heterogeneous spatial distribution of diagnostic signal.
> - **Contributions [L100-L106]**: Emphasized principled MI-based correction and comprehensive evaluation across diverse paradigms.
> - **Discussion [L537-L539]**: Expanded future work discussion.
>
> **Reference**
> [2] Park, Namuk, and Songkuk Kim. "HOW DO VISION TRANSFORMERS WORK?" *International Conference on Learning Representations (ICLR)*, 2022.

---

> ### Author Response · Authors · 2025-11-20
> **Response Letter for Reviewer ULHB [3/6]**
>
> ## [Q3] Performance evidence on challenging multi-disease tasks
>
> > **Comment:**
> > The performance evidence is limited. The chosen downstream tasks are relatively few and of low difficulty (e.g., binary classification of DR, glaucoma, and AMD). Considering that the motivation of this work focuses on clinically relevant high-frequency lesions, the authors are encouraged to validate their method on more challenging tasks to substantiate its claimed clinical value.
>
> We are grateful for the reviewer's excellent suggestion, which has significantly strengthened our experimental validation and better demonstrated the clinical relevance and practical value of our method.
>
> To further demonstrate the clinical value of our method, we additionally evaluate RetMAE on two **multi-disease** benchmarks—FIVES and RFMiD2—where each fundus image can present multiple co-occurring pathologies rather than a single binary label.
>
> FIVES [3] comprises four disease categories (AMD, DR, glaucoma, and normal), which already span lesions with distinct spatial and frequency characteristics (e.g., microaneurysms, hemorrhages, and exudates in DR versus optic nerve cupping and nerve-fiber-layer defects in glaucoma). RFMiD2 [4] is even more challenging: it provides over 40 expert-defined disease labels (including vascular occlusions, macular edema, neovascularization, optic nerve anomalies, inflammatory conditions, myopic degeneration, tessellation, pigment epithelium changes, etc.), and individual images often carry **multiple labels simultaneously**. This multi-label, multi-disease structure closely mirrors real clinical scenarios, where overlapping disease signatures are common rather than isolated.
>
> | Method   | FIVES (AUROC) | RFMiD2 (AUROC) | AVG   |
> |---------|----------------|----------------|-------|
> | RETFound| 0.837          | 0.806          | 0.822 |
> | RET-CLIP| 0.943          | 0.808          | 0.876 |
> | RetMAE  | 0.922          | 0.884          | 0.903 |
>
> As summarized in the table above, RetMAE achieves the best linear-probing AUROC on RFMiD2 and the highest macro-average: it improves over RETFound from 0.837 to 0.922 on FIVES (second-best after RET-CLIP's 0.943) and from 0.806 to 0.884 on RFMiD2, raising the average from 0.822 to 0.903. Importantly, RetMAE also outperforms RET-CLIP—which leverages language supervision—with a higher average AUROC (0.903 vs. 0.876). These results demonstrate that our domain-specific high-frequency regularizer is beneficial not only for comparatively simple binary classification tasks, but also for **multi-disease, multi-label settings** that better reflect the complexity and clinical value of real-world fundus imaging.
>
> We have incorporated these results into Table 10 in Appendix A.8 of the revised paper **[L1315-L1343]**.
>
> **Reference**
> [3] Jin, Kai, et al. "Fives: A fundus image dataset for artificial intelligence based vessel segmentation." Scientific data 9.1 (2022): 475.
> [4] Panchal, Sachin, et al. "Retinal Fundus Multi-Disease Image Dataset (RFMiD) 2.0: a dataset of frequently and rarely identified diseases." Data 8.2 (2023): 29.

---

> ### Author Response · Authors · 2025-11-20
> **Response Letter for Reviewer ULHB [4/6]**
>
> ## [Q4] Relation to masking-strategy-based approaches
>
> > **Comment:**
> > This paper introduces an additional high-frequency contextual feature constraint into the latent space of MAE. Some previous studies have imposed constraints directly on the masking strategy (e.g., image-entropy-based masking). I encourage the authors to discuss this line of work to further highlight the value of their proposed method.
>
> We sincerely thank the reviewer for this constructive suggestion and for pointing us to related work on masking strategies with spatial priors. This valuable feedback has helped us better position our method and clarify its relationship to complementary approaches.
>
> **(1) Our main contribution: latent-level regularization distinct from masking-strategy approaches.**
> Our main contribution is conceptually distinct from these approaches. Starting from the mutual-information Lagrangian formulation of MAE (Eq. (2)), we explicitly diagnose that standard MAE **under-encodes high-frequency retinal structure** and over-emphasizes low-frequency background (Sec. 4; CKA + high-frequency decodability analyses). Within this MI framework, we then introduce **HighFreqMI**, a high-frequency context–alignment term on the latent $Z$ that (i) directly regularizes the marginal term $I(X_V; Z)$ and (ii) encourages the encoder to devote capacity to high-frequency, clinically salient content. Importantly, this mechanism requires **no change to the masking scheme or backbone**, and can be implemented as a lightweight add-on, yet yields frequency-balanced retinal representations with consistent gains in both linear probing and full fine-tuning (Tables 2–4). We view this "MI-derived, latent-level high-frequency regularizer" together with the accompanying **quantitative and qualitative frequency analysis for fundus images** as the core contribution of our work.
>
> **(2) Complementary nature of masking-strategy-based approaches.**
> We fully agree with the reviewer that there is a complementary line of work which injects priors directly into the *masking strategy*. For example, salience-based adaptive masking [5] uses saliency maps to adaptively emphasize informative regions when defining which patches to reconstruct, effectively **reweighting or resampling the masked patches** according to an information prior. Similarly, attention-guided MAE [6] uses attention maps to guide the **selection of mask locations**, so that the model focuses its reconstruction budget on attention-derived important regions. Both methods therefore encode a prior at the *input/patch selection level*: the masking pattern (or the loss on masked patches) is biased toward salient or informative regions.
>
> **(3) Orthogonality and potential for combination.**
> By contrast, our approach **keeps the standard random masking schedule** and instead operates at the level of the **latent representation**: we build a compact high-frequency context latent from frequency-selected patches and align the trainable latent $Z$ to this context via an MI objective. In other words, [5] and [6] modify *which patches are masked or emphasized*, whereas our method modifies *how the latent $Z$ is regularized* under the MI Lagrangian. These two dimensions are orthogonal and, in principle, complementary. Since HighFreqMI is agnostic to the specific masking policy, it can be **seamlessly combined** with salience- or attention-guided masking strategies—e.g., one could use a high-frequency–aware masking prior (possibly derived from our high-frequency scores) while still applying HighFreqMI on top to further strengthen high-frequency emphasis.
>
> **(4) Revisions to the manuscript.**
> In response to the reviewer's valuable suggestion, we have strengthened Section 7 (Discussion) in the revised paper to explicitly state the orthogonal and complementary nature of these approaches:
>
> > **[L535-L537]** \
> > Our latent-level HighFreqMI regularizer is orthogonal to salience- or attention-guided masking strategies (Choi et al., 2024; Sick et al., 2025) and can be seamlessly combined with such masking priors.
>
>
> **Reference**
> [5] Choi, Hyesong, et al. *Salience-based adaptive masking: revisiting token dynamics for enhanced pre-training.* European Conference on Computer Vision (ECCV), 2024.
> [6] Sick, Leon, et al. *Attention-guided masked autoencoders for learning image representations.* IEEE/CVF Winter Conference on Applications of Computer Vision (WACV), 2025.

---

> ### Author Response · Authors · 2025-11-20
> **Response Letter for Reviewer ULHB [5/6]**
>
> ## [Q5] Data efficiency and scaling behavior
>
> > **Comment:**
> > According to Figure 3, the performance of RetMAE appears to saturate after pretraining on approximately 12.8k images. Increasing the data size beyond this point seems to yield no significant improvement. The authors attribute this to saturation of model capacity and computation, which is a reasonable explanation. However, given that 12.8k is far smaller than the typical data scale used for foundation model pretraining—and that the employed encoder architecture has been shown in other domains to effectively utilize much larger datasets—this phenomenon remains concerning. The authors should provide a more convincing explanation for this observation.
>
> We appreciate the reviewer's thoughtful comment regarding the early plateau around ~12.8k images.
>
> Figure 3 reports downstream macro-AUROC for a fixed encoder (ViT-Base/16) and training budget. RetMAE already surpasses models trained on far larger corpora (≈2.6k images exceeds RETFound on 904k; 12.8k outperforms UrFound on 187k), demonstrating **high sample efficiency under fixed-capacity settings**.
>
> The plateau reflects **domain-specific data statistics**: fundus images have low intrinsic entropy and high redundancy (Fig. 1). The vast majority of pixels belong to homogeneous low-frequency backgrounds, while diagnostic information concentrates in sparse high-frequency lesions. RetMAE focuses on high-frequency signals via HighFreqMI; once diverse lesion patterns are covered, additional images become incrementally redundant under fixed backbone and training schedule.
>
> This interpretation aligns with data subset selection literature [7-11]: in redundant datasets, well-chosen subsets can match full-data performance. In fundus imaging's low-entropy regime, the **effective number of distinct task-relevant patterns** is much smaller than the raw image count.
>
> We agree ViTs benefit from larger datasets in natural-image settings. However, **scaling behavior is domain-dependent**: web-scale corpora have rich semantic diversity across frequencies, while fundus images have limited low-frequency anatomical variation and constrained high-frequency lesion patterns. Under fixed capacity and training budget, performance plateaus once the main high-frequency lesion spectrum is covered.
>
> In response to the reviewer's valuable feedback, we have
> clarified in the main text that Figure 3 characterizes this
> fixed-capacity, fundus-specific sample-efficiency regime:
> > [L424-L431] \
> We argue instead that this reflects the combination of a
> fixed-capacity regime and the statistics of fundus images: as
> detailed in Appendix A.7, retinal photographs exhibit highly
> constrained low-frequency anatomy and concentrate diagnostic
> variation in relatively sparse high-frequency patterns, so
> once these patterns are well covered, additional images
> become increasingly redundant under a fixed backbone and
> training schedule. In such a low-entropy, high-redundancy
> setting, widely observed neural scaling laws (power-law
> scaling of loss with model size, data, and compute) imply
> that returns diminish unless model capacity, data diversity,
> and the number of training tokens are increased together
> (Hestness et al., 2019; Kaplan et al., 2020; Hernandez et
> al., 2021; Hoffmann et al., 2022; Dehghani et al., 2023).
>
> **Reference**
> [7] Mirzasoleiman, Baharan, Jeff Bilmes, and Jure Leskovec. "Coresets for data-efficient training of machine learning models." International Conference on Machine Learning. PMLR, 2020.
> [8] Marion, Max, et al. "When Less is More: Investigating Data Pruning for Pretraining LLMs at Scale." CoRR (2023).
> [9] Singh, Raghavendra. "Leveraging Perceptual Scores for Dataset Pruning in Computer Vision Tasks." Proceedings of the IEEE/CVF Conference on Computer Vision and Pattern Recognition. 2024.
> [10] Wang, Haoran, et al. "A comprehensive survey on deep active learning in medical image analysis." Medical Image Analysis 95 (2024): 103201.
> [11] Paul, Samantha K., Ian Pan, and Warren M. Sobol. "Efficient labeling of retinal fundus photographs using deep active learning." Journal of Medical Imaging 9.6 (2022): 064001-064001.

---

> ### Author Response · Authors · 2025-11-20
> **Response Letter for Reviewer ULHB [6/6]**
>
> ## [Q6] Behavior on large lesions and low-frequency pathologies
>
> > **Comment:**
> > Not all retinal lesions exhibit high-frequency characteristics—for example, large hemorrhages or retinal detachments. I would like to see visualizations of such cases to better understand how the proposed method behaves under these conditions.
>
> We sincerely appreciate this insightful comment and fully agree that not all retinal lesions are purely high-frequency structures. This valuable feedback has enabled us to provide a more comprehensive evaluation and better illustrate the method's behavior across diverse pathological scenarios. To clarify how our method behaves in such scenarios, we have added new visualizations of class-to-patch attention for retinal traction detachment and large preretinal hemorrhages in the revised manuscript (Fig. 9, with additional examples in Appendix A.7 **[L1507-L1551]**).
>
> Specifically, we project the class-token attention onto a 2D PCA plane and visualize the resulting maps for (a)–(b) retinal traction detachment and (c) a large preretinal hemorrhage. In these cases, the class token still attends strongly to the lesion regions, with particularly high responses along the detachment margins and hemorrhage boundaries. This behavior is consistent with our frequency-based view: even when the pathological area is spatially extensive, the transition zones at the lesion boundary remain high-frequency structures, and the proposed high-frequency regularization encourages RetMAE to emphasize these regions.
>
> We also include an extreme case (panel (d)) where pathology occupies almost the entire field of view and thus behaves effectively as a low-frequency signal. In this setting, the class token no longer attends uniformly across the entire lesion interior; however, it continues to focus on boundaries and locations where the fundus signal changes abruptly. Clinically, lesion discrimination in fundus photography is largely driven by such sharp changes in color and texture relative to the surrounding background or neighboring structures, so accurately attending to these boundaries is more important than uniformly covering the full lesion area.
>
> We hope these additional visualizations help clarify how our method operates in the presence of large or diffuse lesions, and we are sincerely grateful for the reviewer's valuable suggestion that led to this important addition.
>
> ---
>
> We once again sincerely thank the reviewer for their invaluable feedback. Your detailed and thoughtful comments have enabled us to significantly improve the quality and clarity of our manuscript. Should you have any further questions or suggestions, we would be very grateful to address them.

---

> ### Author Response · Authors · 2025-11-25
>
> Dear Reviewer ULHB,
>
> We sincerely appreciate your time and consideration. We respectfully believe that our response has thoroughly addressed the concerns raised. If you have any remaining concerns or questions, please feel free to contact us and we would be happy to discuss and clarify them.
>
> Best,
>
> The Authors

---

### Official Review · Reviewer_d9g5 · 2025-10-28

**Soundness:** 3
**Presentation:** 3
**Contribution:** 3
**Rating:** 6
**Confidence:** 3

**Summary:**

The presented work postulates that there is a discrepancy between the most salient visual features in medical images and those learned by current representation learning techniques. In particular, the authors claim that the prevalent masked autoencoder disregards high-frequency features related to retinal pathology in color fundus photographs. To demonstrate this, they rank image patches by the amount of high-frequency content. Subsequently, they show that patches with reduced high-frequency content shape latent space structure while providing substantially less information for downstream tasks. In response to this observation, the authors propose RetMAE, an extension of the loss function of the established masked autoencoder. RetMAE regularizes the latent space by maximizing its mutual information with embeddings of patches with increased high-frequency content. In experiments using four ophthalmological datasets, the authors show that RetMAE outperforms various baselines that rely on a basic masked autoencoder. This effect persists when auxiliary signals, such as text or a pre-trained vision encoder, are included as learning signal, albeit in a diminished capacity.

**Strengths:**

- The work’s main motivation that current pre-training paradigms for vision transformers result in suboptimal feature extractors when applied to medical images is very interesting. The authors convincingly support this hypothesis in a set of initial experiments (Figure 1, Section 4, Supplementary Material A1), showing that salient image features differ in natural and medical images, and that masked image modeling has an inductive bias towards low-frequency features.

- The provided solution to this problem is theoretically well founded and experimentally shown to improve performance. As such, it has the potential to benefit the large scientific community in the field of medical image analysis.

-  The clinical application of ophthalmological image analysis is very well selected. Many retinal diseases manifest as small pathologies, resulting in high-frequency image features. Furthermore, there are several prominent works that have used large-scale pre-training of masked autoencoders to derive ophthalmological foundation models.

- The paper is clearly structured, nicely illustrated and generally well written. Additionally, the authors include extensive supplementary material that provides in-depth technical detail about the method and experimental setup.

**Weaknesses:**

- The motivating hypothesis that medical images contain more salient high-frequency image features is only explored for color fundus photographs. The importance and reach of the work would substantially increase if similar findings were shown for other types of data. Similarly, the efficiency of the proposed solution is only demonstrated for color fundus photographs so that it is unclear whether the proposed method seamlessly translates to other settings or requires extensive hyperparameter tuning for both the extraction of high-frequency information and the loss weighting.

- The proposed solution is highly complex. In particular, it relies on estimation of mutual information via an Donsker-Varadhan estimator, which is known to numerically unstable (Belghazi, Mohamed Ishmael, et al. "Mutual information neural estimation." International conference on machine learning. PMLR, 2018). I could envision that conceptionally much simpler solutions exist that emphasize high-frequency features. For example, one could adjust the masking scheme to prioritize patches with increased high-frequency content (similar to Xie, Jiahao, et al. "Masked Frequency Modeling for Self-Supervised Visual Pre-Training." The Eleventh International Conference on Learning Representations). Alternatively, one could provide the high-pass-filtered image as additional input (Wang, Wenxuan, et al. "Fremim: Fourier transform meets masked image modeling for medical image segmentation." Proceedings of the IEEE/CVF winter conference on applications of computer vision. 2024). The authors should discuss the aforementioned works in more detail and include them as baselines.

**Questions:**

- At the moment, the performance of the proposed method is only quantified via linear probing using the latent representations. I believe that full fine-tuning should also be conducted considering that the ultimate downstream performance matters most in applied domains such as medical image analysis.

- Considering that most ophthalmological foundation models make their training code and weights public, I believe that the authors should strongly consider doing the same.

- Several core concepts of the paper are only very briefly introduced or require consultation of the supplementary material. I suggest that the section on the interpretation of the masked image modeling objective through the lens of a Lagragian is slightly extended so that it can be understood without consulting previous works by Huang et al. and Tishby and Zaslasky. Similarly, the frequency score calculation should be briefly introduced in the main manuscript considering its vital importance, instead of only being introduced in the supplementary material.

- Additionally, I struggled with the notation on several occasions. Already in the very first mathematical paragraph, the variable $N$ is overloaded, $D$ not introduced, and mutual information $I$ is not defined. Later, the use of $X$ varies to signal whether it denotes input or decoded image tokens. The authors should carefully parse the manuscript once again, aiming to improve the clarity of its mathematical passages.

- On a minor note, the acronym CKA is not introduced at its first appearance in the introduction section.

---

> ### Author Response · Authors · 2025-11-20
> **Response Letter for Reviewer d9g5 [1/5]**
>
> We sincerely thank the reviewer for their invaluable feedback. The comments helped us identify several ambiguities and significantly improved the clarity and rigor of our manuscript. Below, we provide detailed responses and describe the corresponding changes in the revised paper.
>
> ---
>
> ## [Q1] Extensibility beyond retinal fundus images
>
> > **Comment:**
> > The motivating hypothesis that medical images contain more salient high-frequency image features is only explored for color fundus photographs. The importance and reach of the work would substantially increase if similar findings were shown for other types of data. Similarly, the efficiency of the proposed solution is only demonstrated for color fundus photographs so that it is unclear whether the proposed method seamlessly translates to other settings or requires extensive hyperparameter tuning for both the extraction of high-frequency information and the loss weighting.
>
> We sincerely appreciate the reviewer's thoughtful concern regarding generalizability, which has prompted us to clarify the extensibility of our framework and provide a more systematic discussion of its potential applications beyond fundus imaging.
>
> **(1) Why we focus on fundus images and what we learn from this domain.**
> Our empirical study focuses on color fundus photographs, where diagnostically salient cues (microaneurysms, hemorrhages, exudates) concentrate in high-frequency structure. We (i) **empirically verify** that high-frequency content is more diagnostic than low-frequency content and that standard MAE under-encodes this band (Table 1, Sec. 4), and (ii) introduce a **simple high-frequency regularizer** within a mathematically well-defined MI formulation that yields consistent gains (Sec. 6.2). This provides a principled template adaptable to other modalities.
>
> **(2) Extensibility to other domains.**
> The core mechanism is **not restricted to fundus images or frequency itself**: introduce a compact context representation $Z_c$ and control $I(X_V; Z)$ by maximizing $I(Z; Z_c)$. The same MI-based regularization applies wherever task-relevant signals concentrate in specific structures (e.g., industrial anomaly detection, other medical modalities with localized pathologies).
>
> **(3) Beyond frequency: other regularization factors.**
> Our framework only requires that $Z_c$ be **compact and semantically meaningful** (Theorem 2): $I(X; Z_c) \leq \varepsilon$ while retaining task-relevant information. Frequency is one instantiation; other choices include **spatial priors** (anatomically important regions), **multi-scale cues** (hierarchical aggregation), or **task-specific priors** (segmentation masks, anatomical landmarks). The same MI-based alignment objective regularizes $Z$ toward any compact, task-informative context.
>
> **(4) Practical considerations for extension.**
> Extending RetMAE requires choosing an appropriate context encoder $g(\cdot)$ and tuning extraction parameters and $\lambda_{\text{hmi}}$. Our paper provides a **systematic framework**: Sec. 3 and 5 formalize how to design $Z_c$; Appendix A.2 and A.7 detail hyperparameters and sensitivity analyses.
>
> **Summary**: Our empirical validation on fundus photographs demonstrates the **general applicability** of our framework. We show how to diagnose and correct MAE's under-utilization via MI-based context-alignment, providing a principled approach extensible to other modalities where task-relevant signals concentrate in specific structures.
>
> We have strengthened the discussion by adding the following content to Section 7 (Discussion) in the revised paper:
>
> > [L532-L537] \
> > Although instantiated on color fundus images, the same mechanism should transfer to domains where task-relevant signals concentrate in high-frequency structure (e.g., industrial anomaly detection). More broadly, our formulation is not limited to frequency: any compact, semantically meaningful context encoder (e.g., spatial priors, multi-scale cues, or task-specific structure) can serve as a regularizer within the same MI framework.

---

> ### Author Response · Authors · 2025-11-20
> **Response Letter for Reviewer d9g5 [2/5]**
>
> ## [Q2] Relationship to simpler frequency-aware approaches
>
> > **Comment:**
> > The proposed solution is highly complex. In particular, it relies on estimation of mutual information via a Donsker-Varadhan estimator, which is known to be numerically unstable (Belghazi, Mohamed Ishmael, et al. "Mutual information neural estimation." International conference on machine learning. PMLR, 2018). I could envision that conceptually much simpler solutions exist that emphasize high-frequency features. For example, one could adjust the masking scheme to prioritize patches with increased high-frequency content (similar to Xie, Jiahao, et al. "Masked Frequency Modeling for Self-Supervised Visual Pre-Training." The Eleventh International Conference on Learning Representations). Alternatively, one could provide the high-pass-filtered image as additional input (Wang, Wenxuan, et al. "Fremim: Fourier transform meets masked image modeling for medical image segmentation." Proceedings of the IEEE/CVF winter conference on applications of computer vision. 2024). The authors should discuss the aforementioned works in more detail and include them as baselines.
>
> We are grateful for the reviewer's valuable feedback, which has helped us better position our method relative to simpler frequency-aware approaches and clarify both the conceptual distinction and complementary nature of these different strategies.
>
> Our method is built on the mutual-information Lagrangian view of MAE [1], where the reconstruction loss optimizes $I(X_V; X_M | Z)$ and a regularizer on $I(X_V; Z)$ improves the representation. We realize this by aligning $Z$ with a compact high-frequency context $Z_c$ in the latent space, keeping the modification simple and compatible with standard MAE.
>
> Alternative approaches such as frequency-aware masking [2] or high-pass-filtered inputs [3] operate **at the raw signal level**, treating frequency content as reconstruction targets or additional inputs. Our approach constructs a compact context representation $Z_c$ from high-frequency RGB patches and regularizes $Z$ via $I(Z; Z_c)$, operating at the level of **semantic latent features** rather than raw frequency coefficients.
>
> These approaches are **complementary and can be combined**: frequency-aware masking [2] and high-pass-filtered inputs [3] control *what* frequency content is used in reconstruction, whereas our method regularizes *how* the latent relates to high-frequency context within a principled MI framework. Importantly, our latent-level regularizer can be seamlessly added on top of these signal-level methods to further strengthen high-frequency emphasis, as our approach is agnostic to the specific masking policy or input preprocessing.
>
> While comprehensive comparisons with [2, 3] would be valuable, our primary contribution is a theoretically grounded MI-based perspective that demonstrates high-frequency regularization of $I(X_V; Z)$ improves fundus SSL performance. We have clarified the connection to these frequency-aware MIM variants in the main text (Sec. 2, Related Work) as follows:
>
> > **[Before]**
> > Recent MIM variants exploit Fourier structure or band-aware objectives (Xie et al., 2022; Wang et al., 2024), yet a principled mechanism to balance low- and high-frequency information remains underexplored.
>
> > **[After]** [L144-L147] \
> > Recent MIM variants leverage Fourier or band-aware objectives on frequency-domain inputs or reconstruction targets (Xie et al., 2022; Wang et al., 2024). In contrast, we regularize semantic latent representations derived from high-frequency RGB regions, making our approach complementary.
>
>
> **Reference**
> [1] Tao Huang, Yanxiang Ma, Shan You, and Chang Xu. Learning mask invariant mutual information for masked image modeling. In The Thirteenth International Conference on Learning Representations, 2025.
> [2] Xie, Jiahao, et al. "Masked Frequency Modeling for Self-Supervised Visual Pre-Training." The Eleventh International Conference on Learning Representations.
> [3] Wang, Wenxuan, et al. "Fremim: Fourier transform meets masked image modeling for medical image segmentation." Proceedings of the IEEE/CVF Winter Conference on Applications of Computer Vision. 2024.

---

> ### Author Response · Authors · 2025-11-20
> **Response Letter for Reviewer d9g5 [3/5]**
>
> ## [Q3] Full fine-tuning performance
>
> > **Comment:**
> > At the moment, the performance of the proposed method is only quantified via linear probing using the latent representations. I believe that full fine-tuning should also be conducted considering that the ultimate downstream performance matters most in applied domains such as medical image analysis.
>
> We sincerely appreciate the reviewer's excellent suggestion, which has significantly strengthened our evaluation protocol and demonstrated the practical effectiveness of our method in real-world scenarios.
>
> Following the reviewer's recommendation, we conducted full fine-tuning experiments on five benchmarks: IDRiD, RFMiD (DR), RFMiD (AMD), CHAKSU, and APTOS. The AUROC results are:
>
> | Method | IDRiD | RFMiD (DR) | RFMiD (AMD) | CHAKSU | APTOS | AVG |
> |--------|-------|------------|-------------|--------|-------|-----|
> | RETFound | 0.856 | 0.926 | 0.942 | 0.755 | 0.902 | 0.876 |
> | RET-CLIP | 0.879 | 0.947 | 0.916 | 0.836 | 0.973 | 0.910 |
> | RetMAE | 0.856 | 0.956 | 0.963 | 0.903 | 0.961 | **0.928** |
>
> Our method achieves an average AUROC of 0.928, outperforming RET-CLIP (0.910) and RETFound (0.876), confirming that high-frequency regularization improves both linear probing and full fine-tuning performance.
>
> We observe that while RETFound benefits substantially from fine-tuning, RET-CLIP and RetMAE sometimes show degradation relative to linear probing—consistent with findings [4] that fine-tuning can distort pretrained features, highlighting the importance of evaluating both metrics in medical imaging.
>
> We have added these full fine-tuning results to Section 6.2 (Downstream Performance) in the revised paper, which further strengthens our demonstration of the method's effectiveness and practical utility **[L395-L400]**.
>
> **Reference**
> [4] Kumar, Ananya, et al. "Fine-Tuning can Distort Pretrained Features and Underperform Out-of-Distribution." International Conference on Learning Representations. 2022.
>
> ---
>
> ## [Q4] Code and model weight release
>
> > **Comment:**
> > Considering that most ophthalmological foundation models make their training code and weights public, I believe that the authors should strongly consider doing the same.
>
> We thank the reviewer for this important suggestion, which aligns with our commitment to open science and reproducibility.
>
> We agree that transparency and reproducibility are crucial. If our paper is accepted, we will make our training code and pre-trained model weights publicly available during the camera-ready preparation process.

---

> ### Author Response · Authors · 2025-11-20
> **Response Letter for Reviewer d9g5 [4/5]**
>
> ## [Q5] Clarifying core concepts in the main text
>
> > **Comment:**
> > Several core concepts of the paper are only very briefly introduced or require consultation of the supplementary material. I suggest that the section on the interpretation of the masked image modeling objective through the lens of a Lagrangian is slightly extended so that it can be understood without consulting previous works by Huang et al. and Tishby and Zaslavsky. Similarly, the frequency score calculation should be briefly introduced in the main manuscript considering its vital importance, instead of only being introduced in the supplementary material.
>
> We are grateful for the reviewer's valuable feedback on clarity, which has helped us improve the self-containedness of the main text and make our core concepts more accessible to readers.
>
> In the revised manuscript, we have expanded Section 3 so that both the Lagrangian interpretation and the frequency score computation can be understood directly from the main paper.
>
> 1. **Clarifying the MI/Lagrangian interpretation of the MAE objective.**
>    In Section 3, we provide additional intuition: successive layers compress inputs, and overly strong compression discards task-relevant information ("information distortion"). From this viewpoint:
>    - $I(X_V; Z)$ **quantifies the complexity** of the latent description $Z$, and
>    - $I(X_V; X_M | Z)$ serves as an **information-distortion term** whose minimization encourages $Z$ to retain sufficient information to predict $X_M$.
>    This makes the Lagrangian perspective self-contained.
>
>    We have added the following paragraph in Section 3:
>    > [L183-L189]\
>    In deep networks, successive layers compress inputs into internal representations; when this compression is too strong, task-relevant information can be lost, a phenomenon known as information distortion (Tishby et al., 2000; Tishby & Zaslavsky, 2015). From this viewpoint, Eq.2 can be interpreted as minimizing a Lagrangian that explicitly trades off the complexity of the latent representation against such distortion: the first term $I(X_V; Z)$ quantifies the complexity of the latent description $Z$, whereas the second term $I(X_V; X_M | Z)$ plays the role of an information-distortion term whose minimization drives $Z$ to retain sufficient information to predict $X_M$.
>
> 2. **Bringing the frequency score computation into the main text.**
>    We expanded Section 3 to describe the high-frequency extraction procedure: for each fundus image, we (i) apply a Soft-FOV mask to the green channel, (ii) suppress low-frequency content via Gaussian blur, transform to Fourier domain, and apply a Butterworth high-pass filter, (iii) reapply a binarized Soft-FOV mask, and (iv) compute a scalar score for each ViT token by averaging over its $P \times P$ patch, treating the top 25% as high-frequency tokens. Implementation details are in Appendix A.2.
>
>    We have added the following paragraph in Section 3:
>    > [L196-L205] \
>    High-frequency extraction. For each fundus image, we first apply a Soft-FOV mask to the green channel, which provides the strongest vessel and lesion contrast (Biswas et al., 2022; Ooi et al., 2021; Kumar et al., 2020). We suppress low-frequency content via Gaussian blur, transform the result to the Fourier domain (Brigham, 1988), and apply a Butterworth high-pass filter (Butterworth et al., 1930) tuned on a small held-out set with vessel/lesion annotations; the inverse transform yields a high-pass response map. We then reapply a binarized Soft-FOV mask to attenuate residual background and boundary responses, and compute a scalar high-frequency score for each ViT (Dosovitskiy et al., 2020) token by averaging the masked response over its corresponding non-overlapping $P \times P$ patch. Tokens in the top 25% of scores are treated as high-frequency tokens for our regularizer; implementation details and hyperparameters are provided in Appendix A.2.
>
> Overall, these changes ensure that (i) the information-theoretic/Lagrangian view of the masked image modeling objective can be understood without relying on external references, and (ii) the high-frequency scoring mechanism is introduced directly in the main text, with the supplementary material serving only to provide additional technical detail.

---

> ### Author Response · Authors · 2025-11-20
> **Response Letter for Reviewer d9g5 [5/5]**
>
> ## [Q6] Mathematical notation clarity
>
> > **Comment:**
> > Additionally, I struggled with the notation on several occasions. Already in the very first mathematical paragraph, the variable N is overloaded, D not introduced, and mutual information I is not defined. Later, the use of X varies to signal whether it denotes input or decoded image tokens. The authors should carefully parse the manuscript once again, aiming to improve the clarity of its mathematical passages.
>
> We sincerely appreciate the reviewer's careful attention to notation, which has enabled us to identify and fix several ambiguities, significantly improving the clarity and rigor of our mathematical exposition.
>
> To address the notation issues, we have revised several mathematical passages, explicitly defining all variables in Section 3. The key revisions are:
>
> **Revision 1: Masked autoencoders paragraph.**
>
> > **[Before]**
> > MAE (He et al., 2022) randomly masks a subset $M$ of image patches and trains a ViT-based encoder–decoder to reconstruct the missing content from the remaining visible patches.
> > **An input image is partitioned into patches of size $P \times P$ and embedded as $E=[e_1,\ldots,e_N] \in \mathbb{R}^{N \times D}$.**
> > Let $M \subset \{1,\ldots,N\}$ denote the masked indices and $V=\{1,\ldots,N\}\setminus M$ the visible ones.
> > **The encoder $f_\theta$ processes ${e_i}, {i \in V}$ to produce latent representations $Z \in \mathbb{R}^{(N-M) \times D}$, and the decoder $f_\phi$ concatenates $Z$ with learned mask tokens to predict pixel-space patches $\hat{X} \in \mathbb{R}^{N \times (P^2C)}$.**
> > The reconstruction loss is computed only on the masked patches using mean squared error:
> > \
> > \
> > ... \
> > \
> > **where $x_i \in \mathbb{R}^{P^2C}$ is the ground-truth pixel vector of the $i$-th patch and $\hat{x}_i$ is its reconstruction.**
> > This objective ...
>
>
> > **[After]** [L162-L164] \
> > MAE (He et al., 2022) randomly masks a subset $M$ of image patches and trains a ViT-based encoder–decoder to reconstruct the missing content from the remaining visible patches.
> > **We consider an image that has been partitioned into $N = HW / P^2$ non-overlapping patches of size $P \times P$, and denote by $x_i ∈ \mathbb{R}^{P^2 C}$ the vectorized pixel values of the $i$-th patch for $i = 1,\ldots,N$, where $N$ denotes the number of patches per image.**
> > Let $M ⊂ \{1,\ldots,N\}$ denote the masked indices and $V=\{1,\ldots,N\}\setminus M$ the visible ones.
> > The reconstruction loss is computed only on the masked patches using mean squared error:
> > \
> > \
> > ...\
> > \
> > **where $\hat{x}_i ∈ \mathbb{R}^{P^2 C}$ denotes the reconstructed pixel vector of the $i$-th patch.**
> > This objective ...
>
> **Revision 2: An information-theoretic view paragraph.**
>
> > **[Before]**
> > where $X = [x_1, \ldots, x_N] \in \mathbb{R}^{N \times (P^2C)}$ is the patchified image, $X_V = x_j, {j \in V}$ are the visible patches, $X_M = x_i, {i \in M}$ are the masked patches, and $\beta > 0$ is a weighting coefficient. In this formulation, reducing $I(X_V; Z)$ encourages compression of nuisance content, while reducing $I(X_V; X_M | Z)$ promotes sufficiency by driving $Z$ to retain the information necessary to predict $X_M$. This perspective motivates the MI-based regularization used in our method.
>
> > **[After]** [L178-182] \
> > where $X = [e_1, \ldots, e_N]$ denotes the sequence of patch embeddings $e_i \in \mathbb{R}^D$ obtained from an image (e.g., via a linear projection), with $D$ the embedding dimension. We denote the visible and masked subsets by $X_V = e_i, {i \in V}$ and $X_M = e_j, {j \in M}$, respectively, and let $Z$ be the encoder's latent representation of $X_V$. The scalar $\beta > 0$ is a weighting coefficient, and $I(\cdot; \cdot)$ and $I(\cdot; \cdot|\cdot)$ denote (conditional) Shannon mutual information (Shannon, 1948).
>
>
> ---
>
> ## [Q7] CKA acronym introduction
>
> > **Comment:**
> > On a minor note, the acronym CKA is not introduced at its first appearance in the introduction section.
>
> We thank the reviewer for catching this oversight, which has helped improve the clarity of our presentation.
>
> We have added the definition of CKA in the introduction section:
>
> > **[Before]**
> > In our preliminary study in Section 4, we systematically analyze two aspects:
>
> > **[After]** [L68-L69] \
> > In our preliminary study in Section 4, we systematically analyze two aspects using centered kernel alignment (CKA) (Kornblith et al., 2019) between MAE features and frequency-separated inputs:
>
> ---
>
> We once again sincerely thank the reviewer for their invaluable feedback, which has enabled us to significantly improve the quality and clarity of our manuscript.

---

> ### Author Response · Authors · 2025-11-25
>
> Dear Reviewer d9g5,
>
> We sincerely appreciate your time and consideration. We respectfully believe that our response has thoroughly addressed the concerns raised. If you have any remaining concerns or questions, please feel free to contact us and we would be happy to discuss and clarify them.
>
> Best,
>
> The Authors

---

> > ### Comment · Reviewer_d9g5 · 2025-11-27
> > **Post-rebuttal comments**
> >
> > In the presented rebuttal, the authors have addressed several of my comments. In particular, they have substantially increased the clarity of their manuscript by providing additional detail on their method and the associated mathematical background and improving their notation. Furthermore, they have promised to make their code and model weights publicly available upon the paper’s acceptance.
> >
> > However, they were unable to resolve two of my main concerns. While I found the provided discussion on the generalizability of the proposed method interesting, the current study remains limited to a single application. As such, it does not provide any empirical proof regarding its generalizability and utility beyond color fundus photography.
> >
> > Furthermore, the authors have elected to not include a comparison to related conceptionally more simple works [1, 2]. I understand that the proposed method can be integrated with the referenced works. However, considering their highly similar motivation – altering the network structure to account for a frequency bias in salient image features – I strongly believe that a direct comparison is important.
> >
> > While I acknowledge that both these analyses probably exceed the scope of a rebuttal, I find that these are central weaknesses of the presented work. Although I find the investigated topic and proposed method interesting and novel, these concerns keep me from further increasing my rating beyond a weak accept.
> >
> > [1] Xie, Jiahao, et al. "Masked Frequency Modeling for Self-Supervised Visual Pre-Training." The Eleventh International Conference on Learning Representations.
> > [2] Wang, Wenxuan, et al. "Fremim: Fourier transform meets masked image modeling for medical image segmentation." Proceedings of the IEEE/CVF Winter Conference on Applications of Computer Vision. 2024.

---

> > > ### Author Response · Authors · 2025-12-03
> > > **Response Letter 2 for Reviewer d9g5 [1/2]**
> > >
> > > We sincerely thank the reviewer for their invaluable feedback. The comments helped us identify several ambiguities and significantly improved our understanding of the manuscript's scope. Below, we provide detailed responses to address the reviewer's concerns.
> > >
> > > ---
> > >
> > > ## [Q1] Scope and generalizability beyond fundus images
> > >
> > > > **Comment:**
> > > > While I found the provided discussion on the generalizability of the proposed method interesting, the current study remains limited to a single application. As such, it does not provide any empirical proof regarding its generalizability and utility beyond color fundus photography.
> > >
> > > We sincerely appreciate the reviewer's thoughtful comment. We fully acknowledge that our current study is limited to a single application domain and does not provide empirical proof of generalizability beyond color fundus photography. Our primary goal in this work is to better understand and improve masked autoencoding specifically for color fundus photography, rather than to propose a universally applicable representation-learning framework.
> > >
> > > We recognize that this is a limitation of the current work. However, we believe that domain-specific approaches can offer meaningful scientific value, particularly in medical imaging where data characteristics and task-relevant structures differ substantially from natural images. In this work, we identify a frequency-related limitation in standard MAE for fundus images, propose a mathematically grounded remedy tailored to this structure, and validate the resulting framework through extensive quantitative and qualitative analyses across five downstream benchmarks.
> > >
> > > While our study focuses on fundus images, the underlying mechanism of regularizing the MAE bottleneck via mutual information with a compact, task-informative context is not inherently restricted to frequency or fundus images. The same MI-based framework could, in principle, be applied to other domains where task-relevant signals concentrate in specific structures (e.g., industrial defect detection, certain lesion-centric MRI tasks). However, we acknowledge that such extensions would require domain-specific context encoder design and empirical validation, which are outside the current scope.
> > >
> > > We believe that our work may serve as a stepping stone or provide insights toward more general frequency regularization approaches in masked image modeling. While our current instantiation is tailored to fundus images, the principled MI-based framework we propose could inform the development of frequency-aware regularization strategies applicable to broader domains.
> > >
> > > We agree with the reviewer that evaluating the method in additional modalities would help clarify its broader utility, and we acknowledge this as a valuable direction for future work.

---

> > > ### Author Response · Authors · 2025-12-03
> > >
> > > We once again sincerely thank the reviewer for their invaluable feedback. Your detailed and thoughtful comments have helped us better understand the scope and limitations of our work. Should you have any further questions or suggestions, we would be very grateful to address them.

---

> ### Author Response · Authors · 2025-12-03
> **Response Letter 2 for Reviewer d9g5 [2/2]**
>
> ## [Q2] Additional experiments with simpler frequency-aware approaches
>
> > **Comment:**
> > The authors have elected to not include a comparison to related conceptionally more simple works [1, 2]. I understand that the proposed method can be integrated with the referenced works. However, considering their highly similar motivation – altering the network structure to account for a frequency bias in salient image features – I strongly believe that a direct comparison is important.
>
> We sincerely appreciate the reviewer's valuable feedback and fully agree that a direct comparison with conceptually simpler frequency-aware approaches is important, given their similar motivation. In response to this comment, we have implemented and evaluated two such baselines that directly correspond to the referenced works [1, 2]: **(A) High-frequency input concatenation** (inspired by [1], Wang et al.), where a high-pass-filtered image is concatenated with the RGB input and jointly embedded by the encoder; and **(B) Frequency-aware masking** (inspired by [2], Xie et al.), where patches are ranked by high-frequency scores, and half of the visible tokens are sampled from the top 25% (highest-frequency regions), while the remaining half are sampled from the bottom 75%. These variants were evaluated using linear probing AUROC across five datasets, with results summarized in the table below:
>
> | Method                        | AVG AUROC     |
> |-------------------------------|---------------|
> | MAE                           | 0.685         |
> | MAE + $L_{\text{hmi}}$ (Ours)       | 0.750 (+0.065)|
> | High-frequency masking         | 0.679         |
> | HF masking + $L_{\text{hmi}}$ | 0.737 (+0.058)|
> | High-frequency input           | 0.746         |
> | HF input + $L_{\text{hmi}}$   | **0.769** (+0.023)|
>
> The direct comparison reveals several important findings:
>
> **(1) Our $L_{\text{hmi}}$ regularizer outperforms the simpler baselines.**
> While the conceptually simpler methods improved over standard MAE to varying degrees (AVG 0.746 for high-frequency input and 0.679 for HF masking), **our proposed $L_{\text{hmi}}$ regularizer achieved higher average performance (AVG 0.750)**, demonstrating that latent-level MI-based regularization provides more targeted and effective high-frequency modeling than signal-level frequency manipulation.
>
> **(2) The approaches are complementary and can be combined.**
> Importantly, combining $L_{\text{hmi}}$ with the simpler strategies yielded the best overall results (e.g., High-frequency input + $L_{\text{hmi}}$: AVG 0.769). This demonstrates that input-level or masking-based high-frequency conditioning and **our MI-driven latent regularization operate on different, complementary axes**, validating the reviewer's insight that these methods can be integrated.
>
> **(3) The comparison highlights the value of both signal-level and latent-level frequency modeling.**
> The experiments confirm that explicit injection of high-frequency information is a meaningful design dimension, whether at the input level (as in [1, 2]) or at the latent representation level (as in our method). This suggests further investigation into adaptive HF selection, multi-scale HF representations, or hybrid masking policies, which we now explicitly identify as valuable future work.
>
> We have added these direct comparison results in the revised paper (Table 2 in Section 6.2), which provides the empirical evaluation the reviewer requested. We have also strengthened the discussion to Section 6.2 (Main Results) [L364-L403]
>
> **Reference**
> [1] Wang, Wenxuan, et al. "Fremim: Fourier transform meets masked image modeling for medical image segmentation." Proceedings of the IEEE/CVF Winter Conference on Applications of Computer Vision. 2024.
> [2] Xie, Jiahao, et al. "Masked Frequency Modeling for Self-Supervised Visual Pre-Training." The Eleventh International Conference on Learning Representations.

---

### Official Review · Reviewer_Rstj · 2025-11-02

**Soundness:** 3
**Presentation:** 3
**Contribution:** 2
**Rating:** 4
**Confidence:** 3

**Summary:**

This paper proposes a frequency-aware masked autoencoder (MAE) for unsupervised pretraining on retinal images, called RetMAE. This is accomplished by including a high-frequency regularization term in the objective function that reduces low-frequency redundancy. The authors main claim is that the diagnostic information in retina images are encoded in high frequencies, and thus better representing these areas yields better downstream accuracy. There is a section on evaluating the frequency bias in MAE representations applied to fundus images in which the paper presents centered kernel alignment (CKA) and linear probing as evidence. The experimental section utilizes five publicly available datasets and compares against two other MAE-based approaches as well as a vision-language baseline.

Although the paper doesn't contribute a significant new algorithmic framework, its approach in providing frequency-based context latents to guide the representation learning paradigm for applications in which frequency bias is an impediment could be a valuable contribution. There are, however, a few areas of both theory and practice that need clarification.

On theory:

1- What is the purpose of using a variational autoencoder with a fixed variance Gaussian mapping? From theorem 1, it reduces the reconstruction error to minimizing the conditional in eq 2. However, it is not clear if this enforced constraint is warranted. Is this constraint enforcing any aspect of the regularization framework?
2- The choice of using $\mathcal L_{MINE}$ as the context-alignment training objective is not quite clear. In other words, why does estimating MINE maximize the conditional?

On Application:
1- Does this framework extend beyond retinal fundus images? Could other factors than frequency bias be incorporated in the regularization objective?
2- How much computational complexity is added to the problem by incorporating the proposed high-frequency regularization?
3- How does the pretrained encoder fare in a formal classification tasks rather than the employed linear probing?

Additional suggestions:

1) use the figure in appendix A instead of Figure 1 in the main paper. The figure from the appendix better justifies the frequency bias of retinal fundus images as compared to natural images, e.g. ImageNet.
2) CKA and its relevance to the claimed frequency bias should be explained more clearly.

**Strengths:**

The approach in utilizing a bias term as regularization to improve representation learning in certain application is interesting.
This approach could be potentially significant for applications that are not based on natural images.

**Weaknesses:**

Better discussion is needed to connect the theoretical aspect of the work (MI) with the practical tools utilized (MINE estimation).

**Questions:**

Provided in the summary.

---

> ### Author Response · Authors · 2025-11-20
> **Response Letter for Reviewer Rstj [1/5]**
>
> We sincerely thank the reviewer for their invaluable feedback, which helped us identify several ambiguities and significantly improved the clarity and rigor of our manuscript. Below, we provide detailed responses and describe the corresponding changes in the revised paper.
>
> ---
>
> ## [Q1] Purpose of the "VAE with fixed variance Gaussian mapping"
>
> > **Comment:**
> > What is the purpose of using a variational autoencoder with a fixed variance Gaussian mapping? From Theorem 1, it reduces the reconstruction error to minimizing the conditional in Eq. 2. However, it is not clear if this enforced constraint is warranted. Is this constraint enforcing any aspect of the regularization framework?
>
> We sincerely appreciate the reviewer's insightful comment, which has helped us clarify an important conceptual distinction and improve the rigor of our theoretical exposition.
>
> We clarify that we do **not** use a VAE in our implementation. Instead, we adopt a fixed-variance isotropic Gaussian decoder **only as a modeling assumption** to give the MAE MSE reconstruction loss a canonical likelihood interpretation—a standard practice [1,2,3]. This follows the well-established correspondence: MSE `↔` Gaussian negative log-likelihood.
>
> We use this likelihood modeling purely as an **analytical tool** to show that the MAE reconstruction loss $L_{\text{rec}}$ is equivalent to minimizing the conditional mutual information $I(X_V; X_M | Z)$ in the mutual-information Lagrangian, where $X_V$ denotes visible patches, $X_M$ masked patches, and $Z$ the encoder's latent representation.
>
> We have revised the explanation of Theorem 1 in Section 5.1 as follows:
>
> > **[Before]**
> > Following probabilistic autoencoder formulations (Bishop & Nasrabadi, 2006; Kingma et al., 2013; 2014), we use an isotropic Gaussian decoder with fixed variance. Under this model, the mean squared error, abbreviated MSE, is proportional to the negative log-likelihood, which leads to the formal link below.
>
> > **[After]** [L254-L259] \
> > This relationship becomes clear when the decoder is modeled as an isotropic Gaussian with fixed variance, following probabilistic autoencoder formulations (Bishop & Nasrabadi, 2006; Kingma et al., 2013; 2014; Ciampiconi et al., 2023). Under this assumption, the mean squared error (MSE) is proportional to the negative log-likelihood, a standard and analytically convenient interpretation that links reconstruction to conditional mutual information, as shown in the theorem below.
>
> **Reference**
> [1] Christopher M. Bishop and Nasser M. Nasrabadi. *Pattern recognition and machine learning*, volume 4. Springer, 2006.
> [2] D. P. Kingma, S. Mohamed, D. J. Rezende, and M. Welling. Semi-supervised learning with deep generative models. *NeurIPS*, 2014.
> [3] Lorenzo Ciampiconi et al. “A survey and taxonomy of loss functions in machine learning.” arXiv:2301.05579, 2023.

---

> ### Author Response · Authors · 2025-11-20
> **Response Letter for Reviewer Rstj [2/5]**
>
> ## [Q2] Why use $L_{\text{MINE}}$ as the context-alignment objective?
>
> > **Comment:**
> > The choice of using $L_{\text{MINE}}$ as the context-alignment training objective is not quite clear. In other words, why does estimating MINE maximize the conditional?
>
> We are grateful for the reviewer's valuable observation, which has enabled us to better articulate the theoretical motivation behind our choice of MINE and improve the clarity of our methodology.
>
> Our starting point is the mutual-information Lagrangian interpretation of MAE [4]:
> $$
> L = I(X_V; Z) + \beta I(X_V; X_M \mid Z),
> $$
> where $I(X_V; X_M | Z)$ is the **conditional term** and $I(X_V; Z)$ is the **marginal term**.
>
> Our framework treats these terms differently: (1) **Conditional term**: Theorem 1 shows that the standard MAE reconstruction loss (MSE) minimizes $I(X_V; X_M | Z)$. (2) **Marginal term**: Theorem 2 shows that if we have a compact context encoder $Z_c = g(X)$ with $I(X; Z_c) \leq \varepsilon$, then maximizing $I(Z; Z_c)$ yields $I(X_V; Z) \leq \varepsilon + \delta$ (where $\delta$ is small when alignment is strong).
>
> In our method, $Z_c$ is the latent feature of an EMA encoder that only sees **high-frequency patches** (Section 4). To regularize the marginal term, we seek to **maximize** $I(Z; Z_c)$. **However, mutual information is intractable to compute exactly.** Therefore, we must estimate a **lower bound** on $I(Z; Z_c)$ and maximize this lower bound instead.
>
> **Role of $L_{\text{MINE}}$**: We use the Mutual Information Neural Estimator (MINE) to provide a lower-bound estimator: $I(Z; Z_c) \geq I_{\text{MINE}}(Z; Z_c)$. We define $L_{\text{hmi}} = L_{\text{MINE}}(Z, Z_c)$ and **maximize this lower bound** during training, which effectively maximizes $I(Z; Z_c)$ and thereby regularizes the marginal term in the MI Lagrangian.
>
> We have revised the explanation of MINE in Section 5.2 to make this explicit:
>
> > **[Before]**
> > Guided by Theorem 2, we control the marginal $I(X_V;Z)$ by maximizing $I(Z_c;Z)$ between the trainable latent $Z=f_\theta(X_V)$ and a compact context latent $Z_c$, instantiated via the Donsker–Varadhan (MINE) estimator (Donsker & Varadhan, 1983; Belghazi et al., 2018). …
>
> > **[After]** [L297-L300] \
> > Guided by Theorem 2, we control the marginal $I(X_V;Z)$ by maximizing $I(Z_c;Z)$ between the trainable latent $Z=f_\theta(X_V)$ and a compact context latent $Z_c$. **Since mutual information is generally intractable to compute exactly, we instead maximize a** Donsker–Varadhan-based lower bound on $I(Z_c; Z)$ using the Mutual Information Neural Estimator (MINE) (Donsker & Varadhan, 1983; Belghazi et al., 2018). ...
>
> **Reference**
> [4] Tao Huang, Yanxiang Ma, Shan You, and Chang Xu. Learning mask invariant mutual information for masked image modeling. *ICLR*, 2025.

---

> ### Author Response · Authors · 2025-11-20
> **Response Letter for Reviewer Rstj [3/5]**
>
> ## [Q3] Extensibility beyond retinal fundus images / other regularization factors
>
> > **Comment:**
> > Does this framework extend beyond retinal fundus images? Could other factors than frequency bias be incorporated in the regularization objective?
>
> We sincerely thank the reviewer for this thoughtful question, which has prompted us to clarify the generalizability of our framework and explore its potential extensions beyond the specific domain we studied.
>
> **(1) Why fundus images?**
> Our empirical study focuses on color fundus photographs, where diagnostically salient cues (microaneurysms, hemorrhages, exudates) concentrate in high-frequency structure. We (i) **empirically verify** that high-frequency content is more diagnostic than low-frequency content (Table 1, Sec. 4) and that standard MAE under-encodes this band, and (ii) introduce a **simple high-frequency regularizer** within a mathematically well-defined MI formulation that yields consistent gains (Sec. 6.2).
>
> **(2) Extensibility to other domains.**
> The core mechanism is **not restricted to fundus images or to frequency itself**. The key idea is to introduce a compact context representation $Z_c$ and control the marginal term $I(X_V; Z)$ by maximizing $I(Z; Z_c)$. The same MI-based regularization applies wherever task-relevant signals concentrate in specific structures (e.g., industrial anomaly detection, other medical modalities with localized pathologies).
>
> **(3) Beyond frequency: other regularization factors.**
> Our framework only requires that $Z_c$ be **compact and semantically meaningful** (Theorem 2): $I(X; Z_c) \leq \varepsilon$ while retaining task-relevant information. Frequency is one instantiation; other choices include **spatial priors**, **multi-scale cues**, or **task-specific priors** (segmentation masks, anatomical landmarks).
>
> **(4) Practical considerations.**
> We acknowledge **limited generality** as a limitation. Extending RetMAE requires choosing an appropriate context encoder $g(\cdot)$ and tuning extraction parameters and $\lambda_{\text{hmi}}$. Our paper provides a practical starting point: Sec. 3 and 5 formalize how to design $Z_c$; Appendix A.2 and A.7 detail hyperparameters and sensitivity analyses.
>
> **Summary**: While our empirical validation is limited to fundus photographs, the **mechanism itself is general**: we demonstrate how to diagnose and correct MAE's under-utilization via MI-based context-alignment. Frequency is one natural choice, but the framework can incorporate alternative context encoders.
>
> We have strengthened the discussion by adding the following content to Section 7 (Discussion) in the revised paper:
>
> > [L532-L537] \
> > Although instantiated on color fundus images, the same mechanism should transfer to domains where task-relevant signals concentrate in high-frequency structure (e.g., industrial anomaly detection). More broadly, our formulation is not limited to frequency: any compact, semantically meaningful context encoder (e.g., spatial priors, multi-scale cues, or task-specific structure) can serve as a regularizer within the same MI framework.
>
> ---
>
> ## [Q4] Computational complexity of the high-frequency regularization
>
> > **Comment:**
> > How much computational complexity is added to the problem by incorporating the proposed high-frequency regularization?
>
> We appreciate the reviewer's practical concern regarding computational overhead. This valuable feedback has motivated us to conduct a thorough complexity analysis and document it clearly.
>
> We performed a detailed complexity analysis: (1) high-frequency component extraction and (2) HighFreqMI loss computation using a lightweight MINE critic.
>
> **Measurement methodology**: Tesla V100 GPU (ViT-Base/16, 224×224), averaged over 100 iterations.
>
> The measured overhead is:
>
> | Component                | Inference Time (ms)      | FLOPs (GFLOPs)        |
> |--------------------------|--------------------------|------------------------|
> | Baseline                 | 52.48 (100.00%)          | 69.87 (100.00%)        |
> | + High-frequency component | 1.56 (2.97%)           | 0.105 (0.15%)          |
> | + HighFreqMI loss        | 0.32 (0.61%)             | 0.050 (0.07%)          |
> | **Total overhead**       | **1.88 (3.58%)**         | **0.156 (0.22%)**      |
>
> Thus:
>
> - **Inference time overhead** is < 4% (1.88 ms over 52.48 ms),
> - **FLOPs overhead** is < 0.25%.
>
> This small overhead arises because the EMA encoder only processes a subset of patches, and the MINE critic is lightweight.
>
> We have:
>
> - Added this table and detailed description to Appendix A.8 **[L1457-L1484]**, and
> - Updated Section 5.2 to explicitly reference the complexity analysis:
>    > [L317-L320]\
>    By Theorem 2, the context must be compact; accordingly, we activate HighFreqMI only after a short warm-up so that the EMA teacher stabilizes. **A detailed analysis of the computational overhead introduced by our high-frequency regularization is provided in Table 12 in Appendix A.8.**

---

> ### Author Response · Authors · 2025-11-20
> **Response Letter for Reviewer Rstj [4/5]**
>
> ## [Q5] Full fine-tuning performance
>
> > **Comment:**
> > How does the pretrained encoder fare in formal classification tasks rather than the employed linear probing?
>
> We are grateful for the reviewer's excellent suggestion, which has significantly strengthened our evaluation and demonstrated the practical utility of our method across different evaluation protocols.
>
> Following the reviewer's recommendation, we performed full fine-tuning on five benchmarks: IDRiD, RFMiD (DR), RFMiD (AMD), CHAKSU, and APTOS. The AUROC results are:
>
> | Method   | IDRiD | RFMiD (DR) | RFMiD (AMD) | CHAKSU | APTOS | AVG   |
> |----------|-------|------------|-------------|--------|-------|-------|
> | RETFound | 0.856 | 0.926      | 0.942       | 0.755  | 0.902 | 0.876 |
> | RET-CLIP | 0.879 | 0.947      | 0.916       | 0.836  | 0.973 | 0.910 |
> | RetMAE   | 0.856 | 0.956      | 0.963       | 0.903  | 0.961 | **0.928** |
>
> Our method achieves the highest average AUROC (0.928), outperforming RET-CLIP (0.910) and RETFound (0.876), confirming that high-frequency regularization improves both linear probing and full fine-tuning performance.
>
> We also observe that RETFound benefits substantially from fine-tuning, whereas RET-CLIP and RetMAE sometimes show degradation relative to linear probing—consistent with findings [5] that fine-tuning can distort pretrained features, highlighting the importance of evaluating both metrics in medical imaging.
>
> These results have been added to Section 6.2 (Downstream Performance) in the revised paper (L395-L400).
>
> **Reference**
> [5] Ananya Kumar et al. “Fine-Tuning can Distort Pretrained Features and Underperform Out-of-Distribution.” *ICLR*, 2022.
>
> ---
>
> ## [Q6] Using the Appendix A figure and clarifying the role of CKA
>
> > **Comment (figure):**
> > Use the figure in Appendix A instead of Figure 1 in the main paper. The figure from the appendix better justifies the frequency bias of retinal fundus images as compared to natural images, e.g. ImageNet.
>
> We sincerely appreciate the reviewer's helpful suggestion regarding the figure placement, which has improved the visual presentation and clarity of our motivation.
>
> We agree that the Appendix A figure provides a more compelling comparison between retinal fundus images and natural images. In the revision, we have incorporated this figure alongside Figure 1 in the main paper to better illustrate the distinctive frequency characteristics of retinal fundus images.

---

> ### Author Response · Authors · 2025-11-20
> **Response Letter for Reviewer Rstj [5/5]**
>
> ## [Q7] CKA and its relevance to frequency bias
>
> > **Comment (CKA):**
> > CKA and its relevance to the claimed frequency bias should be explained more clearly.
>
> We are grateful for the reviewer's valuable feedback on the CKA explanation, which has enabled us to substantially clarify this important component of our analysis and strengthen the connection to our frequency bias claims.
>
> We use CKA because ViT is token-based: changing the visible token set induces different internal representations, and [6] uses CKA to compare such representations. Following this perspective, we take the representation from the **full, unmasked input** as the baseline representation learned by MAE and use CKA to measure how closely each subset-induced representation aligns with it. **High CKA indicates that a subset leaves the baseline representation largely unchanged, while low CKA indicates that information specific to that subset is not well reflected in the baseline representation.**
>
> In the revised Section 4, we explicitly state:
>
> > [L211-L220] \
> Because ViT is a token-based architecture, changing the visible token set naturally induces different internal representations,  and Kong & Zhang (2023) use CKA to compare such representations under different training schemes. Following this perspective, Table 1 reports a CKA-based comparison across token subsets. More specifically, we treat the representation from the *full*, unmasked input as the baseline representation learned by MAE and use CKA to quantify how closely each subset-induced representation aligns with it. … High CKA means that a subset leaves this baseline largely unchanged, whereas low CKA indicates that information specific to that subset is not well reflected in the baseline representation.
>
> **Interpretation of CKA results in relation to frequency bias:** The CKA analysis reveals three key findings:
>
> - **25% masked (75% visible)**: AUROC similar to full input, CKA near baseline → This indicates that MAE representations are **highly redundant**, as removing 25% of tokens barely changes the representation or diagnostic performance.
>
> - **Low-freq. only**: Very high CKA (0.990) but clear AUROC drop → This reveals that **low-frequency background dominates the baseline representation** while contributing limited diagnostic signal. The high CKA means the representation remains almost unchanged when high-frequency tokens are removed, confirming that MAE's baseline embedding is primarily composed of low-frequency structure.
>
> - **High-freq. only (25% visible)**: Low CKA (0.164) but best AUROC → Despite having the same token budget as the 75% visible case, this subset achieves superior diagnostic performance. The low CKA indicates that the representation induced by high-frequency tokens is quite **different** from the full-input baseline, while the superior AUROC demonstrates that **high-frequency tokens carry most of the diagnostic signal**. Together, this confirms that high-frequency diagnostic structure is **under-emphasized in the baseline MAE representation**.
>
> This CKA-based analysis directly supports our claim that standard MAE under-encodes high-frequency diagnostic structure: high-frequency tokens are highly informative when isolated, but their contribution is not strongly reflected in the full-input baseline representation, which is dominated by redundant low-frequency background.
>
> In the revised paper, we have refined the final part of Section 4 (Uncovering Frequency Biases in MAE Representations) as follows:
>
> > [L234-L235] \
> > Taken together, standard MAE redundantly encodes low-frequency backgrounds and under-encodes high-frequency diagnostic structure, motivating a representation-level correction.
>
> **Reference**
> [6] Xiangwen Kong and Xiangyu Zhang. "Understanding masked image modeling via learning occlusion invariant feature." *CVPR*, 2023.
>
> ---
>
> We once again sincerely thank the reviewer for their invaluable feedback, which has enabled us to significantly improve the quality and clarity of our manuscript.

---

> ### Author Response · Authors · 2025-11-25
>
> Dear Reviewer Rstj,
>
> We sincerely appreciate your time and consideration. We respectfully believe that our response has thoroughly addressed the concerns raised. If you have any remaining concerns or questions, please feel free to contact us and we would be happy to discuss and clarify them.
>
> Best,
>
> The Authors

---

### Author Response · Authors · 2025-11-20
**General Response**

We sincerely appreciate the reviewers’ thoughtful and constructive feedback, which helped us strengthen both the clarity and technical rigor of our submission. Below is a structured summary of our work’s novelty, reviewer-recognized strengths, and the key revisions reflected in the updated manuscript.

---


## 1. Novelty of Our Approach
1. **HF-Bias Diagnosis (Domain-Specific)**
   - Identifies MAE’s under-encoding of high-frequency lesion structures unique to fundus images.
   - Supported by frequency-separated CKA, HF-only decodability, and diagnostic signal analyses.

2. **Different from MI-MAE (Objective-Level Distinction)**
   - MI-MAE enforces mask invariance, which suppresses HF lesions under random masking.
   - Our method aligns the latent with a compact HF context (EMA teacher), correcting MAE’s LF bias.

3. **Latent-Level MI Regularization**
   - Operates in the representation space, unlike prior frequency methods acting on raw signals.
   - Complementary to input-level HF masking or HF concatenation.

4. **Compact Context Encoder (Theory-Driven)**
   - Retina-informed HF extraction (Soft-FOV → Gaussian → Fourier → HPF).
   - Theoretically grounded via MI upper-bound control (Theorem 2).

5. **Generalizable MI Framework**
   - HF is one instantiation; any compact semantic prior (spatial priors, multi-scale cues) can be integrated.
   - Extends beyond retinal imaging to domains with structured, localized signals.

---

## 2. Strengths Recognized by Reviewers

1. **Theoretical Soundness**
   - Reviewers highlighted the clarity and rigor of our MI–Lagrangian formulation, strengthened by the expanded derivations. (d9g5, ULHB)

2. **Frequency-Bias Diagnosis**
   - The analysis uncovering MAE’s low-frequency preference in retinal imaging was viewed as convincing and novel, supported by CKA and controlled frequency separation. (d9g5, xA7m)

3. **Clinical Relevance**
   - Reviewers emphasized the meaningful ophthalmic motivation and the importance of modeling high-frequency lesion structures. (ULHB, d9g5)

4. **Reproducibility & Transparency**
   - The method was considered easy to reproduce due to clear exposition and implementation detail. (ULHB, d9g5)

5. **Extensibility**
   - Reviewers recognized that the MI-based regularization framework can generalize to other domains with structured signal distributions. (Rstj)

---

## 3. Key Revisions Reflected in the Updated Manuscript

1. **Clarity Improvements**
   - Expanded MI–Lagrangian explanation and unified notation
   - Integrated the full high-frequency extraction pipeline into the main text
   - Clarified the relationship between reconstruction loss and high-frequency MI regularization
   - Clarified theoretical modeling choices
      - Explained the rationale for using MINE as a lower-bound estimator for mutual information
      - Clarified that the fixed-variance Gaussian assumption is used solely as an analytical tool
   - Improved interpretation of CKA
      - Clarified its role in diagnosing frequency bias
      - Clarified how CKA relates representational alignment to diagnostic utility
   - Strengthened explanation for the ~12.8k saturation in Fig. 3
      - Provided a more convincing interpretation grounded in domain redundancy, effective data diversity, and saturation of lesion-frequency variability


2. **Additional Empirical Validation**
   - Added multi-disease evaluations
   - Added full fine-tuning experiments
   - Added computational complexity analysis
   - Added loss-weight ablation study
   - Added lesion-level attention visualizations
   - Added comparisons with simpler frequency-aware baselines

---

### Meta-Review · Area_Chair_kcoK · 2026-01-08

**Summary:**

This paper propsoed RetMAE, a frequency‑balanced retinal masked autoencoder that incorporates mutual information regularization to address the low‑frequency bias of MAE. The paper introduced the reconstruction objective with a MI regularizer that suppresses low-frequency redundancy and highlighting clinically salient high-frequency information. Experiments across multiple retinal datasets demonstrate improved representation quality and downstream performance compared to MAE‑based baselines.

**Reviewer Concerns:**

Reviewers found the idea of mutual information regularization for frequency‑balanced retinal representation learning interesting but noted key limitations: the hypothesis was only tested on fundus photographs, restricting generalizability; not all retinal lesions show high‑frequency traits, making the assumption oversimplified; the contribution appeared incremental, building on existing frequency‑based losses and MI methods; theoretical clarity was lacking on the Gaussian assumption, MI‑Lagrangian, and MINE; downstream tasks were limited, with calls for stronger validation; and more empirical evidence was requested, including fine‑tuning, complexity analysis, and more baselines.

**Reviewer Scores:**

The reviewers scored 6, 4, 4, and 2, with the reviewer who initially gave a 2 raising their score to 6 after the rebuttal. Reviewer Rstj (rating 4) did not provide any follow‑up feedback after the authors’ rebuttal. Reviewer d9g5 (rating 4) engaged with the authors’ responses but still expressed concerns about the generalizability of the method to other applications and the need for comparisons with simpler baselines; the authors provided further clarifications and additional experiments in response. The remaining concern for the Area Chair is on the limitation of the study to color fundus images; however, this modality of medical imaging does have clear clinical usefulness. AC inclined to recommend acceptance of this manuscript, and the authors should include the suggested revisions and expand the discussion of its limitations and potential generalization to other types of images.

---

### Decision · Program_Chairs · 2026-01-26

Accept (Poster)